



# Estimating surface melt in Antarctica from 1979 to 2022, using a statistically parameterized positive degree-day model

Yaowen Zheng[1], Nicholas R. Golledge[1], Alexandra Gossart[1], Ghislain Picard[2], and
Marion Leduc-Leballeur[3]

[1]Antarctic Research Centre, Victoria University of Wellington, Wellington, New Zealand
[2]Univ. Grenoble Alpes, CNRS, Institut des Géosciences de l'Environnement (IGE), UMR 5001, Grenoble, France
[3]Institute of Applied Physics "Nello Carrara", National Research Council, 50019 Sesto Fiorentino, Italy

**Correspondence:** Yaowen Zheng (yaowen.zheng@vuw.ac.nz)

**Abstract.**

Surface melt is one of the primary drivers of ice shelf collapse in Antarctica. Surface melting is expected to increase in the future as the global climate continues to warm, because there is a statistically significant positive relationship between air temperature and melt. Enhanced surface melt will negatively impact the mass balance of the Antarctic Ice Sheet (AIS) and,

5 through dynamic feedbacks, induce changes in global mean sea level (GMSL). However, current understanding of surface melt in Antarctica remains limited in past, present or future contexts. Continental-scale spaceborne observations of surface melt are limited to the satellite era (1979–present), meaning that current estimates of Antarctic surface melt are typically derived from surface energy balance (SEB) or positive degree-day (PDD) models. SEB models require diverse and detailed input data that are not always available and require considerable computational resources. The PDD model, by comparison, has fewer input

10 and computational requirements and is therefor suited for exploring surface melt scenarios in the past and future. The use of PDD schemes for Antarctic melt has been less extensively explored than their application to surface melting of the Greenland Ice Sheet, particularly in terms of a spatially-varying parameterization. Here, we construct a PDD model, force it only with 2-m air temperature reanalysis data, and parameterize it by minimizing the error with respect to satellite observations and SEB model outputs over the period 1979 to 2022. We compare the spatial and temporal variability of surface melt from our PDD

15 model over the last 43 years with that of satellite observations and SEB simulations. We find that the PDD model can generally capture the same spatial and temporal surface melt patterns. Although there were at most four years over/under- estimation on ice shelf regions in the epoch, these discrepancies reduce when considering the whole AIS. With the limitations discussed, we suggest that an appropriately parameterized PDD model can be a valuable tool for exploring Antarctic surface melt beyond the satellite era.

## 1 Introduction

Surface melting is common and well-studied over the Greenland Ice Sheet (GrIS) (e.g. Mernild et al., 2011; Colosio et al., 2021; Sellevold and Vizcaino, 2021), and is known to play an important role in the net mass balance of the ice sheet and changes in global mean sea level (GMSL), both now and in the past (e.g. Ryan et al., 2019). It is likely to become even more



important in the future. Even though Antarctica is currently much colder than Greenland, projected Antarctic near-surface
warming (e.g. Kittel et al., 2021) means that increased surface melting is to be expected over coming decades – both in terms
of area and frequency of melting. However, these are currently less understood over Antarctica than Greenland, either in the
past or at present. This is concerning as surface melting will likely become an increasingly important component of Antarctic
Ice Sheet (AIS) mass balance through this century and the next.

In recent decades, ice shelf collapse in Antarctica has been found to be related to surface melt. Following retreat that started
in 1940s (Rott et al., 1996), Larsen-A, a 4200 $km^2$ ice shelf in the Antarctic Peninsula, experienced a collapse of one third of
its area over only a few days in 1995(e.g. Rott et al., 1996; Doake et al., 1998; Rack and Rott, 2004), contributing to consistent
post-collapse mass loss in the region (e.g. Shuman et al., 2011). A few years later, in 2002, around 3200 $km^2$ of Larsen-B ice
shelf disintegrated after consistent retreat following the collapse of Larsen-A (Rack and Rott, 2004; van den Broeke, 2005).
The area of this ice shelf decreased rapidly after the collapse of Larsen-A to March 2002, from around 11512 $km^2$ to around
2667 $km^2$ (Rack and Rott, 2004). In 2008, three break-up events were observed in Wilkins Ice Shelf, Antarctic Peninsula,
which led to a combined reduction in ice shelf area of around 1805 $km^2$ (e.g. Humbert and Braun, 2008; Braun and Humbert,
2009; Scambos et al., 2009). In April 2009, partial collapse of the Wilkins Ice Shelf led to a further area reduction of 330 $km^2$
following the break-up events of 2008 (Rankl et al., 2017).

The collapses of Larsen A and B were found to be related to increased melt, following atmospheric warming across the
Antarctic Peninsula (e.g. Rott et al., 1996). The break-up events of Wilkins Ice Shelf in 2008 were suggested to be related to
surface meltwater, following increased surface melt there (Scambos et al., 2009). Prior to Larsen-A collapse, the mean surface
air temperature during the 1994–1995 summer had risen to 0.6 °C. Similarly, an increasing surface air temperature trend was
found prior to Larsen B collapse, with an even warmer summer record of 1.3 °C reported at Matienzo Base near the Larsen
Ice Shelf (Skvarca et al., 2004). Associated with these increased summer surface air temperature in the region were prolonged
melt days and more extensive surface meltwater (Skvarca et al., 2004). This intensification of surface melt has been suggested
as one of the contributors to ice shelf mechanical fragmentation (Glasser and Scambos, 2008).

Although the warming taking place over the Antarctic Peninsula has not been consistent over the past two decades (Turner
et al., 2016), surface melt has most likely been accelerated by the rapid increase of atmospheric temperatures through the late
20th century (Vaughan and Doake, 1996; Turner et al., 2005, 2016; Hogg and Gudmundsson, 2017). The atmospheric warming
in the Antarctic Peninsula during the late 20th century may also have contributed to acceleration of outlet glaciers in the region
(Tuckett et al., 2019). Moreover, the positive feedback of albedo, in which the absorption of shortwave radiation increases
when snow melts to water, amplifies this melting (Lenaerts et al., 2017). However, recent studies have found large inter-annual
variability of surface melt in Antarctica with no statistically significant trend (Kuipers Munneke et al., 2012; Johnson et al.,
2022). Projecting Antarctic surface melt is therefore still a challenge, partly because of uncertainties introduced by clouds
(Kittel et al., 2022), atmospheric rivers (e.g. Clem et al., 2022), or other localized climate phenomena.

Positive degree-day (PDD) schemes have been used in many Antarctic numerical ice sheet models (e.g. Winkelmann et al.,
2011; Larour et al., 2012) as empirical approximations to compute surface mass balance based on temperature and precipitation
fields. Several studies have been conducted with PDD models to explore surface melt in Antarctica, particularly in the Antarctic



Peninsula (e.g. Golledge et al., 2010; Barrand et al., 2013; Costi et al., 2018). The PDD model calculates surface melt based
on the temperature-melt relationship (Hock, 2005). Although it is empirical, it is often sufficient for estimating melt on a
catchment scale (Hock, 2003, 2005) because of its two physical bases: (a) the majority of the heat required for snow and ice
melt is primarily a function of near-surface air temperature, and (b) the near-surface air temperature is correlated with the
longwave atmospheric radiation, shortwave radiation and sensible heat fluxes (Ohmura, 2001).

A typical PDD model has two parameters: (1) the threshold temperature ($T_0$), which controls the decision of melt or no-melt,
and (2) the degree-day factor (DDF), which controls the amount of melt. Wake and Marshall (2015) reported that using the
Gaussian distribution sigma as a linear function of the monthly temperature can improve the performance of the PDD approach
in terms of accurately capturing surface melt on the AIS, compared to the traditional fixed sigma value. This suggests that
Antarctic surface melt can be estimated solely from monthly temperature. However, as the DDF is related to all terms of the
surface energy balance (SEB) (Hock, 2005), the PDD model may not be appropriate for universal usage unless the model can
incorporate DDFs that vary spatially and temporally (e.g. Hock, 2003, 2005; van den Broeke et al., 2010). This is because
topographic influences that are generally strongest in mountainous terrain, together with seasonal variations in radiation, can
introduce spatial and temporal variabilities of DDF, respectively (Hock, 2005). Spatial and temporal parameterisation of DDF
(model calibration), as well as model verification, therefore need to be considered. Moreover, compared to PDD model ap-
proaches established (e.g. Reeh, 1991; Braithwaite, 1995) and improved (Fausto et al., 2011; Jowett et al., 2015; Wilton et al.,
2017) for Greenland over many decades, such assessments for the PDD approach for the Antarctic domain are limited and a
spatially parameterized Antarctic PDD model has not yet been achieved.

In this study, we focus on constructing a computationally efficient PDD model to estimate surface melt in Antarctica through
the past four decades, by statistically optimizing the parameters of the PDD model individually in each Antarctic drainage
basin (Zwally et al., 2012) and ice shelf region. We use the European Centre for Medium-Range Weather Forecasts Reanalysis
v5 (ECMWF ERA5) (Hersbach et al., 2018a, b) 2-m air temperature as input and compare the simulated presence of melt
to satellite observations of melt days from three satellite products and the Regional Atmospheric Climate Model version
2.3p2 (RACMO2.3p2) surface melt simulations. We then examine the distributions of melt days and melt volume from PDD
experiments that use varying model parameters against satellite-based and RACMO2.3p2 estimations. Following this, we use
the PDD model to estimate and analyse the surface melt in Antarctica in terms of occurrence and amount from 1979 to 2022.

## 2 Data

### 2.1 Reanalysis data

The dataset we use in this study is the ECMWF ERA5 reanalysis (Hersbach et al., 2018b) (Table 1). It has hourly data for three-
dimensional (pressure level) atmospheric fields (Hersbach et al., 2018a) and on a single level for atmosphere and land-surface
(Hersbach et al., 2018b). It replaced the previous ECMWF reanalysis product ERA-Interim in 2019 (Hersbach et al., 2020),
and has become the new state-of-the-art ECMWF reanalysis product for global and Antarctic weather and climate (Hersbach
et al., 2020; Gossart et al., 2019).

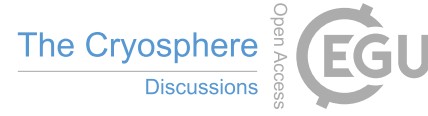

**Table 1.** Table of data that we use in this study.

| Data type | Time period | Spatial resolution | Temporal resolution | Reference |
|---|---|---|---|---|
| ERA5 reanalysis data[a] | 1979–2021 | $0.25° \times 0.25°$ lon/lat | Hourly | Hersbach et al. (2018b) |
| Zwally Antarctic drainage basin | – | 1000 m | – | Zwally et al. (2012) |
| Ice shelf collection | – | $30 \times 30$ km$^2$ | – | This study |
| Satellite SMMR and SSM/I[b] | 1979–2021 | $25 \times 25$ km$^2$ | Daily | Picard and Fily (2006) |
| Satellite AMSR-E[c] | 2002–2011 | $12.5 \times 12.5$ km$^2$ | Daily | Picard et al. (2007) |
| Satellite AMSR-2[c] | 2012–2021 | $12.5 \times 12.5$ km$^2$ | Daily | This study |
| RACMO2.3p2[d] | 1979–2021 | $27 \times 27$ km$^2$ | Monthly | Van Wessem et al. (2018) |

[a] The 2-m air temperature data are on single level (Hersbach et al., 2018b). [b] Satellite local acquisition times over Antarctica are around 6 am and 6 pm. [c] Satellite local acquisition times over Antarctica are around 12 am (descending) and 12 pm (ascending). [d] RACMO2.3p2 surface melt simulations.

The particular ERA5 product we use in this study is the 2-m air temperature data which has been evaluated and used previously for studies in Antarctica (e.g. Gossart et al., 2019; Tetzner et al., 2019; Zhu et al., 2021). Assessments have shown that ERA5 near-surface (or 2-m) air temperature data is a robust tool for exploring Antarctic climate (e.g. Gossart et al., 2019; Zhu et al., 2021). ERA5 performs better than its predecessor ERA-Interim, the Climate Forecast System Reanalysis (CFSR), and the Modern-Era Retrospective Analysis for Research and Applications, version 2 (MERRA-2) (Gossart et al., 2019). It is continuously being updated and is one of the most state-of-the-art reanalysis datasets available. However, compared to 48 automatic weather station (AWS) observations, it is reported to have a cold bias over the entire continent apart from the winter months (June-July-August) (Zhu et al., 2021). This cold bias is reported at 0.34 °C annually and at 1.06 °C during December-January-February (DJF) (Zhu et al., 2021).

## 2.2 Satellite data

The number of melt days retrieved from the satellite observations is used to parameterize the threshold temperature ($T_0$) for the PDD model. We use the satellite 42-year daily (once in two days before 1988) Antarctic surface melt dataset produced by Picard and Fily (2006) (Table 1). It contains daily observations as a binary of melt and no-melt on a $25 \times 25$ km$^2$ southern polar stereographic grid. It is obtained by applying the melt detecting algorithm (Torinesi et al., 2003; Picard and Fily, 2006) on the scanning Multichannel Microwave Radiometer (SMMR) and three Special Sensor Microwave Imager (SSM/I) observed passive-microwave data from the National Snow and Ice Data Center (NSIDC) (Picard and Fily, 2006). SMMR and SSM/I sensors are carried by sun-synchronous orbit satellites observing Earth at least twice per day (Picard and Fily, 2006). For Antarctica, the local acquisition times are around 6 am and 6 pm. The brightness temperature is the daily average of all the passes (those around 6 am and those around 6 pm). This dataset is being continually updated and is freely available via the website https://snow.univ-grenoble-alpes.fr/melting/. There is a reported data gap longer than a month during the period from December 1987 to January 1988 (Torinesi et al., 2003; Johnson et al., 2022), and we find additional missing data during the prolonged summer (from November to March) in 1986/1987 (13 days), 1987/1988 (44 days), 1988/1989 (8 days) and





1991/1992 (9 days), which are significantly longer than the length of the missing data period of the remaining 38 years (zero
115  or one day, Figure A1 in the Appendix A). We therefore omit those periods from our analysis.

More recently, there is a newly developed satellite melt day dataset which uses a similar algorithm as Torinesi et al. (2003);
Picard and Fily (2006) on the Advanced Microwave Scanning Radiometer for EOS (AMSR-E) and the Advanced Microwave
Scanning Radiometer 2 (AMSR-2) observed passive-microwave data from the Japan Aerospace Exploration Agency (JAXA,
Table 1). This dataset is on a $12.5 \times 12.5$ km$^2$ southern polar stereographic grid, which has a twice finer spatial resolution than
120  satellite SMMR and SSM/I product. It has twice-daily observations over Antarctica covering 2002 to 2011 (AMSR-E) and
2012 to 2021 (AMSR-2, Table 1). These sensors have a local acquisition time over Antarctica of around 12 am (descending)
and 12 pm (ascending).

### 2.3 Surface energy balance model data

SEB modeling is a physics-based numerical approach used to calculate the surface energy budget in order to estimate how
much energy is available for snow/ice melting. A number of studies have used SEB modeling forced by climate model outputs
and AWS data to assess surface melting on GrIS and AIS (e.g. Van den Broeke et al., 2011; Zou et al., 2021). To parameterize
the PDD model, we compare our ERA5 forced numerical experiments to the Antarctic surface melt simulations from the
RACMO2.3p2 (Van Wessem et al., 2018). The RACMO2.3p2 simulates Antarctic surface melt by solving the SEB model
which is defined as (Van Wessem et al., 2018):

$$\mathrm{Q}_M = \mathrm{SW}_\downarrow + \mathrm{SW}_\uparrow + \mathrm{LW}_\downarrow + \mathrm{LW}_\uparrow + \mathrm{SHF} + \mathrm{LHF} + \mathrm{G}_s \tag{1}$$

where $\mathrm{Q}_M$ is the energy available for melting, $\mathrm{SW}_\downarrow$ and $\mathrm{SW}_\uparrow$ are the downward and upward shortwave radiative fluxes, $\mathrm{LW}_\downarrow$
and $\mathrm{LW}_\uparrow$ are the downward and upward longwave radiative fluxes, SHF and LHF are the sensible and latent turbulent heat
fluxes and $\mathrm{G}_s$ is the subsurface conductive heat flux (Van Wessem et al., 2018).

The RACMO2.3p2 Antarctic surface melt simulations used here cover the time period from January 1979 to February 2021
with monthly temporal resolution and $27 \times 27$ km$^2$ spatial resolution (Table 1).

### 2.4 Interpolation and research domain

The spatially coarsest dataset used in this study is the ERA5 reanalysis data which is in 0.25° longitude × 0.25° latitude
geographic coordinates (Table 1). For consistency with the other data we analyse, we use the southern polar stereographic
coordinates instead of the geographic coordinates. We use the Climate Data Operators (CDO) (Schulzweida, 2021) to bilinearly
remap ERA5 reanalysis data from longitude-latitude geographic coordinates to NSIDC Sea Ice Polar Stereographic South
Projected Coordinate System (NSIDC, 2022) (hereafter "polar stereographic grid"). We use a spatial resolution of 30 km,
minimising the number of missing pixels and maximising the resolution. For consistency, we also use CDO to remap all data
we use in this study (Table 1) to the same $30 \times 30$ km$^2$ polar stereographic grid.

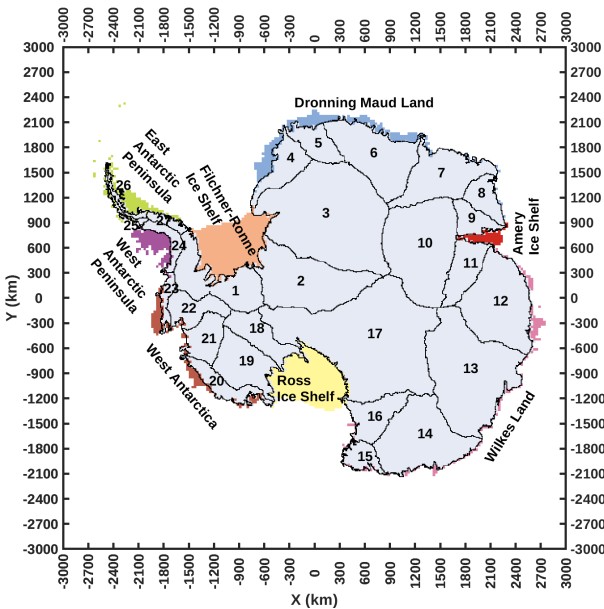

**Figure 1.** Map of the 27 Antarctic drainage basins (Zwally et al., 2012) and 8 ice shelf regions we use in this study. At the continental scale, 27 basins and 8 ice shelf regions are used. At the regional scale, we consider each of the basins and ice shelf regions individually. This map is also the mask matrix we use in this study. The mask matrices are on polar stereographic grid with 30 kilometer resolution.

Figure 1 shows the research domain of this study. At the continental scale, we look at the AIS and ice shelves. To parameterize
the model, estimate and analyse the surface melt in Antarctica spatially, we use the 27 Antarctic drainage basins defined by Zwally et al. (2012) and 8 regional collections of ice shelves defined in this study (Figure 1 and Table 1).

## 3 Methods

### 3.1 PDD model

Using an empirical relationship between air temperature and melt, temperature-index models are the most used method for
assessing surface melt of ice and snow due to their simplicity as they are only meteorologically forced by the air temperature (Hock, 2005). Not only does the simplicity of the approach enable fast run times and requires low computational resources, but the air temperature input data are also much easier to obtain than the full inputs (e.g. radiation fluxes, temperature, wind speed, humidity, ice/ snow density and surface roughness (van den Broeke et al., 2010)) required by the SEB model. If appropriately parameterized, the temperature-index approach offers accurate performance (Ohmura, 2001) and provides a robust surface melt
representation.

The PDD model calculates the water equivalent of surface snow melt (M, mm w.e.). It integrates the near-surface air temperatures above a predefined threshold, which are multiplied by the empirical DDF (mm w.e. $°\mathrm{C}^{-1}$ $\mathrm{day}^{-1}$). The adjusted PDD





model we use in this study can be written as:

$$\sum_{i=1}^{day} M = \frac{1}{24} DDF \sum_{i=1}^{day} \sum_{j=1}^{24} T^{\star}$$

$$T^{\star} = \begin{cases} T - T_0 & \text{if } T - T_0 > 0 \\ 0 & \text{otherwise} \end{cases} \tag{2}$$

where T is the hourly temperature and $T_0$ is the threshold temperature.

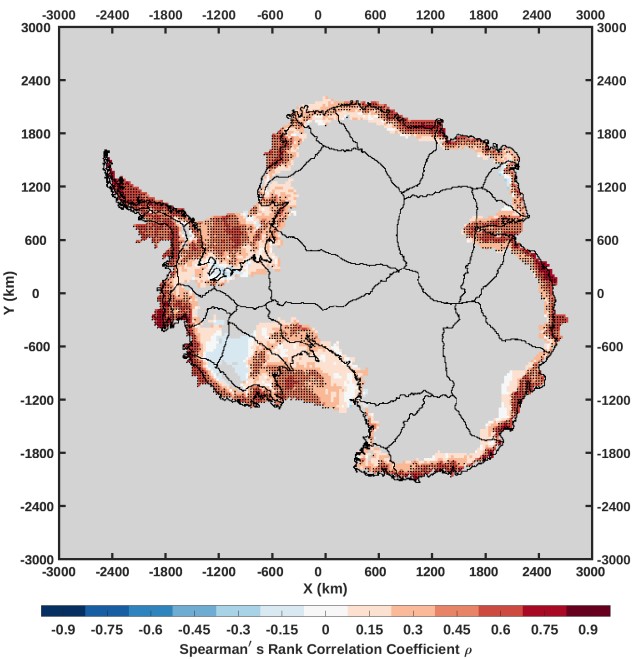

**Figure 2.** Correlation map between the mean DJF ERA5 2-m air temperature and the RACMO2.3p2 annual surface melt amount for the period from 1979/1980 to 2019/2020. It is calculated by the Spearman's rank correlation coefficient on each cell. Black dots mark the cells that the correlation are statistically significant (p < 0.05). Grey cells are either outside our research area (as shown in Figure 1) or have not ever melted during the period.

The positive relationship between 2-m air temperature and surface melt on Antarctic ice shelves (Trusel et al., 2015) allows us to use temperature to empirically estimate Antarctic surface melt via the PDD model. To assess this positive relationship, we calculate the Spearman's rank correlation between the mean summer (DJF) ERA5 2-m air temperature and the RACMO2.3p2 annual surface melt amount for the period from 1979/1980 to 2019/2020 (Figure 2). It shows that most of the cells in Antarctic

ice shelves and drainage basin coastal zones, apart from the Ross Ice Shelf or nearby basins (17, 18 and 19), have statistically significant (p < 0.05) positive correlations. Although the interior basins 19, 20 and 21 show negative correlations without statistical significance (p ≥ 0.05), the annual melt there is negligible compared to the ice shelves and coastal areas. Overall,





the correlation map shows a result consistent with Trusel et al. (2015): Antarctic ice-shelf near-surface temperature and surface melt are positively correlated, which can allow us to empirically construct a temperature-index model to explore surface melt
in Antarctica and especially Antarctic ice shelves.

## 3.2   Model parameterisation

### 3.2.1   Threshold temperature $T_0$

To parameterize the threshold temperature ($T_0$) for our PDD model, we firstly focus on the melt/no-melt signal. We use the ERA5 2-m air temperature data to force the model and run 101 numerical experiments with a set of $T_0$ ranging from -5.0 °C
to +5.0 °C with 0.1 °C intervals. We define a melt day ($MD^\star$) as a day during which there is at least one hour of ERA5 2-m air temperature exceeding the $T_0$. In each $T_0$ experiment, we calculate the total number of melt days from the 1st April of that year to the 31st March of the following year as the "annual number of melt days". The modified Equation 2 can be written as:

$$\text{Annual number of melt days} = \sum_{i=t_1}^{t_2} MD^\star$$

$$t_1 = 01 - April - Year$$

$$t_2 = 31 - March - (Year+1) \tag{3}$$

$$MD^\star = \begin{cases} 1 & \text{if at least one hour } T - T_0 > 0 \\ 0 & \text{otherwise} \end{cases}$$

Because the satellite melt day product of SMMR and SSM/I (Table 1) is retrieved from the local acquisition times around 6
am and 6 pm, we select the 6 am and 6 pm ERA5 2-m air temperature data and calculate the daily averages of the 6 am and 6 pm. For the satellite product from AMSR-E and AMSR-2 (Table 1), we repeat the calculations using the daily averages of the 12am and 12pm ERA5 2-m air temperature data as of their local acquisition times. Next, we calculate the result of Equation 3 in each $T_0$ experiment.

In order to obtain the optimal $T_0$, we calculate the RMSE between the time series of the annual number of melt days for the
satellite observations and the model experiments. As we treat each computing cell individually, all calculations are carried out on each cell independently in each iteration ($T_0$ experiment).

Next, we explore the optimal $T_0$ for the whole continent and by region. To do this, we multiply the mask matrices (cells inside the region have a value of one, and cells outside the region have a value of zero) by the RMSE of each $T_0$ experiment to generate the RMSE for each $T_0$ experiment on each region. The mask matrices for those regions are defined by multiplying
each mask matrix of the 38 regions of interest (Figure 1) by the mask matrix of the satellite observational area (Figure A2 in the Appendix A). Then we calculate the average of RMSE across all computing cells (RMSE per computing cell) in each targeted region in each $T_0$ experiment. Although these three satellite products have different time periods (SSMI and SSM/I covers the period from 1979/1980 to 2020/2021 (1986/1987–1988/1989 and 1991/1992 omitted), AMSR-E covers the period from 2002/2003 to 2010/2011 and AMSR-2 covers the period from 2012/2013 to 2020/2021), we assume their comparability





as these satellite products are derived from the same algorithm and threshold (Picard and Fily, 2006). We therefore calculate
the average of the regional-average RMSE across three satellites (hereafter, the regional RMSE). Finally, we define the optimal
$T_0$ of each targeted region where the $T_0$ experiment has the minimal regional RMSE.

### 3.2.2    Degree Day Factor DDF

To parameterize the DDF for our PDD model, we substitute the optimal $T_0$ found in Section 3.2.1 into the Equation 2, and
run a series of numerical experiments: we firstly set the DDF to 1 $\mathrm{mm\ w.e.\ °C^{-1}\ day^{-1}}$ then we iterate 241 times with 0.1
$\mathrm{mm\ w.e.\ °C^{-1}\ day^{-1}}$ increments.

     In order to address the optimal DDF, we repeat the calculations for the RMSE between the annual melt amount calculated
in each DDF experiment and the melt amount from RACMO2.3p2 simulations. Similarly, we define the optimal DDF where
the experiment has the minimal regional RMSE.

### 3.3    Significance testing

The two-sample Kolmogorov–Smirnov test (hereafter two-sample KS test) has been used in testing the significant difference
between two non-Gaussian climatic distributions when parametric tests are inappropriate (e.g. Deo et al., 2009; Zheng et al.,
2021). It has also been used as an alternative way to test the dissimilarity of climatic data as a validation of tests on statistical
parameters such as the mean (Zheng et al., 2021). The two-sample KS test non-parametrically tests the distributional dissim-
ilarity between two samples by quantifying the distance of two sample-derived empirical distribution functions (Lanzante,
2021). The null hypothesis is that the two samples are from the same continuous distribution. The test result returns a logical
index that either accepts or rejects the null hypothesis at the 5% significance level ($p < 0.05$).

     Limited by the duration of satellite era and reanalysis data, the annual data for each computing cell is no larger than 45 with
non-normality. To test the significance of the optimal $T_0$ and DDF, we therefore perform the two-sample KS tests between the
annual number of melt days/ melt amount from the satellite observations/ RACMO2.3p2 and from the PDD model $T_0$/ DDF
experiments. We define a 'same distribution cell' as a cell with no statistically significant evidence from the two-sample KS
test for the rejection of the null hypothesis (that the two samples are from the same continuous distribution). To quantify the
test result in each targeted region, we calculate the percentage of the same distribution cells for each $T_0$/ DDF experiment on
each targeted region. We specifically discuss and interpret the results of this test approach in Appendix B.

## 4    Results and discussion

### 4.1    Model parameterisation

Figure 3 shows the result of regional RMSE for each targeted region in each $T_0$ experiment. $T_0$ equal to -1.8 °C minimises
the regional RMSE for the whole continent, indicating that the PDD model with $T_0$ at -1.8 °C has the best agreement with the
satellite observations on estimating the annual number of melt days over the AIS and ice shelves. In Figure 3a-a, the RMSE



**Figure 3.** (a-a) to (b-l), red curves are the averages of the regional-average RMSE across all satellites along each $T_0$ experiment. There are 101 experiments covering $T_0$ from -5.0 °C to +5.0 °C. In each experiment, we calculate the RMSE between the PDD model and each satellite. Blue envelopes cover the span of the three individual satellite results, and the red curves are the averages of the three satellites results. Purple vertical lines mark the optimal $T_0$ suggested by the minimal RMSE. Black dash lines mark the rounded optimal $T_0$. As the RMSE range varies on each region because of the regionally varying surface melt, we set the varying y-axis for clarity.

at the point which $T_0$ equals to 0 °C has higher value than the RMSE at the optimal $T_0$ (-1.8 °C), showing the lower ability of the PDD model to estimate surface melt using the threshold $T_0 = 0$ °C. This is consistent with another study: Jakobs et al.



(2020) reported that there was a significant underestimation on surface melt events with a 72.5% unrecognizability by using a $T_0$ equal to 0 °C; on ice shelves in the East Antarctic Peninsula, this unrecognizability is $\sim 65.3\%$; in Dronning Maud Land, this unrecognizability is more conspicuous because $\sim 92.4\%$ melt days occurred below 0 °C. Taken together, a negative and spatially varying $T_0$ may be more appropriate for PDD models.

Figure 3 highlights that for a number of regions such as the whole Antarctic continent (Figure 3a-a), all ice shelves (Figure 3a-b), West and East Antarctic Peninsula (Figure 3a-f and a-g), Dronning Maud Land (Figure 3a-i) and Basin 26 (Figure 3b-k), the optimal $T_0$ is not sensitive to the location of RMSE minimum where the RMSE gradient is equal to zero and is flat around the optimal $T_0$. The dissimilarity of RMSE between the optimal $T_0$ and its nearby points are negligible, compared to the dissimilarity of RMSE between the optimal $T_0$ and its further tails. For example, for the East Antarctic Peninsula (Figure 3a-g), the minimal RMSE equals 18.79 which gives a $T_0$ at -2.1 °C, while its RMSE is 18.81 at its nearest integer $T_0 =$ -2 °C (Table B1 in the Appendix B) which is around 0.1% difference ($(1 - 18.79/18.81) \times 100\% \approx 0.1\%$) between the values of their RMSE. The RMSE differences between the optimal $T_0$ and their nearest integer $T_0$ are negligible with differences not exceeding 5% for 36 of 38 regions apart from the West Antarctica and Basin 5 (Figure B1 and Table B1 in the Appendix B). This could shed a light on the further simplification of the PDD model by rounding the $T_0$ to integers and grouping the same rounded $T_0$ regions to reduce the number of model parameters (Appendix B). However, these parameters are empirically defined by the statistics and there is no implied physical explanation for the value of them or the changes to RMSE before and after rounding. By its nature the PDD model is one dimensional, which is computationally efficient, and reducing the number of parameters will not change its basic behaviour or improve much on the computational efficiency. Furthermore, because the PDD parameters are related to all terms of the SEB (Hock, 2005), the optimal parameters given by the minimal RMSE from the experiments could be misleadingly precise. Rounding the parameters into integers avoids implying a level of precision that, even though they are defined by the parameterisation experiments, may be physically unrealistic. We therefore round the optimal $T_0$ and substitute the rounded optimal $T_0$ into the Equation 2 for the numerical experiments on DDF as described in the Section 3.2.2.

Figure 4 shows the result of regional RMSE for each region in each DDF experiment. At the continental scale for the whole Antarctic continent (Figure 4a-a), a DDF = 2.8 $\mathrm{mm\ w.e.\ °C^{-1}\ day^{-1}}$ together with a $T_0$ = -2.0 °C minimizes the RMSE between the PDD model estimated surface melt amount and the RACMO2.3p2 surface melt simulations. Our results suggest that the optimal DDF better estimates surface melt for the Antarctic basins than the ice shelf regions, because the RMSEs for the basins are relatively smaller (Table B2 in the Appendix B), although this may be due to the low melt amount in those basins.

Taking all individual basins together ("Drainage basins" as in Figure 4a-c), we see there are 93% computing cells at the optimal DDF for all basins showing statistically significant ($p < 0.05$) same distributions as that of RACMO2.3p2 (Figure B2a-c in the Appendix B). Moreover, the RMSE minimum also maximises the same two-sample KS cells (Figure B2a-c). This is interesting, because we can also see that the RMSE minimum in the $T_0$ experiments also maximises the same two-sample KS cells between the PDD model and the satellite observations with 86% computing cells (Figure B1a-c). This may lead to a single combination of $T_0$ and DDF used as PDD model parameters for all the Antarctic drainage basins ("Drainage basins").



**Figure 4.** (a-a) to (b-l), red curves are the regional-average RMSE along each DDF experiment. There are 241 experiments covering DDF from 1.0 mm w.e. $°C^{-1}$ $day^{-1}$ to 25.0 mm w.e. $°C^{-1}$ $day^{-1}$. In each experiment, we calculate the RMSE between the PDD model and RACMO2.3p2. Purple vertical lines mark the optimal DDF suggested by the minimal RMSE. Black dash lines mark the rounded optimal DDF. As the RMSE range varies on each region because of the regionally varying surface melt, we set the varying y-axis for clarity.

For ice shelves (Figure 4a-b), the optimal DDF is in agreement with the two-sample KS test maximum, where the RMSE minimum maximises the same KS cells (Figure B2a-b). However, the maximum of two-sample KS test for all ice shelves (Figure B2a-b) is lower than the Ross Ice Shelf (Figure B2a-d), West Antarctica (Figure B2a-e) and Dronning Maud Land





(Figure B2a-i). The largest drop is around 22% computing cells on the Ross Ice Shelf where we see its two-sample KS test
maximum is around 83% (Figure B2a-d). Different from the Antarctic drainage basins (Figure 4a-c) as we discussed above,
the large DDF variations across each ice shelf region (Figure 4a-d to a-k) may suggest a requirement for spatially distributed
PDD model parameters over ice shelf regions. Consistent with the rounding of optimal $T_0$, we also round the optimal DDF for
the PDD model.

The resulting PDD model parameters are listed in Table 2. Spatial maps for the distribution of those parameters are shown
in Figure B3 in the Appendix B.

**Table 2.** Table of PDD model parameters: $T_0$ (°C) and DDF (mm w.e. °C$^{-1}$ day$^{-1}$).

| Region | $T_0$ | DDF | Region | $T_0$ | DDF | Region | $T_0$ | DDF |
|---|---|---|---|---|---|---|---|---|
| Ross Ice Shelf | -1 | 8 | Basin 5 | -2 | 2 | Basin 17 | 0 | 5 |
| West Antarctica | -2 | 3 | Basin 6 | -1 | 8 | Basin 18 | -1 | 4 |
| West Antarctic Peninsula | -2 | 2 | Basin 7 | -1 | 20 | Basin 19 | -2 | 3 |
| East Antarctic Peninsula | -2 | 2 | Basin 8 | -1 | 10 | Basin 20 | -2 | 4 |
| Filchner-Ronne Ice Shelf | 0 | 17 | Basin 9 | -3 | 2 | Basin 21 | -2 | 4 |
| Dronning Maud Land | -2 | 3 | Basin 10 | -4 | 3 | Basin 22 | -3 | 1 |
| Amery Ice Shelf | -4 | 1 | Basin 11 | -2 | 5 | Basin 23 | -1 | 8 |
| Wilkes Land | -3 | 2 | Basin 12 | -3 | 3 | Basin 24 | -2 | 2 |
| Basin 1 | -1 | 12 | Basin 13 | -2 | 4 | Basin 25 | -1 | 6 |
| Basin 2 | 0 | 10 | Basin 14 | -2 | 4 | Basin 26 | -3 | 2 |
| Basin 3 | -1 | 5 | Basin 15 | -4 | 2 | Basin 27 | -3 | 2 |
| Basin 4 | -1 | 10 | Basin 16 | -3 | 3 | Antarctica | -2 | 3 |

## 4.2   Spatial and temporal variability of surface melt

In order to examine the spatial and temporal variability of surface melt derived from satellite and models, we calculate the mean,
standard deviation and trend for the PDD annual melt days and amount, the satellite annual melt days and the RACMO2.3p2
annual melt amount, as shown in Figure 5 and Figure 6. We see that the Antarctic Peninsula has both the largest surface
melt means, with around annual 70–90 melt days and up to 300–450 mm w.e. melt magnitude in coastal cells (Figure 5a, d,
and Figure 6a ,d), and standard deviations, given by the highest mean DJF 2-m air temperature (Figure 7b) and a large area
of low elevation (Figure 7a). In agreement with Liu et al. (2006); Kingslake et al. (2017), we also see lower latitude areas
have larger melt intensity than the remaining regions, and they usually correspond to relatively lower elevations (Figure 7a)
and higher temperatures (Figure 7b). Spatial features of Antarctic surface melt derived from satellite observations have been
explored by other studies (e.g. Liu et al., 2006; Johnson et al., 2022). In agreement with Johnson et al. (2022), we also find
that generally over the whole of Antarctica, both the number of melt days and the amount of melt (Figure 5a, d, and Figure 6a





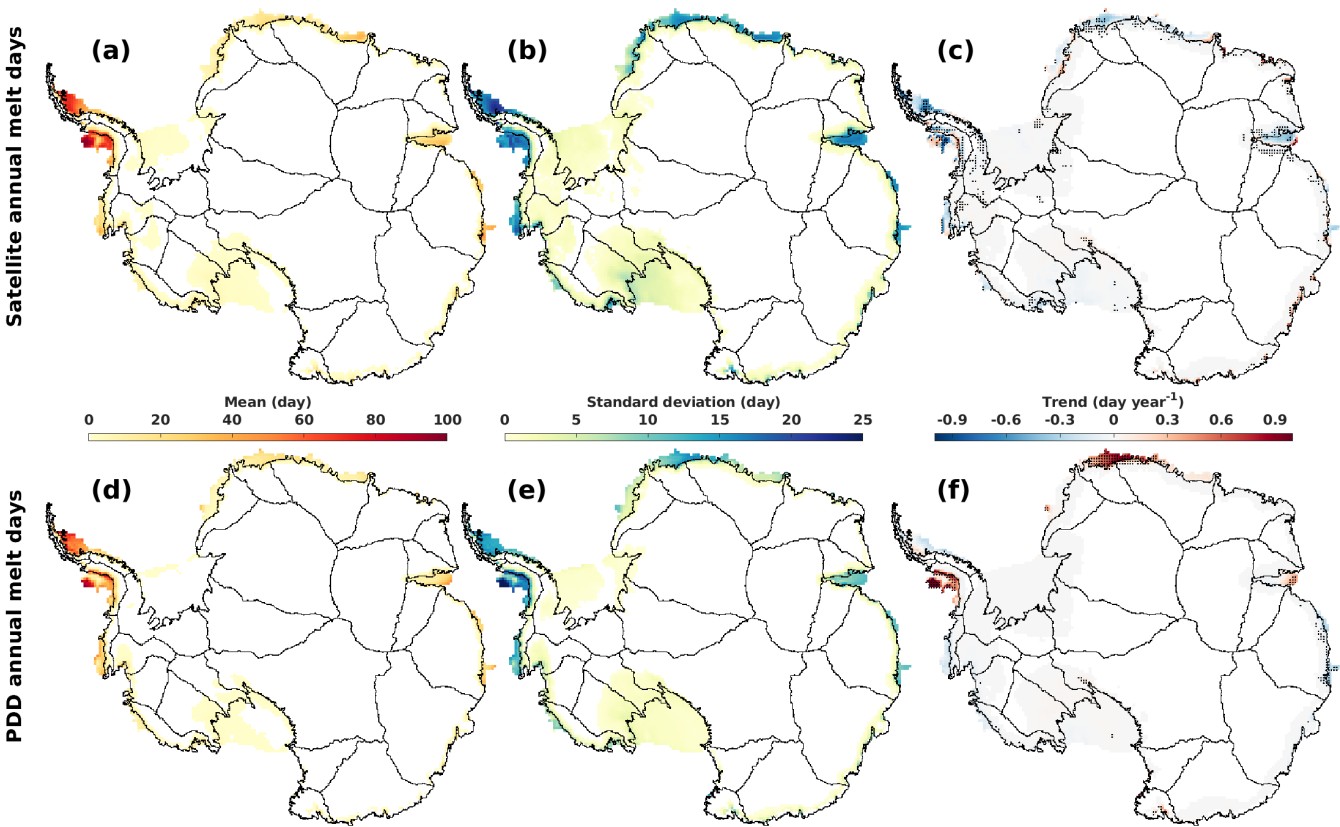

**Figure 5.** Mean, standard deviation and trend for satellite annual melt days (a, b and c) and PDD annual melt days (d, e and f) in the period from 1979/1980 to 2019/2020. For satellite and PDD annual melt days, 1986/1987, 1987/1988, 1988/1989 and 1991/1992 are omitted. Black dots in (c) and (f) mark the statistical significance (p < 0.05) of trend on each computing cell.

,d) decrease from the marine edges towards the interior of the continent, with the increasing surface orography (Figure 7a) and decreasing surface temperature (Figure 7b). By visual examination, we see that the PDD model has the ability to capture the main spatial patterns of surface melt when compared to the satellite observations and RACMO2.3p2 simulations, apart from
the absence of melt from the PDD model on south Filchner and south-west Ronne Ice Shelves where we see the occurrence of around one week satellite observed melt days with around 25 mm w.e. RACMO2.3p2 simulated magnitude. A similar result is found by quantitative examination from spatial RMSE and two-sample KS tests that are calculated between PDD and satellite/ RACMO2.3p2 (Figure C1 in the Appendix C). That the PDD model captures the main spatial patterns of melt is not surprising, given the statistically significant positive correlation between surface melt and 2-m air temperature in most of the Antarctic ice
shelf and coastal cells used in the calculations (Figure 2).

    Next, we examine the ability of the PDD model to capture the temporal variability of Antarctic surface melt. We see that computing cells experiencing relatively high surface melt also show large temporal variations (Figure 5b, e and Figure 6b ,e). The relatively large standard deviation together with the statistically non-significant (p ≥ 0.05) trend of surface melt may

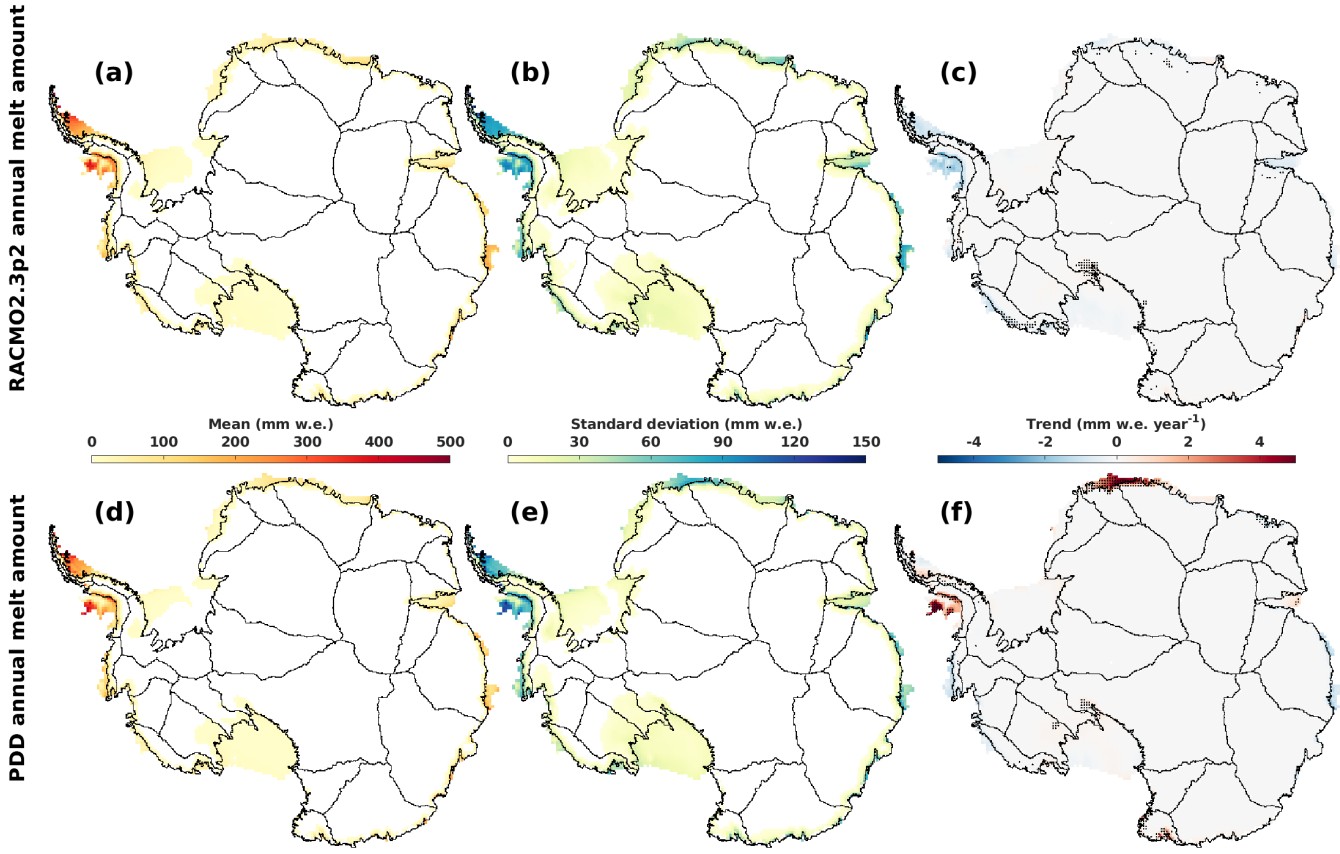

**Figure 6.** Mean, standard deviation and trend for RACMO2.3p2 annual melt amount (a ,b and c) and PDD annual melt amount (d, e and f) in the period from 1979/1980 to 2019/2020. Black dots in (c) and (f) mark the statistical significance (p < 0.05) of trend on each computing cell.

suggest a large inter-annual variability in melt events (Liu et al., 2006). The standard deviation of the PDD model-calculated

melt is approximately in agreement with the satellite observations and the RACMO2.3p2 simulations in most of the computing cells that experience relatively lower melt (annual melt days < 40 day and magnitude < 200 mm w.e.) (Figure 5b, e and Figure 6b ,e). However, we see a number of cells mostly located on ice shelves in the West Antarctic Peninsula and Dronning Maud Land that show opposite trends between the PDD model calculations and the satellite and RACMO2.3p2 (Figure 5c, f and Figure 6c ,f). This opposition suggests that the PDD model perhaps lacks some ability in capturing the trend of surface

melt in both occurrence and magnitude compared to the satellite observations and RACMO2.3p2, at least in those cells. Cells having the largest opposition are distributed in coastal West Antarctic Peninsula ice shelves, ice shelves in north-west Dronning Maud Land and the coastal margins of Basin 15 (Figure 5c, f and Figure 6c ,f), where a relatively large RMSE is also evident (Figure C1). The PDD model shows a statistically significant (p < 0.05) positive trend in those regions, whilst the trend is negative with/ without statistical significance according to satellite/ RACMO2.3p2. It is worth noting that on the marine edge





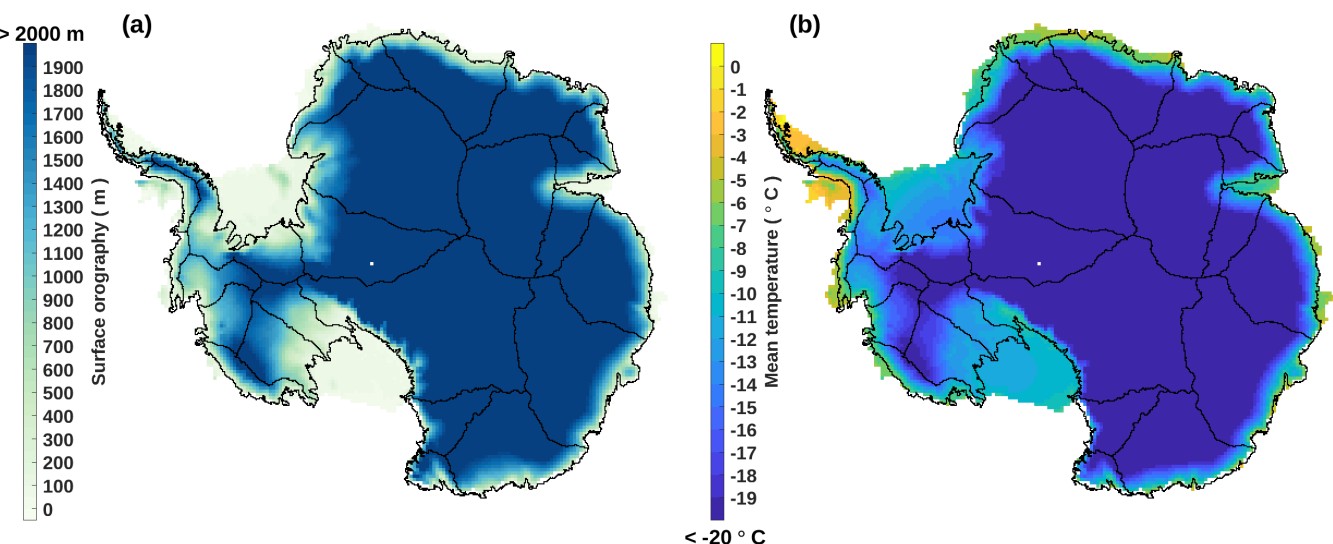

**Figure 7.** (a) ERA5 surface orography (m) for Antarctica. It is calculated by dividing the ERA5 surface geopotential height ($m^2\,s^{-2}$) (Hersbach et al., 2019) by 9.80665 ($m\,s^{-2}$). (b) Mean DJF ERA5 monthly 2-m air temperature for the period 1979/1980 – 2020/2021.

of Amery Ice Shelf, the satellite and PDD both show statistically significant ($p < 0.05$) positive trend whilst the RACMO2.3p2 shows an insignificant trend (Figure 5c, f and Figure 6c, f). Similarly, we find that the PDD and RACMO2.3p2 are in agreement of statistically non-significant ($p \geq 0.05$) zero or small negative ($\sim$ -1 mm w.e. year$^{-1}$) trend of annual melt amount over the ice shelves in Wilkes Land. However, for the annual melt days, West Ice Shelf (part of the ice shelves in Wilkes Land) shows a statistically significant ($p < 0.05$) negative trend suggested by the PDD model, whilst the satellite suggests a statistically

significant ($p < 0.05$) positive trend. We see there are differences in trends and the PDD model does not fully capture the trend as satellite or RACMO suggests. However, this does not necessarily require that we reject the PDD model, as the trend presented by the PDD model is a reflection of the trend of the input temperature (Figure C2), because of the linear relationship. The disagreement in trends, therefore, is actually between the satellite/RACMO2.3p2 and ERA5 2-m temperature, rather than between the satellite/RACMO2.3p2 and the PDD model itself.

To examine the temporal stability of the PDD parameters, we perform a time series analysis at the regional scale. Although it is shown in Figure 5 and Figure 6 that the disagreement in temporal variability in Basin 15 is not negligible compared to the remaining basins, the annual melt days and amount are relatively small compared to each ice shelf regions (Figure 5a, b and Figure 6a, b). We therefore gather all 27 drainage basins for the next stage of analysis.

Figure 8 and Figure 9 show the time series of cumulative melting surface (CMS) (day km$^2$) which is also known as a melt

index (e.g. Trusel et al., 2012) calculated by multiplying the annual number of melt cells by the cell area ($30 \times 30$ km$^2$), and the time series of melt amount (mm w.e.), for Antarctica (Figure 8a and Figure 9a), all ice shelves (Figure 8b and Figure 9b), all basins (Figure 8c and Figure 9c) and each ice shelf region (Figure 8d to k and Figure 9d to k). Although the inter-annual variability for ice shelves in the West Antarctic Peninsula and Dronning Maud Land are notably large compared to the remain-

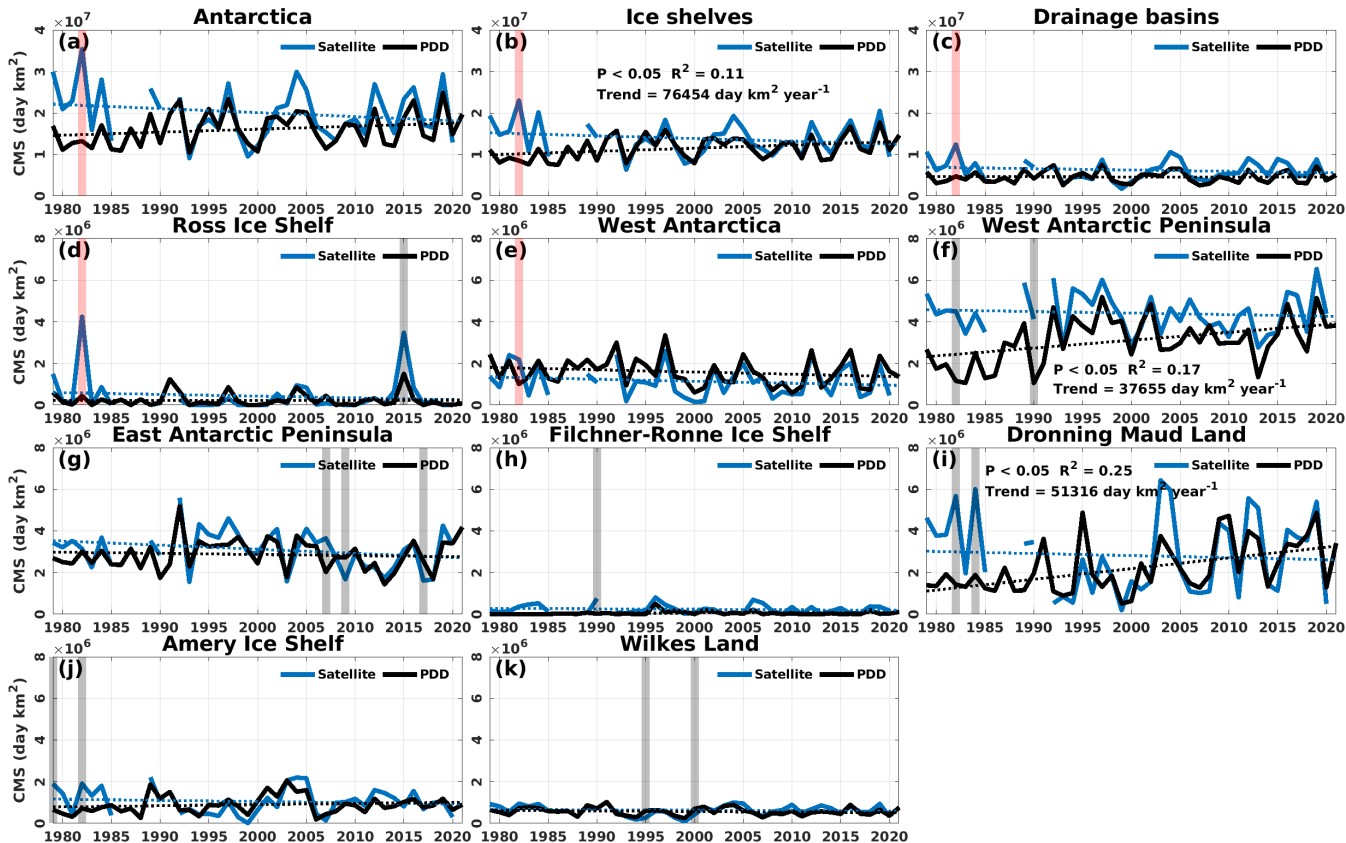

**Figure 8.** Blue curves are time series of satellite CMS (day km$^2$). Black curves are time series of PDD CMS (day km$^2$). Note that periods for satellite from 1986/1987 to 1988/1989 and 1991/1992 are omitted. Period from 2002/2003 to 2010/2011 for satellite is the average of SMMR and SSM/I, and AMSR-E. Period from 2012/2013 to 2020/2021 is the average of SMMR and SSM/I, and AMSR-2. PDD covers the period from 1979/1980 to 2021/2022. Dotted lines show the trends that are calculated by fitting ordinary least squares linear regressions during the overlapped period of PDD and satellite. Trends that are statistically significant (p < 0.05) are annotated by text with same color in the figure panel (e.g. Figure 8b). Shaded areas mark the years that have residuals larger than three (red) and 1.96 (grey) standard deviations (Figure C3).

ing collections of ice shelves (Figure 8 and Figure 9), they are the only two collections showing a statistically significant (p <

0.05) positive melting trend suggested by the PDD model (Figure 8f, i and Figure 9f, i). However, the trend calculated by the satellite and RACMO2.3p2 is negative without statistical significance. This could be explained by other players driving surface melting, such as the Southern Annular Mode (SAM) (Torinesi et al., 2003; Tedesco and Monaghan, 2009; Johnson et al., 2022) which explains ∼11%–36% of the melt day variability (Johnson et al., 2022). Besides, the PDD model generally captures the inter-annual variability of both the CMS and melt time series in those two regions, particularly the period after 1992/1993 ex-

cluding 2003/2004 where we observe a relatively large underestimation of melt by PDD (Figure 8i and Figure 9i) in Dronning Maud Land. This underestimation is not an outlier as it is within the 95% confidence interval (CI) of residuals (Figure C3i and



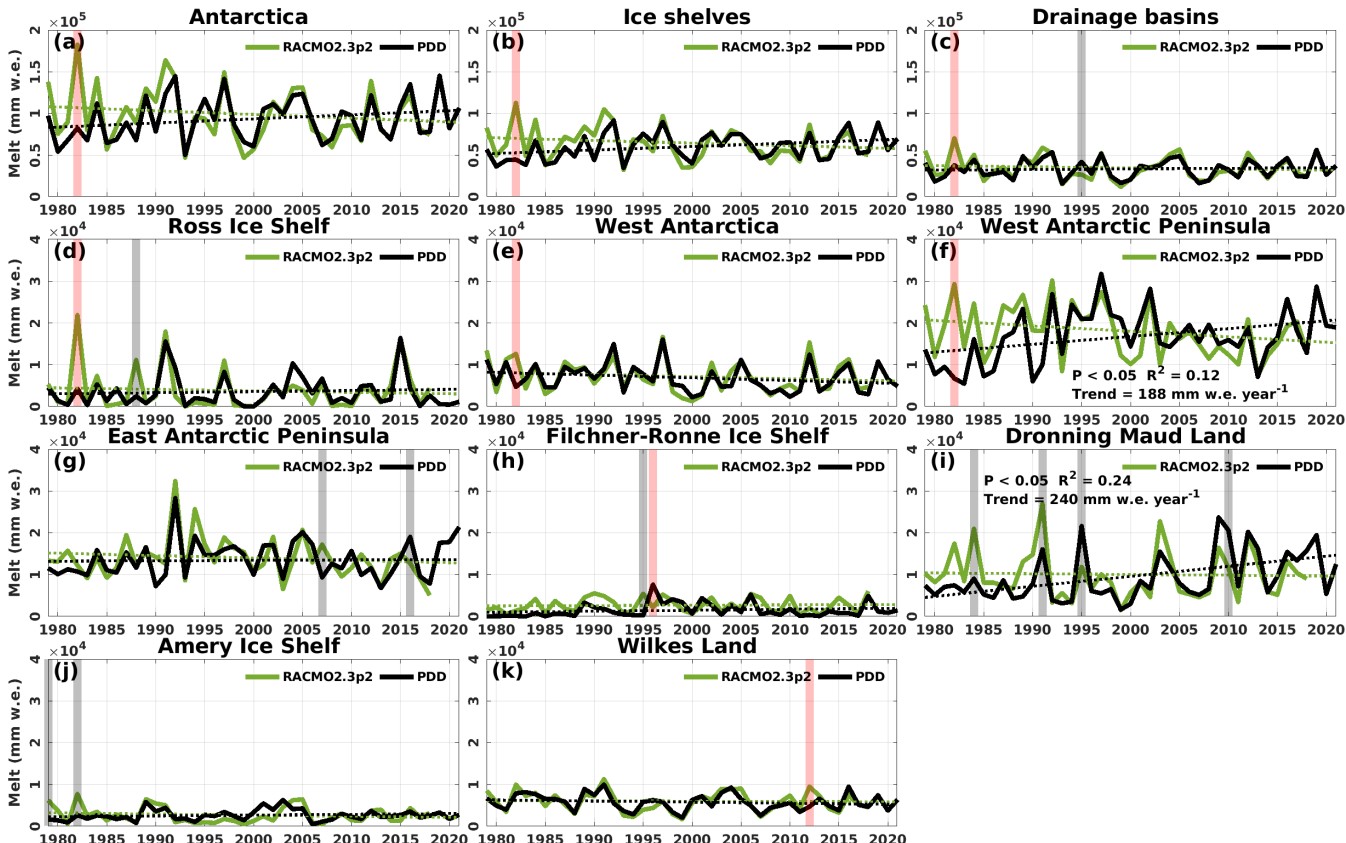

**Figure 9.** Green curves are time series of RACMO2.3p2 annual melt amount (mm w.e.). Black curves are time series of PDD annual melt amount (mm w.e.). RACMO2.3p2 covers the period from 1979/1980 to 2019/2020. PDD covers the period from 1979/1980 to 2021/2022. Dotted lines show the trends that are calculated by fitting ordinary least squares linear regressions during the overlapped period of PDD and RACMO2.3p2. Trends that are statistically significant ($p < 0.05$) are annotated by text with same color in the figure panel (e.g. Figure 9f). Shaded areas mark the years that have residuals larger than three (red) and 1.96 (grey) standard deviations (Figure C4).

Figure C4i). However, there are two outlier years (1995/1996 and 2010/2011) of overestimation of PDD melt detected by the residuals (Figure C3i). These two years are not detected by the residuals between satellite and PDD CMS (Figure C4i).

We find consistency in trend in the remaining six ice shelf regions. Although the sign of the trend for the Ross Ice Shelf,
eastern Antarctic Peninsula (melt amount), Filchner-Ronne Ice Shelf (CMS) and Amery Ice Shelf are opposite, none of them show any statistical significance due to the large inter-annual variability. In addition, the trends with their 95% CI between the satellite/RACMO2.3p2 and PDD for all regions overlap (Table 3). This includes regions where the trends are statistically significant ($p < 0.05$) with opposite signs between the satellite/RACMO2.3p2 and the PDD (western Antarctic Peninsula and Dronning Maud Land). The trend of the PDD CMS for all ice shelves is statistically significant and positive ($p<0.05$) leading
to an approximately 2.13 annual days of melt increase in each computing cell of all Antarctic ice shelves (1594 cells in total)





**Table 3.** Trend of the satellite and PDD CMS ( day km$^2$ year$^{-1}$) for the period from 1979/1980 to 2020/2021 with 1986/1987 to 1988/1989 and 1991/1992 omitted. Trend of the RACMO2.3p2 and PDD melt (mm w.e. year$^{-1}$) for the period from 1979/1980 to 2019/2020. The 95% confidence interval (CI) is calculated by the trend $\pm$ its 1.96 standard error. Bold text mark the trends that are statistically significant (p < 0.05).

| Time series | Satellite CMS | | PDD CMS | | RACMO2.3p2 melt | | PDD melt | |
|---|---|---|---|---|---|---|---|---|
| | Trend ($\times 10^4$) | $R^2$ | Trend ($\times 10^4$) | $R^2$ | Trend ($\times 10^2$) | $R^2$ | Trend ($\times 10^2$) | $R^2$ |
| Antarctica | -9.71 ± 16.36 | 0.04 | 7.23 ± 10.55 | 0.05 | -4.46 ± 8.71 | 0.03 | 4.92 ± 6.74 | 0.05 |
| Ice shelves | -6.46 ± 10.18 | 0.04 | **7.65 ± 6.99** | **0.11** | -3.16 ± 5.22 | 0.03 | 4.15 ± 4.07 | 0.09 |
| Drainage basins | -3.25 ± 6.53 | 0.03 | -0.41 ± 3.83 | 0 | -1.3 ± 3.71 | 0.01 | 0.77 ± 2.96 | 0.01 |
| Ross Ice Shelf | -0.84 ± 2.37 | 0.01 | 0.04 ± 0.84 | 0 | -0.35 ± 1.34 | 0.01 | 0.26 ± 1.01 | 0.01 |
| West Antarctica | -0.96 ± 1.95 | 0.03 | -1.07 ± 1.86 | 0.03 | -0.5 ± 1.01 | 0.02 | -0.64 ± 0.88 | 0.05 |
| West Antarctic Peninsula | -0.73 ± 2.57 | 0.01 | **3.77 ± 2.75** | **0.17** | -1.31 ± 1.67 | 0.06 | **1.88 ± 1.63** | **0.12** |
| East Antarctic Peninsula | -1.97 ± 2.43 | 0.07 | -0.53 ± 1.97 | 0.01 | -0.56 ± 1.31 | 0.02 | 0.1 ± 1.14 | 0 |
| Filchner-Ronne Ice Shelf | -0.22 ± 0.58 | 0.02 | 0.05 ± 0.25 | 0 | 0.09 ± 0.45 | 0 | 0.23 ± 0.45 | 0.03 |
| Dronning Maud Land | -0.98 ± 4.97 | 0 | **5.13 ± 2.91** | **0.25** | -0.18 ± 1.51 | 0 | **2.4 ± 1.33** | **0.24** |
| Amery Ice Shelf | -0.57 ± 1.65 | 0.01 | 0.54 ± 1.13 | 0.02 | -0.27 ± 0.51 | 0.03 | 0.19 ± 0.35 | 0.03 |
| Wilkes Land | -0.19 ± 0.63 | 0.01 | -0.28 ± 0.46 | 0.04 | -0.07 ± 0.65 | 0 | -0.27 ± 0.51 | 0.03 |

in the past four decades. However, it only explains 11% of the variations (Table 3). Overall, the maximum $R^2$ is only 0.25 and for most of them is less than 0.1, exhibiting very low (39 of 44 are less than 10%) explanation of variations (Table 3).

It is worth noting that there are a few years of abnormal PDD over/under- estimation suggested by the residuals of the PDD against the satellite/ RACMO2.3p2 (Figure C3 and Figure C4). Outlier years are detected from residuals distributed outside the three standard deviation range (out of the 99.73% probability on the idealised probability distribution, Appendix C). There are three outlier years detected, which are 1982/1983 for a remarkably strong underestimation over the Ross Ice Shelf, West Antarctica, western Antarctic Peninsula (the CMS is out of the 95% probability, Figure 8f and Figure C3f), whole Antarctica, all ice shelves and all drainage basins, 1996/1997 for an overestimation of melt amount over the Filchner-Ronne Ice Shelf by PDD, and 2012/2013 for an underestimation of melt amount in Wilkes Land (Figure 8 and Figure 9).

Other abnormal PDD estimations are addressed out of the 95% probability (1.96 standard deviation, Appendix C). Although we find there are around 1–4 years of over/under- estimations on melt days and amount for each ice shelf region in the past four decades, these discrepancies reduce when considering the whole AIS. For a larger spatial scale including the whole of Antarctica, all ice shelves and all drainage basins, all residuals disregarding 1982/1983 and 1995/1996 (only for drainage basins melt amount) are distributed within the 95% CI overlapping with zero, with close-to-zero means (Figure C3 and Figure C4). The residuals 95% CI for each of eight ice shelf regions are all overlapping with zero, with close-to-zero means, and their distributions are approximately symmetric along zero (Figure C3 and Figure C4).



The abnormally extensive melt in 1982/1983 has been reported by previous studies (Zwally and Fiegles, 1994; Liu et al., 2006; Johnson et al., 2022). It is suggested to be driven by the SAM, because of an inverse relationship between the number of melt days in Dronning Maud Land and southward migration of the southern Westerly Winds (Johnson et al., 2022). In this extensive melt event, we see relatively high melt presence/ amount captured by both the satellite and RACMO2.3p2 over 8 of 11 regions apart from the Filchner-Ronne Ice Shelf, and ice shelves in East Antarctic Peninsula and Wilkes Land (Figure 8 and Figure 9). However, the PDD model does not capture this extensive melt event in any of the eight extensive melt regions indicated by satellite and RACMO2.3p2 (Figure 8 and Figure 9). The disagreement of the PDD model for this extensive melt event is most likely explained by the absence of any substantial temperature anomaly in the input ERA5 2-m temperature, because of the temperature-dependency of the PDD model (Equation 2) and the temperature-melt relationship (Figure 2). It could also partly be explained by the fact that the PDD parameters were defined based on fitting multi-decadal timeseries between PDD experiments and satellite/ RACMO2.3p2 (Section 3.2), meaning that some inter/inner- annual signals may not be fully captured.

## 4.3 Limitations of the PDD model

The PDD model has the notable advantage of high computational efficiency due to its one-dimensional nature and being solely forced by 2-m air temperature. However, in reality the 2-m air temperature is not the sole driver of Antarctic surface melting (Figure 2). A primary limitation of the PDD model is systematically introduced by the temperature-dependency, making it difficult to accurately estimate surface melt strengthened/ weakened or triggered by other components of the surface energy budget that may accompany climatic phenomena such as the SAM (e.g. Tedesco and Monaghan, 2009; Johnson et al., 2022), El Niño Southern Oscillation (Tedesco and Monaghan, 2009; Scott et al., 2019), föhn winds (e.g. Turton et al., 2020), atmospheric rivers (Wille et al., 2019), sea ice concentrations (Scott et al., 2019), or proximity to dark surfaces such as bare rock (Kingslake et al., 2017). Although we combine observations and model simulations to robustly establish our PDD parameterization and consider the spatial variability of model parameters, the PDD model cannot fully replicate a few extensive melt events presented by satellites and RACMO2.3p2.

Besides, the model simply multiplies a scaling number (DDF) by the summation of temperature above a certain threshold ($T_0$). It lacks the ability to simulate or account for other physical mechanisms such as the meltwater ponding, percolation through the snowpack, refreezing, and so on. As the model is parameterized and calibrated by satellite- and SEB-derived estimates, which also include a variety of assumptions, and due to the scarcity of surface melt data from in situ measurements (Gossart et al., 2019), our PDD output has yet to be confirmed by other datasets.

## 5 Conclusions

We have constructed a PDD model based on the temperature-melt relationship (e.g. Hock, 2005; Trusel et al., 2015), and used it to estimate surface melt in Antarctica in the past four decades. We parameterized the PDD model by running numerical experiments on each individual computing cell to iterate over various combinations of the threshold temperature and the DDF





(Section 3.2). We selected an optimal parameter combination by locating the minimal RMSE between the PDD and satellite
observations, and SEB simulations. We independently performed two-sample KS tests in each experiment in order to quantify
the percentage of cells that have statistically significant ($p < 0.05$) same surface melt distributions for each targeted region. We
have found that rounding the PDD optimal parameters not only simplifies the calculations, without introducing considerable
differences either on the RMSE or two-sample KS percentage, but also avoids suggesting a level of precision defined by the
parameterisation experiments that may not be physically realistic.

Examining the spatial and temporal variability between the PDD estimations, satellite observations and RACMO2.3p2 sim-
ulations, we found that the PDD model has the ability to capture the main spatial and temporal features for a majority of cells
in Antarctica (Section 4.2). As the parameters were parameterized spatially, the PDD is overall in a good agreement with the
spatial patterns shown by the satellite and RACMO2.3p2 data, with the exception of an underestimation on the south Filchner
and south-west Ronne ice shelves, where we found relatively weak temperature-melt correlations (Figure 2). We found that our
optimized PDD parameters were temporally stable – at least for 37 of 41 years in the epoch. The most inadequate estimation
was in 1982/1983, during which we found a significant (residual > three standard deviation) PDD underestimation of surface
melt widely across Antarctica covering Ross Ice Shelf, ice shelves in West Antarctica and western Antarctic Peninsula, Dron-
ning Maud Land and Amery Ice Shelf. We suggest this underestimation corresponded to SAM-influenced climatic conditions,
and that the PDD lacks the ability to accurately capture melt if it arises from effects such as föhn winds or atmospheric rivers
that are not present in the temperature fields used to force the calculations (e.g. Turton et al., 2020; Wille et al., 2019). Other
over/under- estimations detected by 1.96 standard deviations of residuals are found over ice shelves in Dronning Maud Land
for at most four years (1984/1985, 1991/1992, 1995/1996 and 2010/2011, Figure 9i and Figure C4i). We suggest this is due to
the limitations of PDD model in capturing inter/inner- annual signals as a result from 40-yearly defined parameters.

These limitations aside, we found that the PDD model can not only relatively accurately estimate surface melt in Antarctica
compared with the satellite observations and more sophisticated SEB model, but it is also highly computationally efficient.
These advantages may allow us to use the PDD model to explore Antarctic surface melt in a longer-term context into the future
and over periods of the geological past when neither satellite observations nor SEB components are available. However, due to
the systematical limitations of the PDD model and the scarcity of Antarctic surface melt data available (Gossart et al., 2019),
more work is needed such as model evaluation, exploration of the temporal variability of PDD parameters, and discussions of
approximations to the physical processes (e.g. refreezing) taking place after surface melting. Nevertheless, PDD models have
been used in many numerical ice sheet models for the empirical approximation of surface mass balance computations, due to
their unique advantages in terms of their simple temperature-dependency and computational efficiency. We propose that our
spatially-parameterized implementation extends the utility of the PDD approach and, when parameterized appropriately, can
provide a valuable tool for exploring surface melt in Antarctica in the past, present and future.



*Data availability.* The ERA5 reanalysis data are available from https://www.ecmwf.int/en/forecasts/dataset/ecmwf-reanalysis-v5 (last access: 02 August 2022). The Zwally Antarctic drainage basin (Zwally et al., 2012) data are available from http://imbie.org/imbie-3/drainage-basins/. The satellite SMMR and SSM/I products (Picard and Fily, 2006) are available from https://snow.univ-grenoble-alpes.fr/melting/ (last access: 02 August 2022). The satellite AMSR-E product is available from Picard et al. (2007). The satellite AMSR-2 product is available by contacting ghislain.picard@univ-grenoble-alpes.fr. The RACMO2.3p2 data are available from https://doi.org/10.5194/tc-12-1479-2018 (Van Wessem et al., 2018). The monthly PDD model data (this study) is available in this study. The daily PDD model data (this study) is available by contacting yaowen.zheng@vuw.ac.nz.

## Appendix A: Satellite data

The number of melt days and the area of surface melt can be detected using the microwave brightness temperature data since 1979 (e.g. Torinesi et al., 2003; Picard and Fily, 2006). The theoretical basis of this approach is that changes between dry and wet snow can be distinguished by the upwelling microwave brightness temperature change (Chang and Gloersen, 1975). When dry snow is melting, the meltwater at the surface significantly changes the dielectric properties of the surface by increasing absorption and increasing microwave emission (Chang and Gloersen, 1975; Zwally and Fiegles, 1994). By applying an empirical threshold with an appropriate surface melt detecting algorithm (Torinesi et al., 2003), the number of melt days and the spatial extent of surface melt can be detected (e.g. Torinesi et al., 2003; Picard and Fily, 2006). This satellite observational approach has been developed and used for Antarctic surface melt investigations (e.g. Picard and Fily, 2006; Johnson et al., 2022), showing it as a valuable and powerful tool that can be used to study and understand the surface melt frequency in Antarctica on both continental and regional scales (Johnson et al., 2022). However, this approach does not allow melt volume to be retrieved.



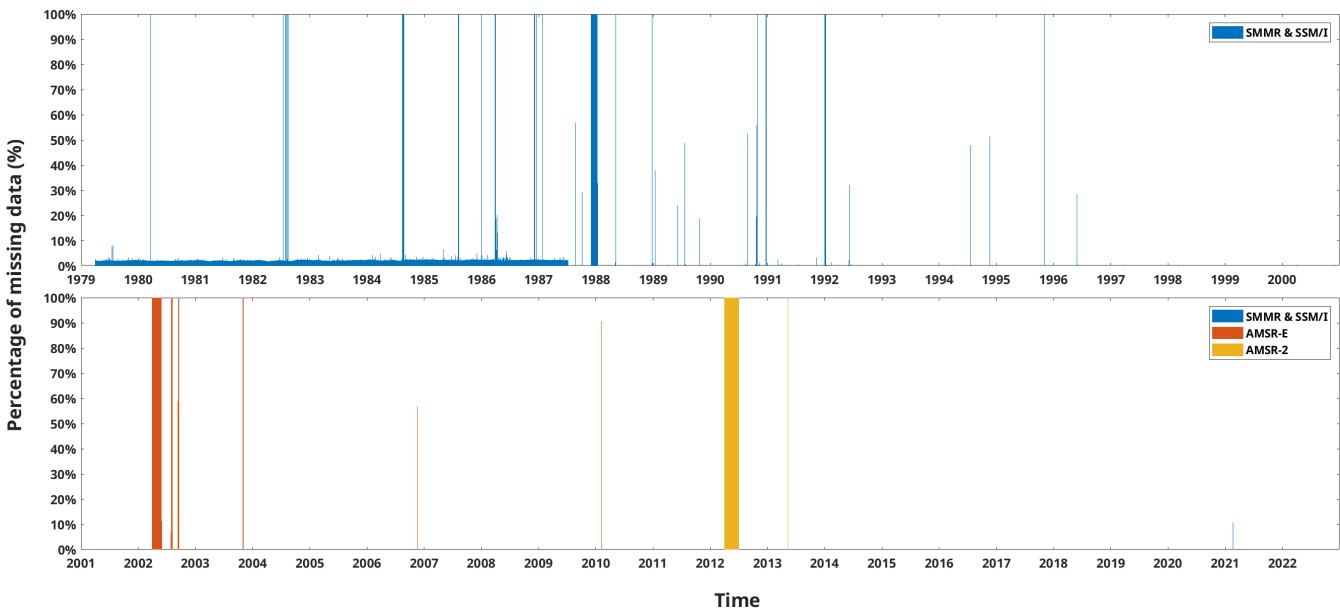

**Figure A1.** Daily percentage of missing data for satellite observations. Satellite SMMR and SSM/I covers the period from 1979-04-01 to 2021-03-31. Satellite AMSR-E covers the period from 2002-04-01 to 2011-03-31. Satellite AMSR-2 covers the period from 2012-04-01 to 2021-12-31.

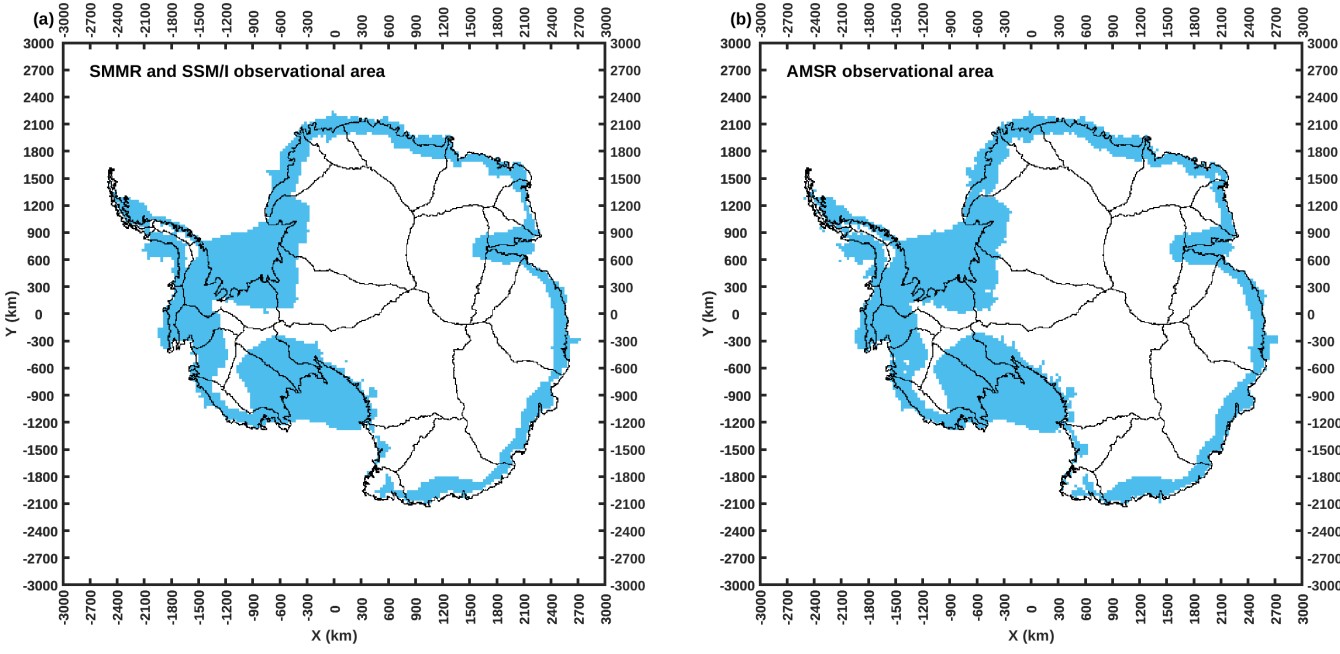

**Figure A2.** (a) mask of the satellite SMMR and SSM/I observational area. (b) mask of the satellite AMSR (AMSR-E and AMSR-2) observational area. Both masks are bilinearly remapped to the $30 \times 30 \text{ km}^2$ polar stereographic grid.





## Appendix B: Significance testing and model simplification

We calculate the RMSE of the annual number of melt days between the PDD (with the $T_0$ and the rounded optimal $T_0$, respectively) and the satellite observations on each computing cell. We then calculate the average of the RMSE for the computing cells in each of our targeted regions, respectively. For the two-sample KS test percentages, hereafter KS-test (%), we firstly perform the two-sample KS tests on the time series of annual number of melt days between the PDD model and the satellite. Then we calculate the percentage of the computing cells that are tested to be statistically significantly similarly distributed (p

< 0.05, two-sample KS test). KS-test (%) for the optimal $T_0$ and the rounded optimal $T_0$ as listed in Table B1 is therefore referred to the percentage of cells that have a statistically significant similar distribution between the PDD model and satellite in the corresponding region. The RMSE difference as listed in Table B1 is calculated by the difference on the RMSE between the optimal $T_0$ and the rounded optimal $T_0$. For example, the RMSE on the annual number of melt days between the PDD model with optimal $T_0$ and the satellite for the targeted region "West Antarctic Peninsula" is 23.9, and for the annual number

of melt days between the PDD model with rounded optimal $T_0$ and the satellite is 25.08. The RMSE difference for the targeted region "West Antarctic Peninsula" is therefore the percentage of the increase on the value of RMSE from the optimal $T_0$ to the rounded optimal $T_0$ (($25.08 / 23.9 - 1) \times 100\% \approx 4.94\%$). For the DDF as listed in Table B2, the calculations are the same but for the different objects. For the $T_0$ (Table B1), the calculation objects are between the PDD model with optimal $T_0$/ rounded optimal $T_0$ and the satellite. For the DDF (Table B2), the calculation objects are between the PDD model with optimal DDF/

rounded optimal DDF and the RACMO2.3p2.

Table B1: Table of $T_0$ ( $C°$), RMSE and KS-test (%) for optimal $T_0$ and rounded optimal $T_0$ on each targeted region. RMSE difference (%) is calculated by the percentage difference between the rounded optimal $T_0$ RMSE and the optimal $T_0$ RMSE.

| | Optimal $T_0$ | | | Rounded optimal $T_0$ | | | |
|---|---|---|---|---|---|---|---|
| Targeted region | $T_0$ | RMSE | KS-test (%) | $T_0$ | RMSE | KS-test (%) | RMSE difference (%) |
| Antarctica | -1.8 | 4.14 | 81 | -2 | 4.17 | 81 | 0.72 |
| Ice shelves | -1.8 | 7.7 | 72 | -2 | 7.76 | 70 | 0.78 |
| Drainage basins | -1.8 | 2.52 | 86 | -2 | 2.53 | 86 | 0.4 |
| Ross Ice Shelf | -1 | 2 | 97 | -1 | 2 | 97 | 0 |
| West Antarctica | -1.6 | 9.8 | 71 | -2 | 10.74 | 69 | 9.59 |
| West Antarctic Peninsula | -2.4 | 23.9 | 54 | -2 | 25.08 | 45 | 4.94 |
| East Antarctic Peninsula | -2.1 | 18.79 | 77 | -2 | 18.81 | 74 | 0.11 |
| Filchner-Ronne Ice Shelf | -0.4 | 1.04 | 98 | 0 | 1.08 | 96 | 3.85 |
| Dronning Maud Land | -1.7 | 14.24 | 60 | -2 | 14.4 | 61 | 1.12 |
| Amery Ice Shelf | -4 | 15.58 | 60 | -4 | 15.58 | 60 | 0 |
| Wilkes Land | -3.3 | 17.23 | 43 | -3 | 17.34 | 38 | 0.64 |





| | | | | | | | |
|---|---|---|---|---|---|---|---|
| Basin 1 | -1.2 | 0.57 | 96 | -1 | 0.58 | 96 | 1.75 |
| Basin 2 | -0.4 | 0.15 | 100 | 0 | 0.15 | 100 | 0 |
| Basin 3 | -1.1 | 0.33 | 97 | -1 | 0.33 | 96 | 0 |
| Basin 4 | -1.3 | 1.87 | 90 | -1 | 1.89 | 88 | 1.07 |
| Basin 5 | -1.6 | 3.49 | 82 | -2 | 3.69 | 83 | 5.73 |
| Basin 6 | -1.2 | 3.11 | 81 | -1 | 3.12 | 80 | 0.32 |
| Basin 7 | -1.3 | 2.86 | 76 | -1 | 2.87 | 74 | 0.35 |
| Basin 8 | -1.1 | 1.81 | 80 | -1 | 1.81 | 80 | 0 |
| Basin 9 | -3.3 | 4.37 | 77 | -3 | 4.43 | 76 | 1.37 |
| Basin 10 | -3.6 | 0.54 | 97 | -4 | 0.56 | 97 | 3.7 |
| Basin 11 | -2.4 | 2.94 | 87 | -2 | 2.96 | 85 | 0.68 |
| Basin 12 | -3.1 | 4.25 | 80 | -3 | 4.26 | 80 | 0.24 |
| Basin 13 | -2.2 | 3.04 | 82 | -2 | 3.05 | 83 | 0.33 |
| Basin 14 | -2.1 | 1.8 | 88 | -2 | 1.8 | 87 | 0 |
| Basin 15 | -3.8 | 7.1 | 51 | -4 | 7.15 | 51 | 0.7 |
| Basin 16 | -2.8 | 1.31 | 87 | -3 | 1.31 | 88 | 0 |
| Basin 17 | -0.3 | 1.27 | 95 | 0 | 1.31 | 94 | 3.15 |
| Basin 18 | -1.3 | 1.13 | 100 | -1 | 1.14 | 100 | 0.88 |
| Basin 19 | -2.2 | 1.73 | 99 | -2 | 1.74 | 99 | 0.58 |
| Basin 20 | -2.2 | 3.26 | 77 | -2 | 3.28 | 75 | 0.61 |
| Basin 21 | -1.8 | 1.7 | 87 | -2 | 1.72 | 88 | 1.18 |
| Basin 22 | -2.5 | 1.75 | 91 | -3 | 1.81 | 91 | 3.43 |
| Basin 23 | -0.8 | 2.75 | 84 | -1 | 2.78 | 85 | 1.09 |
| Basin 24 | -1.7 | 6.69 | 66 | -2 | 6.87 | 64 | 2.69 |
| Basin 25 | -0.5 | 10.73 | 51 | -1 | 10.85 | 52 | 1.12 |
| Basin 26 | -2.8 | 15.19 | 59 | -3 | 15.46 | 63 | 1.78 |
| Basin 27 | -3 | 5.47 | 74 | -3 | 5.47 | 74 | 0 |

Table B2: Table of DDF (mm w.e. $C^{\circ -1} \; day^{-1}$), RMSE and KS test (%) for optimal DDF and rounded optimal DDF on each targeted region. RMSE difference (%) is calculated by the percentage difference between the rounded optimal DDF RMSE and the optimal $T_0$ RMSE.

| | Optimal DDF | | Rounded optimal DDF | |
|---|---|---|---|---|

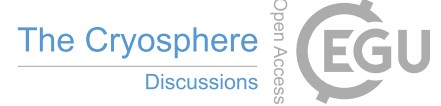

| Targeted region | DDF | RMSE | KS-test (%) | DDF | RMSE | KS-test (%) | RMSE difference (%) |
|---|---|---|---|---|---|---|---|
| Antarctica | 2.8 | 1.76 | 89 | 3 | 1.86 | 89 | 5.68 |
| Ice shelves | 2.5 | 10.44 | 66 | 3 | 13.56 | 64 | 29.89 |
| Drainage basins | 3.5 | 0.67 | 93 | 4 | 0.77 | 93 | 14.93 |
| Ross Ice Shelf | 7.9 | 7.1 | 83 | 8 | 7.1 | 82 | 0 |
| West Antarctica | 3.1 | 18.24 | 69 | 3 | 18.31 | 68 | 0.38 |
| West Antarctic Peninsula | 2 | 65.03 | 52 | 2 | 65.03 | 52 | 0 |
| East Antarctic Peninsula | 2.1 | 49.32 | 55 | 2 | 49.75 | 52 | 0.87 |
| Filchner-Ronne Ice Shelf | 17.4 | 4.03 | 18 | 17 | 4.03 | 18 | 0 |
| Dronning Maud Land | 2.9 | 24.46 | 63 | 3 | 24.58 | 66 | 0.49 |
| Amery Ice Shelf | 1 | 30.93 | 37 | 1 | 30.93 | 37 | 0 |
| Wilkes Land | 2.1 | 28.41 | 34 | 2 | 28.72 | 34 | 1.09 |
| Basin 1 | 12.2 | 0.81 | 88 | 12 | 0.81 | 88 | 0 |
| Basin 2 | 9.6 | 0.02 | 99 | 10 | 0.02 | 99 | 0 |
| Basin 3 | 4.9 | 0.06 | 98 | 5 | 0.06 | 98 | 0 |
| Basin 4 | 10.2 | 1.99 | 75 | 10 | 1.99 | 75 | 0 |
| Basin 5 | 2.1 | 3.07 | 82 | 2 | 3.07 | 82 | 0 |
| Basin 6 | 7.5 | 2.75 | 85 | 8 | 2.76 | 85 | 0.36 |
| Basin 7 | 20.3 | 1.39 | 86 | 20 | 1.4 | 86 | 0.72 |
| Basin 8 | 9.8 | 2.25 | 78 | 10 | 2.25 | 78 | 0 |
| Basin 9 | 1.9 | 5.04 | 77 | 2 | 5.04 | 77 | 0 |
| Basin 10 | 2.9 | 0.11 | 100 | 3 | 0.11 | 100 | 0 |
| Basin 11 | 4.6 | 2.38 | 88 | 5 | 2.4 | 88 | 0.84 |
| Basin 12 | 2.7 | 2.06 | 94 | 3 | 2.19 | 95 | 6.31 |
| Basin 13 | 3.8 | 1.07 | 95 | 4 | 1.08 | 95 | 0.93 |
| Basin 14 | 3.9 | 0.74 | 94 | 4 | 0.74 | 94 | 0 |
| Basin 15 | 1.8 | 4.6 | 68 | 2 | 4.68 | 71 | 1.74 |
| Basin 16 | 2.9 | 0.16 | 93 | 3 | 0.16 | 94 | 0 |
| Basin 17 | 5.3 | 0.14 | 98 | 5 | 0.14 | 98 | 0 |
| Basin 18 | 4 | 1.98 | 95 | 4 | 1.98 | 95 | 0 |
| Basin 19 | 2.6 | 2.04 | 99 | 3 | 2.1 | 97 | 2.94 |
| Basin 20 | 4.2 | 4.43 | 77 | 4 | 4.44 | 77 | 0.23 |
| Basin 21 | 4 | 1.32 | 95 | 4 | 1.32 | 95 | 0 |
| Basin 22 | 1.3 | 0.95 | 76 | 1 | 1.14 | 75 | 20 |





| | | | | | | | |
|---|---|---|---|---|---|---|---|
| Basin 23 | 8 | 6.27 | 63 | 8 | 6.27 | 63 | 0 |
| Basin 24 | 2 | 12.3 | 68 | 2 | 12.3 | 68 | 0 |
| Basin 25 | 5.5 | 19.08 | 32 | 6 | 19.88 | 32 | 4.19 |
| Basin 26 | 1.5 | 27.07 | 48 | 2 | 46.49 | 39 | 71.74 |
| Basin 27 | 1.8 | 7.16 | 66 | 2 | 7.41 | 68 | 3.49 |

Figure B1 shows the results of the KS-test (%) on $T_0$ experiments. The optimal $T_0$ for each ice shelf region varies from -4.0 to 0.0 °C (Figure 3a-d to a-k). The difference of the percentage of same distribution cells between the $T_0$ and the test maximum points does not exceed 9% (Figure B1a-d to a-k). The KS-test (%) results between the $T_0$ and the test maximum are generally in a good agreement over the 27 Antarctic drainage basins (Figure 3a-l to b-l). Apart from Basin 15 (14%, Figure 3a-z), the

other remaining basins show the difference of percentage of same distribution cells between the $T_0$ and the maximum do not exceed 7%. Taken together, we see no obvious evidence to reject these optimal $T_0$.

Figure B2 shows the results of the KS-test (%) on DDF experiments. For the ice shelves, the optimal DDF for the Ross Ice Shelf (Figure 4a-d), West Antarctica (Figure 4a-e) and Filchner-Ronne Ice Shelf (Figure 4a-h) are consistent with the two-sample KS tests. Because the percentage of cells that have the statistically significant ($p < 0.05$) same surface melt distribution

for the optimal DDF and two-sample KS test maximum are approximately equal ($\leq 5\%$ difference) (Figure B2a-d, a-e and a-h). The largest disagreement is on the Wilkes Land with a 18% drop from the two-sample KS test maximum (Figure B2a-k). The remaining four regions have 7–11% difference on the percentage of statistically significant ($p < 0.05$) same surface melt distribution cells against the two-sample KS test maximum (Figure B2a-f, a-g, a-i and a-j).



**Figure B1.** Results of the KS-test (%) on each $T_0$ experiment for each targeted region. Blue envelope covers the range of satellite SMMR and SSM/I, AMSR-E and AMSR-2. Purple vertical line marks the optimal $T_0$. Black vertical line marks the rounded optimal $T_0$. Red vertical line marks the KS-test (%) maximum. Annotated texts in each figure panel indicate the value of the KS-test (%) at the regarding colored $T_0$ (e.g. purple colored texts mark the KS-test (%) values for the rounded optimal $T_0$).

**Figure B2.** Results of the KS-test (%) on each DDF experiment for each targeted region. Blue envelope covers the range of satellite SMMR and SSM/I, AMSR-E and AMSR-2. Purple vertical line marks the optimal DDF. Black vertical line marks the rounded optimal DDF. Red vertical line marks the KS-test (%) maximum. Annotated texts in each figure panel indicate the value of the KS-test (%) at the regarding colored DDF (e.g. purple colored texts mark the KS-test (%) values for the rounded optimal DDF).





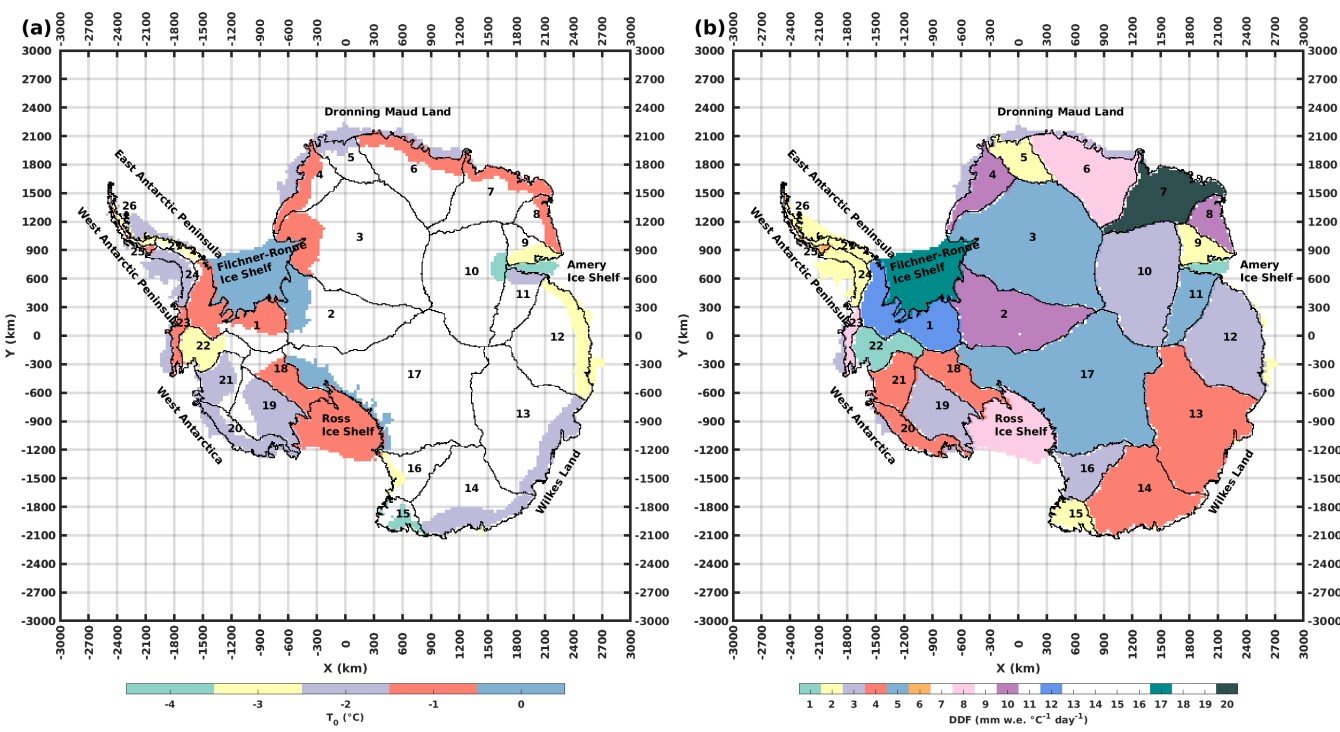

**Figure B3.** (a) Map for the spatial distribution of the PDD parameter $T_0$ (rounded optimal $T_0$). (b) Map for the spatial distribution of the PDD parameter DDF (rounded optimal DDF).





## Appendix C: Spatial and temporal variability

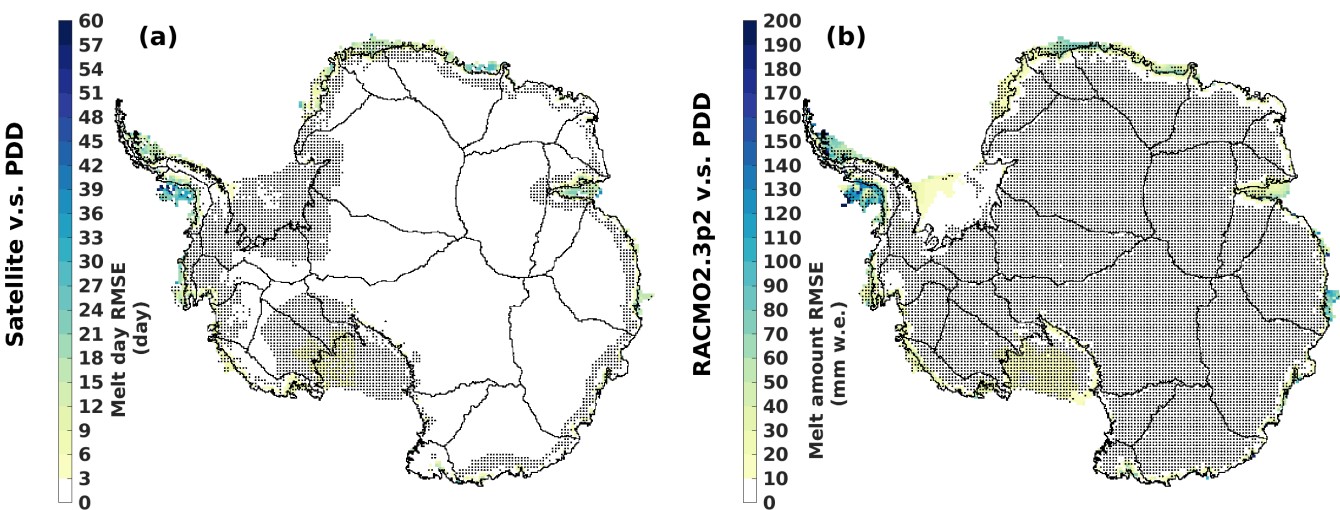

**Figure C1.** (a) RMSE between satellite and PDD annual melt days on each individual computing cell in the period from 1979/1980 to 2019/2020. Note that 1986/1987 to 1988/1989 and 1991/1992 are omitted. Period from 2002/2003 to 2010/2011 for satellite is the average of SMMR and SSM/I, and AMSR-E. Period from 2012/2013 to 2020/2021 is the average of SMMR and SSM/I, and AMSR-2. (b) RMSE between RACMO2.3p2 and PDD annual melt amount on each individual computing cell in the period from 1979/1980 to 2019/2020. Black dots mark the statistically significant (p<0.05) same distribution cells tested by two-sample KS tests.

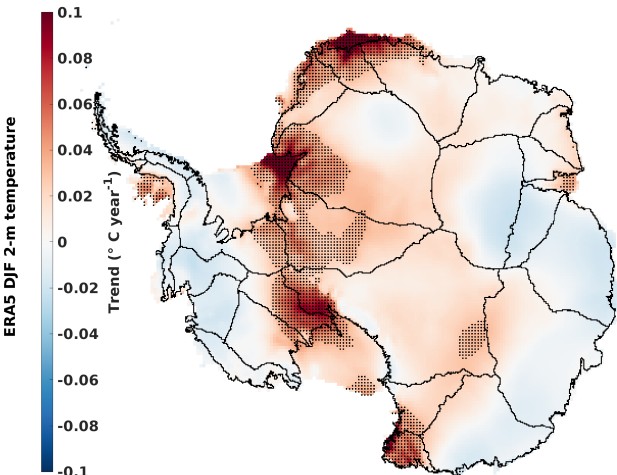

**Figure C2.** Trend of the mean DJF ERA5 2-m temperature on each computing cell during the period 1979/1980–2019/2020. Black dots mark the trends that are statistically significant (p < 0.05).





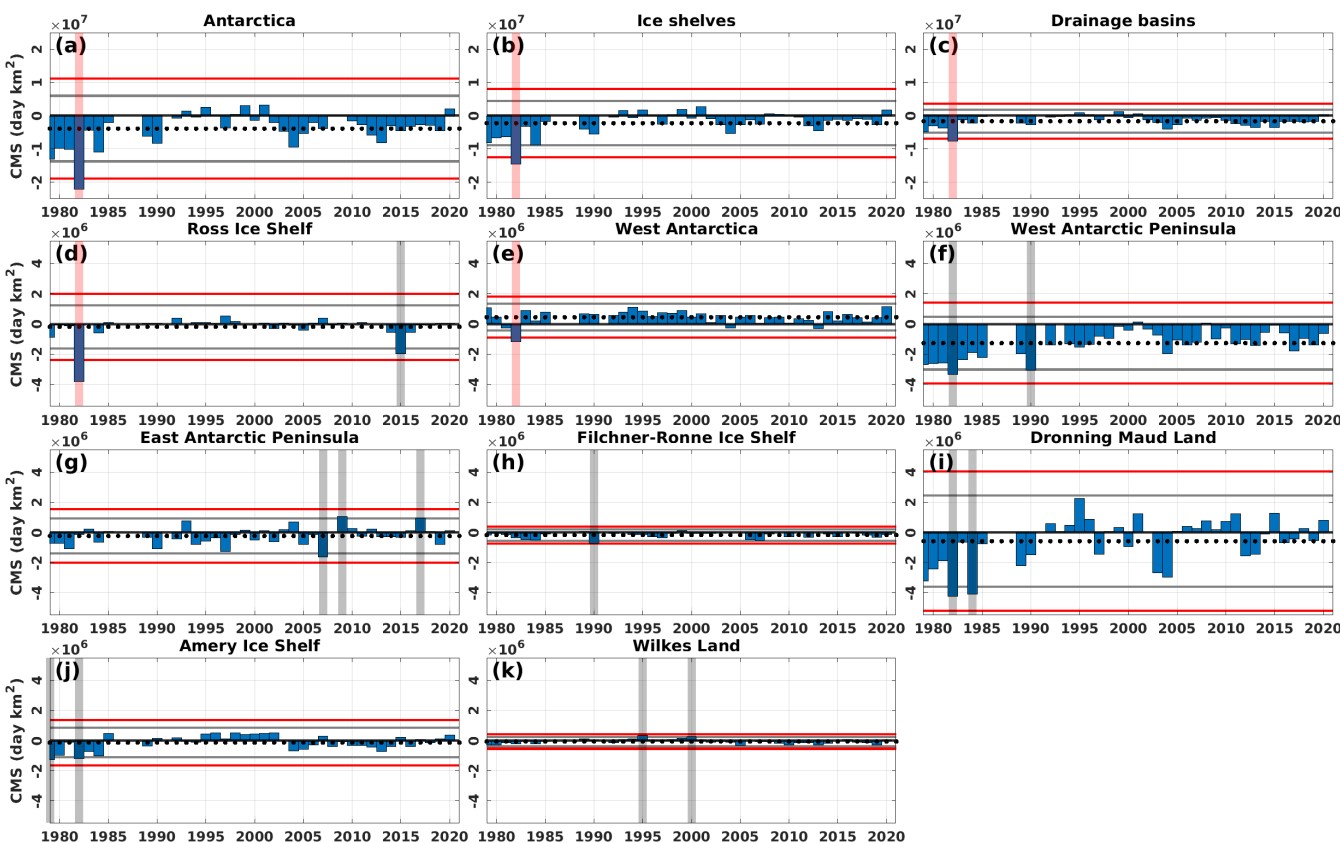

**Figure C3.** Residuals between the PDD model estimation and satellite observations for annual CMS (day $km^2$) in the period from 1979/1980 to 2019/2020. Note that 1986/1987 to 1988/1989 and 1991/1992 are omitted. Period from 2002/2003 to 2010/2011 for satellite is the average of SMMR and SSM/I, and AMSR-E. Period from 2012/2013 to 2020/2021 is the average of SMMR and SSM/I, and AMSR-2. Black horizontal dotted line marks the residuals mean. Grey horizontal line marks the mean +/- 1.96 standard deviation. Red horizontal line marks the mean +/- 3 standard deviation. Grey vertical line marks the year where the residual is larger than 1.96 standard deviation. Red vertical line marks the year where the residual is larger than three standard deviation.





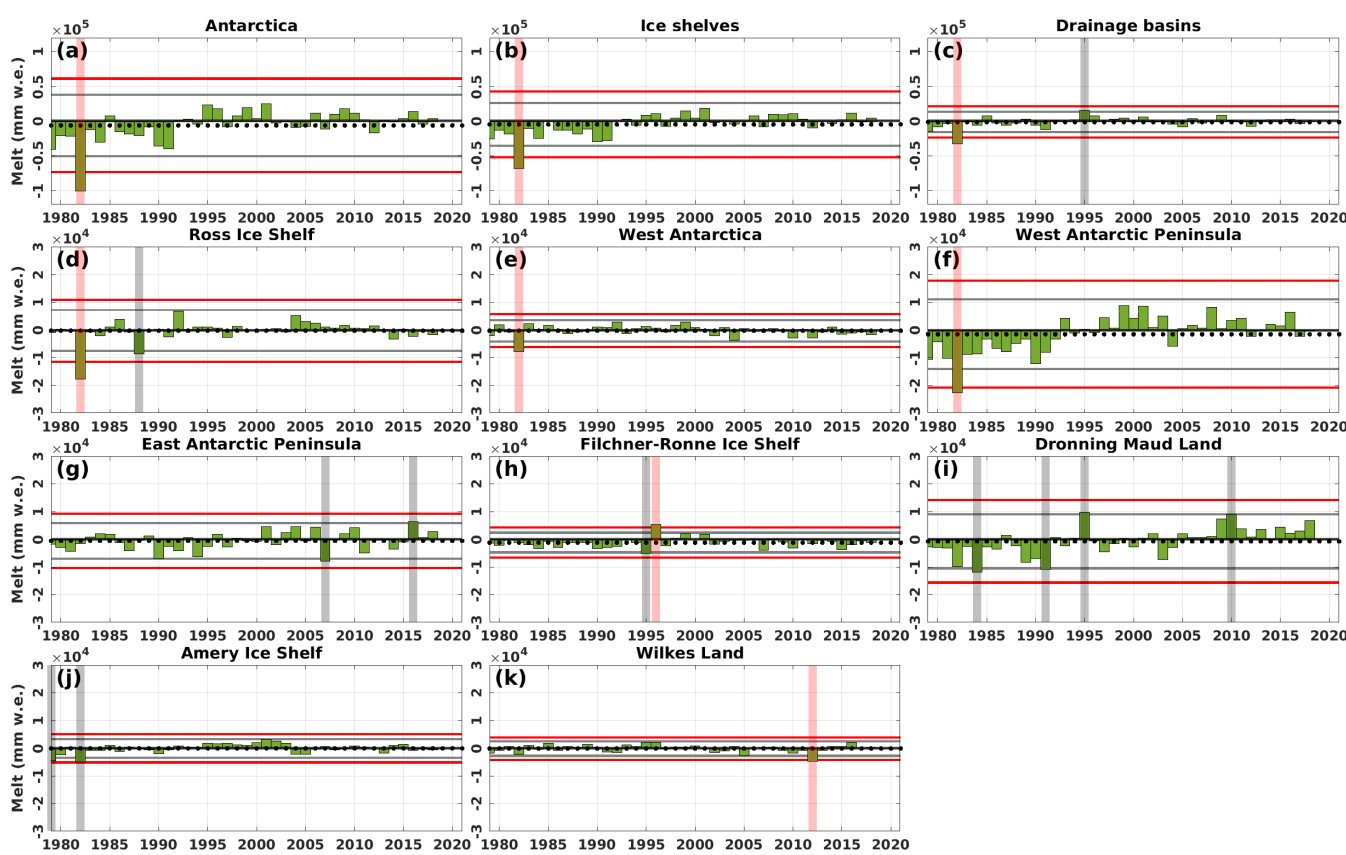

**Figure C4.** Residuals between the PDD model estimation and RACMO2.3p2 for annual melt amount (mm w.e.) in the period from 1979/1980 to 2019/2020. Black horizontal dotted line marks the residuals mean. Grey horizontal line marks the mean +/- 1.96 standard deviation. Red horizontal line marks the mean +/- 3 standard deviation. Grey vertical line marks the year where the residual is larger than 1.96 standard deviation. Red vertical line marks the year where the residual is larger than three standard deviation.



*Author contributions.* YZ, NRG and AG conceived the study. YZ performed the analysis and prepared the original draft of the paper. GP and MLL provided satellite products. All authors contributed to writing the paper.

*Competing interests.* The authors declare that they have no conflict of interest.

*Acknowledgements.* YZ and NRG are supported by the Royal Society of New Zealand, award RDF-VUW1501. NRG and AG are supported by Ministry for Business Innovation and Employment, Grant/Award Number ANTA1801 ("Antarctic Science Platform"). NRG acknowledges
support from Ministry for Business Innovation and Employment, Grant/Award Number RTUV1705 ("NZSeaRise").





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
