# Peer review of "Statistically parameterizing and evaluating a positive degree-day model to estimate surface melt in Antarctica from 1979 to 2022"

_The Cryosphere, 2022_

## Referee Comment (RC1)

***Review of Estimating surface melt in Antarctica from 1979 to 2022, using a statistically parameterized positive degree-day model by Zheng et al., 2022***

Zengh et al., 2022 estimate the melt over the Antarctic Ice Sheet using a PDD model. They also carefully parametrize their model to produce similar results than satellite and RACMO estimations. The calibration of the model is particularly complete and nothing seems to be omitted for a reproducibility of the results. Obtaining a correct PDD model is interesting and important because, as the authors mention, the efficiency of the PDD in terms of computation time allows long simulations where other more complex models do not enable it. However, most of the manuscript focuses on the calibration and the evaluation of the PDD, and only a limited part is dedicated to the new information about melt. From this point of view, the scientific issue of the manuscript could be more emphasized in the title and might have deserved a submission to another journal like GMD (this does not, however, question the quality and robustness of the study. I would add that while being comprehensive is a quality, some passages are difficult to read because too much information is given when that information is sometimes obvious. I also have some more comments listed hereafter. In general, I don't have much to say except to advise to simplify/summarize some passages (introduction, result) and to add some nuances (see major comments).

**Major comments:**

One major limitation of the PDD is that all the calibration is done over present climate where melt is only limited to margins and weak. One could question the validity of the calibration in warmer climates, i.e. for projections. For instance, how could the melt computed by PDD take into account the snow albedo feedback (which is a process often not represented by PDD due to their simplicity)? Or could the PDD correctly represent areas where no melt currently occurs and will likely occur in warmer climates as it was calibrated to reproduce current surface melt? Maybe you could compare the results of a PDD calibrated over only low melt years but evaluated over high melt years (even high present-day melt years will be considered as low melt years in the future).

Furthermore, the calibration of the quantity of melt is solely based on the RCM RACMO, what is the impact of RACMO biases on the PDD? Although Mottram et al, 2021 showed that RACMO is one of the best models to represent the Antarctic climate, they also suggested that RACMO underestimates near-surface air temperature which could also influence melt computed by RACMO. Note that the calibration and evaluation is also not independent as you use the same values.

**Minor comments:**

P1L4: Consider to nuance since this will only be the case if the firn cannot absorb the additional water

P1L13: Satellite observation, I suggest to replace observation by estimates as melt is not directly observed by satellite but derived from brightness temperatures or absorptivity of the

surface under the assumption that the presence of water at the surface is newly-produced melt. (also for P3L81)

P1 L26-27: Is this 100% valid? Surface melt is projected to remain limited to ice shelves (Kittel et al., 2021) beyond 2100 where basal melt should have a much higher influence (Seroussi et al., 2020). Consider nuance.

P1 L29-L46: I would shorten these two paragraphs which, although interesting in historical terms for the evolution of ice shelves, do not bring much information directly related to your topic. This is only a suggestion, feel free to keep as it is.

P4L105: observation => values/estimates

P8L179-181: Is the ERA5 mean also computed between 6am and 6pm in local time for each grid point?

P8 L186-192: These sentences can be considered as an example of too much provided details that can make the manuscript hard to read. I'd say that the only necessary information is that the RMSE from each pixel is averaged to produce a RMSE per region. All the other information seems obvious and may be non necessary.
Also Figure 7 for instance, not necessary how the surface elevation is obtained.

P10 L229: Could this melt associated with relative apparent cold conditions be related to katabatic winds in that area, maybe not correctly represented by ERA5? (ie, could ERA5 actually underestimate temperature in that region leading to suspicious values).

P14 L283: Instead of surface temperature, use air temperature as it is the input variable of the PDD.

P16 L320-321: How do you obtain the quantity of melt from the satellite? Should it be melt area * melt days (instead of melt quantity?). Could you also compute the CMS from RACMO to add a comparison?

---

## Referee Comment (RC2)

**Summary**

This work presents a new method to obtain Antarctic surface melt using only near-surface air temperature and is parameterized for different regions of Antarctica. Overall, the work presented is of high quality and will improve our understanding of AIS surface melt. However, I think the presentation of the methods, results and contribution of this work needs to be modified throughout the paper to clarify how this work fits into previous studies and contributes to ongoing efforts to better understand AIS surface melt. Many of the paragraphs are a bit confusing and difficult to read because unnecessary information is present. I feel confident that once the writing is clarified and condensed a bit that this work will be a great contribution! Nice job on the figures as they are all very clear.

**Major revisions**

I think the main thing that is missing for me is a quantified justification for calculating parameters for your PDD in each region or basin. What do you gain from the by-region parameterization of your PDD model? How much better does your model do in each basin/ice-shelf region compared to observations than if you just chose one value for the whole AIS?

Many of the sentences throughout the paper use "this" or "it" without specifying what "this" or "it" is referring to. For example, in Line 164: "It shows that most of the cells in Antarctic…"

You mention that topography can introduce spatial variability in PDD parameterization. In this work, you parameterize a PDD model for each basin, but what about topographic variability within each basin? I am left wondering how variable the optimal PDD model parameters are withing each basin/region?

I think more emphasis needs to be put on why your PDD model is useful throughout the paper (esp. in the intro and discussions sections). This work provides a new method but not really any new information about AIS melt that cannot be obtained from RACMO or satellite observations. To that end, I think the final sentence in the introduction is a bit misleading because really you just compare output from your PDD model with observations and regional climate model output of surface melt. I would suggest focusing on the novelty of the method and potential applications for it, instead of the fact that you use your model to estimate AIS surface melt from 1979-2022 (because this can be and has been done already with satellite observations and RCMs instead).

**Minor revisions**

I find the abstract a bit vague. There is too much emphasis on introductory material (lines 1-12) while only one sentence touches on the results (lines 16,17) and this sentence does not make sense within the rest of the abstract (e.g. what epoch?). I think this sentence does not do a good job of summarizing your results in the abstract and I am left wondering what to take away from the paper.

Lines 24-28: You talk about future projects of Antarctic surface melt but what about current Antarctic melt? I think it will help to add a sentence or two that covers the current state of knowledge of AIS melt (i.e. Stokes 2021, Arthur 2022, Corr 2022).

Line 28 – Explain how AIS melt impacts ice sheet mass balance.

Lines 29 – 47: Much of these paragraphs is unnecessary. I think that these paragraphs draw focus away from the main topic of this paper (which is not hydrofracture and ice-shelf collapse as it may seem by reading these two paragraphs). I think this can be summarized in a few sentences related to why surface melt is important in Antarctica.

Line 48: "surface melt has most likely been accelerated by the rapid increase of atmospheric temperatures…". I do not believe that this statement is correct as I don't think that AIS surface melt has accelerated. Alison Banwell recently reported results at the Cryosphere 2022 conference in Reykjavik showing a statistically significant *decrease* in surface melt across the AIS (will be published soon). Also, the observational trends reported in Figures 5 and 6 disagree with this statement.

One thing that I think is missing from the introduction – Why are PDD models helpful? What do they add? I think perhaps some of the info from the abstract on PDD models would be better in the introduction.

Line 59 – "The PDD model calculates…" All PDD models or just one in particular?

Line 60 – "… based on the temperature-melt relationship". Earlier (line 57) you mention that PDD models also use the precipitation field. Is this just some PDD models? Or are you talking about the ice sheet models that use precipitation fields to determine SMB?

Lines 65-67 – "Wake and Marshall (2015)…" I find this sentence to be a bit distracting and confusing. I think you could just simplify to: "Wake and Marshall (2015) suggest that Antarctic surface melt can be estimated solely from monthly temperature".

Line 69 – What do you mean by "universal usage"?

Line 71 – "Topographic influences" such as what?

Line 79 – "… ice shelf region…". Refer to Figure 1.

Lines 81 – Specify that you take *melt volume* from RAMCO.

Line 92 – Specify that you use hourly 2-m air temperature ERA5 data in the text (also on line 174).

Line 95 – ERA5 performs better *at what* than those other models?

Line 109 – "around 6 am and 6 pm" – How close to 6 am and 6 pm? Does the acquisition time vary each day?

Line 110 – "This dataset is being continually updated…" Consider moving this sentence to your Data Availability statement.

Line 115 – "We therefore omit those periods from our analysis" – Which analysis in particular? These periods are not omitted from the trend analysis in section 4.2. I would probably suggest to omit these no-data periods from the trend analysis as well (for both satellite and PDD model trends).

Line 116 – For readability, change "More recently, there is a newly developed…" to "We also use a more recently developed…".

Lines 119-120 – It is not necessary to specify that this product has a "twice finer spatial resolution than satellite SMMR and SSM/I product" since you mention the resolution of both products.

Line 123 – Consider changing the section heading to "**Regional climate model melt output".** Additionally, I believe that much of this section is unnecessary. You are just using the melt output from RACMO not doing any of the SEB calculations, correct? Therefore I think this section should focus on describing the RACMO product used instead of explaining SEB modeling as this description is a bit confusing because you do not actually do this in the paper.

Line 147 – Specify that the 27 drainage basins you use are *grounded* ice. I would also specify that you consider "all ice shelves", "all grounded ice" and "all AIS ice (both floating and grounded)" as regions in this study as well.

Line 157 – "…, which are multiplied…". What is this referring to?

Lines 161-170 – It is unclear to me why this paragraph (and Fig. 2) is necessary. Has this relationship between melt and temperature already been shown in other work (ie Trusel 2015)?

Line 173 – add "binary" before "melt/no-melt signal"

Line 180 – Do you use all ERA5 data or just the hours of 6am and 6pm? I am confused because Eq. 3 sets MD* = 1 if at least one hour has T-T0 > 0, but in the line below it sounds like you are only using those two hours (6am and 6pm)?

Line 184 – I believe this is the first time you use the "RMSE" acronym so you should define it here.

Line 187 – It might be helpful to introduce the concept of "mask matrices" for each region in Section 2.4.

Line 190 – I think it is a bit confusing how you use the work "region". Here you say there are 38 regions which I understand to be the 27 drainage basins + ice shelf regions + all ice shelves + all grounded ice + all of Antarctica? This is not entirely clear in all places in the text because the word "region" is also used to describe the ice shelf regions. Maybe consider changing to "area of interest" when talking about the 38 "regions of interest".

Lines 192 – 194 – The text in parenthesis can be deleted because it clutters up the sentence and is mentioned in the data section.

Section 3.2.2 – Perhaps provide an introductory sentence to define the DDF and explain why it is necessary. (I know this is in the intro but might be useful to mention briefly again here)

Section 3.3 – What exactly are you testing here? I find this section to be very confusing!

Line 213 - "45" This number has no context in this sentence. 45 what?

Line 224 – change "AIS and ice shelves" to "*whole* AIS and *all* ice shelves".

Figure 3 (and 4) – Your y-label is "RMSE per computing cell". Is this correct or should it be the regional RMSE?"

Line 225 – "Lower ability" – what exactly do you mean by this?

Lines 226 – 230 – I believe that these sentences can be more succinct and that the part describing the Jakobs et al 2020 study is largely unnecessary and confusing (ie what is "unrecognizability". I think you could shorten this to something like: "In Fig. 3a-a the RMSE at $T_0$=0 $^\circ$C is larger than at $T_0$=-1.8 $^\circ$C (our optimal threshold temperature). This finding indicates that using $T_0$=0 $^\circ$C as a melt threshold may miss events, a finding consistent with other work (Jakobs et al 2020).

Lines 230 – 234 – What exactly are you trying to say with this sentence? Overall, I think this paragraph can be made much shorter. You are just explaining why you round the DDFs right? I would consider mentioning that you choose to round the DDFs at the beginning of the paragraph (right now it is a bit lost at the end of a long paragraph), and then using the rest of the paragraph to explain why you do this.

Line 253 – "the optimal DDF better estimates…" Which DDF? DDF = 2.8 mm… or the one that is calculated for each basin?

Line 259-260 – "This may lead to a single…" I am confused why you mention this because in your work you do not use a singly parameterization for all drainage basins (Table 2).

Section 4.2 – In this section the analysis of melt days and melt volume are intermixed throughout. I would consider separating these two analyses because they use completely different products (satellite obs. Vs RACMO). This section is also really just further *evaluation* of how your PDD model captures trends/variability in surface melt. You aren't really providing any new information here about AIS melt variability and trends (that cannot be obtained from already existing products). Hence, I think it might be useful to re-frame the results from this section as model evaluation.

Figure 5 & 6 – I think it would be helpful to also provide difference maps between the observations/RACMO output and the PDD results.
Figure 6d – How do you get melt volume in areas with no annual melt days in Fig. 5d (e.g. parts of the Ronne-Filchner and Ross ice shelves)?

Line 299 – *Why* do you think that the PDD model does not capture some of the trends seen in the observations?

Line 304 – 306 – "It is worth noting that on the marine edge…". Make the distinction between melt days trend and melt volume trend in this sentence as these are really two different things.

Figure 7 – Is this figure necessary in the main text? It is simply ERA5 output so why is it important?

Line 308 – "West Ice Shelf (part of the ice shelves in Wilkes Land"). By this do you mean to say that West ice shelf is located in Wilkes Land?

Lines 315 – 318 – What do you mean by "temporal stability" and what sort of "time series analysis" do you perform? What do you mean by "We gather all 27 drainage basins for the next stage of analysis"?

Figure 8 (and in-text analysis) – I mentioned this before but I would consider only performing the trend analysis from 1992-2022 so you do not include any of the years with missing satellite data.

In this section (4.2) you mention "residuals" many times. What are the residuals? What do you mean by this? I was a bit confused what you meant every time I read the word "residuals" which I think limited my understanding of the last part of section 4.2.

Section 4.3 – While RACMO and satellite observations are perhaps closer to the truth than a PDD model there are biases that exist in these products too. Are there studies that mention these biases that you could also discuss in this section?

Line 391 – "same surface melt distributions" – what do you mean by this?

**Technical corrections**

Line 6 – "Past, present or future contexts": replace "or" with "and"

Line 60 – Change "it is" in "Although *it is* empirical…" to "PDD models are". Doing this will clarify what "it" is referring to throughout that sentence.

Line 103 – change "(once in two days before 1988)" to "(once *every* two days before 1988)"

Line 104 – change "melt and no-melt" to "melt *or* no-melt"

Line 144 – change "we use" to "products used"

Line 151 – change "requires" to "require"

Line 184 – Change "in" to "for"

Line 225 – Change "at the point which T0 equals 0C" to "where T0 = 0C"

Line 288 – "That the PDD model…". Beginning this sentence differently will help the sentence read better.

---

## Author Comment (AC1)

**Estimating surface melt in Antarctica from 1979 to 2022, using a statistically parameterized positive degree-day model**

Yaowen Zheng, Nicholas R. Golledge, Alexandra Gossart, Ghislain Picard, and Marion Leduc-Leballeur
submitted to The Cryosphere (https://doi.org/10.5194/tc-2022-192)

We gratefully thank the reviewer for the time that they spent reading and reviewing the manuscript. We respond to each of the major and minor comments below. The reviewer's comments are shown in **bold text**, replies are shown in normal text, text from the original manuscript is shown in blue, and proposed changes to the manuscript are shown in red.
* * *
**Zengh et al., 2022 estimate the melt over the Antarctic Ice Sheet using a PDD model. They also carefully parametrize their model to produce similar results than satellite and RACMO estimations. The calibration of the model is particularly complete and nothing seems to be omitted for a reproducibility of the results. Obtaining a correct PDD model is interesting and important because, as the authors mention, the efficiency of the PDD in terms of computation**

10 **time allows long simulations where other more complex models do not enable it. However, most of the manuscript focuses on the calibration and the evaluation of the PDD, and only a limited part is dedicated to the new information about melt. From this point of view, the scientific issue of the manuscript could be more emphasized in the title and might have deserved a submission to another journal like GMD (this does not, however, question the quality and robustness of the study. I would add that while being comprehensive is a quality, some passages are difficult to read**

15 **because too much information is given when that information is sometimes obvious. I also have some more comments listed hereafter. In general, I don't have much to say except to advise to simplify/summarize some passages (introduction, result) and to add some nuances (see major comments).**

**Major comments:**

**One major limitation of the PDD is that all the calibration is done over present climate where melt is only limited to margins and weak. One could question the validity of the calibration in warmer climates, i.e. for projections. For instance, how could the melt computed by PDD take into account the snow albedo feedback (which is a process often not represented by PDD due to their simplicity)? Or could the PDD correctly represent areas where no melt currently occurs and will likely occur in warmer climates as it was calibrated to reproduce current surface melt? Maybe you could compare the results of a PDD calibrated over only low melt years but evaluated over high melt years (even high present-day melt years will be considered as low melt years in the future).**

**Furthermore, the calibration of the quantity of melt is solely based on the RCM RACMO, what is the impact of RACMO biases on the PDD? Although Mottram et al, 2021 showed that RACMO is one of the best models to represent the Antarctic climate, they also suggested that RACMO underestimates near-surface air temperature which could also influence melt computed by RACMO. Note that the calibration and evaluation is also not independent as you use the same values.**

Thank you for this very inspiring and constructive comment. Practically, the Antarctic surface melt data are limited for only around 40 years. This limited time period prevents us from exploring the PDD model via the selection of low/ high melt years. This, because the training and testing samples in that case would be really small (which reduces the reliability of the parameterization as the number of data points used for training is small). We agree that we did not explore the biases on the RACMO and the satellite, and there is also the question of the independence of the calibration and evaluation in our study. To address these questions: Whether the PDD model has applicability to the warmer climates? What do the training biases impact on the PDD model? How to calibrate and evaluate the PDD model from the limited datasets? Here, we conduct a number of new testings and experiments. We will add two new subsections in the Methods section and will change the entire Results and discussion section, and parts of the Abstract and Conclusions accordingly.

Here, we show our proposed changes regarding to the new methods. Please refer to our proposed new manuscript for the according results, discussions and conclusions.

We will add two new subsections (Section 3.3.2 and Section 3.3.3) in the Methods section to describe our new tests and experiments:"

**3.3.2 K-fold cross-validation**

The cross-validation technique has been developed since the 20th century (Stone, 1974) and has became a standard technique in the field of climate and weather predictions (e.g. Mason, 2008; Maraun and Widmann, 2018). It is especially suitable for the usage of statistical models that are calibrated and evaluated on the same data (Maraun and Widmann, 2018).

We consider the spatial variability of PDD parameters by parameterizing the model in each computing cell for the whole time period. However, this does not allow us to explore the variability of the PDD parameters on a temporal scope, as Ismail et al. (2023) suggest that the temporal variability of DDF should also be considered. Due to the short period of the satellite-era

[Figure]

**Figure 1.** Schematic overview of the time periods for each CV folders and the HIGH, LOW sensitivity experiments.

and the scarcity of the in situ Antarctic surface melt data (Gossart et al., 2019), our PDD model is parameterized and evaluated using the same dataset covering the past four decades.

To assess the temporal dependency of the PDD parameters, we perform an adjusted 3-fold cross-validation (hereafter 3-fold CV). The satellite melt occurrence estimates used in this study cover 38 years (four years have been omitted). Therefore, we
55  sequentially divide the satellite estimates into two 13-year folds and a 12-year fold (Figure 1a). Note that in Section 3.2.1 we calculate the RMSE between the PDD and three satellite estimates on their overlapped period, respectively, and calculate the mean of those three RMSE. However, the second fold has actually only 7 years of overlap between the satellite SMMR and SSM/I, and satellite AMSR-E. Here, we firstly calculate the mean of satellite estimates between their overlapping periods prior to the 3-fold CV and then, we perform the 3-fold CV. The 3-fold CV has three members. the first membercontains the first
60  and second fold used to parameterize the PDD model, and the third foldis used to test the model. In Member 2, we take the first and third fold to parameterize the PDD model and test the model on the second fold. In Member 3, we take the second and third fold to parameterize the PDD model and test the model on the first fold. Similarly, we repeat the calculations for the RACMO2.3p2 surface melt amount but the folds are divided into two 14-year folds and a 13-year fold (Figure 1b).

**3.3.3 Sensitivity experiments**

65  Although RACMO2.3p2 is suggested to be one of the best models on replicating Antarctic climate, a cold bias of -0.51 K for the near-surface temperatures is also reported (Mottram et al., 2021). However, it is unclear how much this cold bias influences the output of RACMO2.3p2 snowmelt simulations, at least on the spatial scale. Satellite estimates are more direct products for Antarctic surface melt. However, bias is suggested to be existed as a corollary due to the frequent replacements of satellites that happened at least four times since the satellite-era (Picard et al., 2007).

70  In order to explore how much the biases from satellite estimates and RACMO2.3p2 simulations will influence on the parameterization of the PDD model and the outputs from the parameterized PDD model, or in other words, how sensitive the

parameterization and PDD model to the satellite estimates and RACMO2.3p2 simulations, we perform two sensitivity experiments. In the first sensitivity experiment, we explore how sensitive the $T_0$ and the PDD melt-day (and CMS) outputs to the satellite estimates. We increase (HIGH run) and decrease (LOW run) 10% of the satellite estimates (Figure 1a) for each

75 computing cell then repeat the $T_0$ parameterization as described in Section 3.2.1, respectively. In the second sensitivity experiment, we explore how sensitive the DDF and the PDD melt amount outputs to the RACMO2.3p2 simulations. We increase and decrease 10% of the RACMO2.3p2 simulations (Figure 1b) for each computing cell then repeat the DDF parameterization as described in Section 3.2.2, respectively. Note that in the context of the sensitivity experiments, out optimal parameterization of $T_0$ and DDF in Section 3.2.1 and Section 3.2.2 therefore refers to the CONTROL run.

80 In addition, these sensitivity experiments can also allow us to explore the validity of our PDD model for the application on the future Antarctic surface melt. Even if our PDD parameters are temporally stable for the period that we investigate in this study, the validity of our PDD model for the application on the future Antarctic surface melt will still be uncertain given that the future predictions of Antarctic climate is warmer than the current. Therefore, accessing the PDD model behaviours between these HIGH/ LOW runs will shed a light on the applicable of the PDD model to the warmer climate scenarios.

85 We will include these two new figures into the main text. The corresponding changes in the discussions and conclusions will also be added into the new version of the manuscript.

[Figure]

**Figure 2.** (a) to (f) spatial maps for the differences between the $T_0$/ DDF parameterized in each member of the $T_0$/ DDF 3-fold CV and the optimal $T_0$/ DDF, respectively. (g) to (l) probability histograms for the $T_0$/ DDF of each $T_0$/ DDF 3-fold CV and the optimal $T_0$/ DDF, respectively. Black vertical lines indicate the mean of optimal $T_0$s/ DDFs. Red dotted vertical lines indicate the mean of $T_0$/ DDF for each member, respectively. (m) to (r) cumulative CMS/ annual melt amount for satellite estimates/ RACMO2.3p2 simulations, CONTROL (which is the PDD model run with optimal $T_0$ and DDF) and each member for the period of the testing-fold, respectively. We calculate the difference of cumulative CMS/ annual melt amount between each member and the CONTROL, at the end of the testing fold, respectively. (s) to (x) scatter plots for the CMS/ annual melt amount of each 3-fold CV member against the CONTROL, respectively. The Spearman's $\rho$ and its statistical significance for the testing fold between each member and the CONTROL are calculated, respectively.

[Figure]

**Figure 3.** (a) and (b) spatial maps for the difference between the $T_0$ parameterized in the HIGH/ LOW experiment and the CONTROL (optimal) $T_0$. (c) and (d) spatial maps for the difference between the DDF parameterized in the HIGH/ LOW experiment and the CONTROL (optimal) DDF. (e) and (g) cumulative CMS/ annual melt amount for the satellite estimates/ RACMO2.3p2 simulations and PDD outputs. Note that the period for (e) is from 1979/1980 to 2020/2021 (with 1986/1987 to 1988/1989 and 1991/1992 omitted). The period for (g) is from 1979/1980 to 2019/2020. The upper and lower boundaries of the semi-transparent shaded areas indicates the HIGH/ LOW satellite estimates and the HIGH/ LOW PDD outputs. The percentage difference annotated in the left-bottom corner is calculated between the HIGH/ LOW and the CONTROL for each variable (by "variable", we mean satellite melt occurrence data/ PDD melt occurrence and amount data/ RACMO2.3p2 melt amount data), respectively. (f) and (h) scatter plots and the Spearman's $\rho$ (with its statistical significance) for PDD outputs and satellite/ RACMO2.3p2, from each sensitivity experiment (HIGH, LOW and CONTROL).

**Minor comments:**

**1 P1L4: Consider to nuance since this will only be the case if the firn cannot absorb the additional water**

Thank you for pointing this out. We will change it accordingly in Line 4: "temperature and melt. Enhanced surface melt will  impact the mass balance of the Antarctic Ice Sheet (AIS) and,".

**2 P1L13: Satellite observation, I suggest to replace observation by estimates as melt is not directly observed by satellite but derived from brightness temperatures or absorptivity of the surface under the assumption that the presence of water at the surface is newly-produced melt. (also for P3L81)**

Thank you for pointing this out. We agree using the term "satellite estimates" is better than "satellite observations". We will replace the term "satellite observations" with "satellite estimates" for the entire manuscript.

**3 P1 L26-27: Is this 100% valid? Surface melt is projected to remain limited to ice shelves (Kittel et al., 2021) beyond 2100 where basal melt should have a much higher influence (Seroussi et al., 2020). Consider nuance.**

Thank you for pointing this out. This suggestion overlaps with the sixth comment by the other reviewer Devon Dunmire. We copy our response to that comment below:

Thank you for your suggestion. We agree. We will change Lines 21-28 from: "Surface melting is common and well-studied over the Greenland Ice Sheet (GrIS) (e.g. Mernild et al., 2011; Colosio et al., 2021; Sellevold and Vizcaino, 2021), and is known to play an important role in the net mass balance of the ice sheet and changes in global mean sea level (GMSL), both now and in the past (e.g. Ryan et al., 2019). It is likely to become even more important in the future. Even though Antarctica is currently much colder than Greenland, projected Antarctic near-surface warming (e.g. Kittel et al., 2021) means that increased surface melting is to be expected over coming decades – both in terms of area and frequency of melting. However, these are currently less understood over Antarctica than Greenland, either in the past or at present. This is concerning as surface melting will likely become an increasingly important component of Antarctic Ice Sheet (AIS) mass balance through this century and the next." to "Surface melting is common and well-studied over the Greenland Ice Sheet (GrIS) (e.g. Mernild et al., 2011; Colosio et al., 2021; Sellevold and Vizcaino, 2021), and is known to play an important role in the net mass balance of the ice sheet and changes in global mean sea level (GMSL), both now and in the past (e.g. Ryan et al., 2019). It is likely to become even more important in the future. Antarctica is currently much colder than Greenland. Antarctic ice shelves show no statistically significant trend for the annual melt days (Johnson et al., 2022) and also no significant increase in melt amount in East Antarctica in the past 40 years (Stokes et al., 2022). However, climate projections have suggested that the surface melt will increase in the next century (e.g. Trusel et al., 2015; Kittel et al., 2021; Stokes et al., 2022) – both in terms of area and volume of melting (Trusel et al., 2015; Lee et al., 2017). Studies have suggested that the Antarctic surface melt can impact

the ice sheet mass balance through the surface thinning and runoff, surface meltwater injecting to the bed and increasing ice shelf vulnerability (Bell et al., 2018; Stokes et al., 2022). However, these are currently less understood over Antarctica than Greenland, either in the past or at present. This is concerning as surface melting will likely become an increasingly important player to Antarctic environment through this century and the next.".

**4  P1 L29-L46: I would shorten these two paragraphs which, although interesting in historical terms for the evolution of ice shelves, do not bring much information directly related to your topic. This is only a suggestion, feel free to keep as it is.**

Thank you for pointing this out. We think it is better to include such information to emphasise that Antarctic surface melting is important. We will keep these paragraphs but will change the wording of the Introduction section to make the structure of the Introduction more logical.

**5  P4L105: observation => values/estimates**

Thank you for this suggestion. We agree. We will change at P4L104: "It contains daily  estimates as a binary of melt and no-melt on a $25 \times 25$ km$^2$ southern...".

**6  P8L179-181: Is the ERA5 mean also computed between 6am and 6pm in local time for each grid point?**

Thank you for pointing this out. We agree that this text is unclear. The ERA5 mean for comparing to the SMMR and SSM/I estimates is computed between the data at 6am and at 6pm in local time for each grid point, and the ERA5 mean for comparing to the AMSR-2 and AMSR-E estimates is computed between the data at 12am and at 12pm in local time for each grid point. For clarity, we will replace Lines 179-182: "Because the satellite melt day product of SMMR and SSM/I (Table 1) is retrieved from the local acquisition times around 6 am and 6 pm, we select the 6 am and 6 pm ERA5 2-m air temperature data and calculate the daily averages of the 6 am and 6 pm. For the satellite product from AMSR-E and AMSR-2 (Table 1), we repeat the calculations using the daily averages of the 12am and 12pm ERA5 2-m air temperature data as of their local acquisition times." with "Because the satellite melt day product of SMMR and SSM/I (Table 1) is retrieved from the local acquisition times at around 6am and 6pm, we compute the mean of 6 am and 6 pm ERA5 2-m air temperature data for the input T for the PDD model (Equation 3). For the satellite product from AMSR-E and AMSR-2 (Table 1), we compute the mean of 12am and 12pm ERA5 2-m air temperature data as of their local acquisition times.".

**7    P8 L186-192: These sentences can be considered as an example of too much provided details that can make the manuscript hard to read. I'd say that the only necessary information is that the RMSE from each pixel is averaged to produce a RMSE per region. All the other information seems obvious and may be non necessary. Also Figure 7 for instance, not necessary how the surface elevation is obtained.**

We thank the reviewer for these suggestions. We will change in Lines 184-197 from:

"In order to obtain the optimal $T_0$, we calculate the RMSE between the time series of the annual number of melt days for the satellite observations and the model experiments. As we treat each computing cell individually, all calculations are carried out on each cell independently in each iteration ($T_0$ experiment).

Next, we explore the optimal $T_0$ for the whole continent and by region. To do this, we multiply the mask matrices (cells inside the region have a value of one, and cells outside the region have a value of zero) by the RMSE of each $T_0$ experiment to generate the RMSE for each $T_0$ experiment on each region. The mask matrices for those regions are defined by multiplying each mask matrix of the 38 regions of interest (Figure 1) by the mask matrix of the satellite observational area (Figure A2 in the Appendix A). Then we calculate the average of RMSE across all computing cells (RMSE per computing cell) in each targeted region in each $T_0$ experiment. Although these three satellite products have different time periods (SSMI and SSM/I covers the period from 1979/1980 to 2020/2021 (1986/1987–1988/1989 and 1991/1992 omitted), AMSR-E covers the period from 2002/2003 to 2010/2011 and AMSR-2 covers the period from 2012/2013 to 2020/2021), we assume their comparability as these satellite products are derived from the same algorithm and threshold (Picard and Fily, 2006). We therefore calculate the average of the regional-average RMSE across three satellites (hereafter, the regional RMSE). Finally, we define the optimal $T_0$ of each targeted region where the $T_0$ experiment has the minimal regional RMSE.".

to

"In order to obtain the optimal $T_0$, we calculate the root-mean-square error (RMSE) between the time series of the annual number of melt days for the satellite estimates and the model experiments in their overlapped years. As we treat each computing cell individually, all calculations are carried out on each cell independently in each iteration ($T_0$ experiment). Although these three satellite products have different time periods, we assume their comparability as these satellite products are derived from the same algorithm and threshold (Picard and Fily, 2006). Therefore, we calculate the mean of RMSE between three satellite estimates for each cell. Finally, we define the optimal $T_0$ of each computing cell where the $T_0$ experiment has the minimal RMSE. If there are multi $T_0$ experiments that have same minimal RMSE for their computing cell, we calculate the mean of those $T_0$ as the optimal $T_0$ (this only happened on the cells that have very low melt days). "

**8    P10 L229: Could this melt associated with relative apparent cold conditions be related to katabatic winds in that area, maybe not correctly represented by ERA5? (ie, could ERA5 actually underestimate temperature in that region leading to suspicious values).**

Thank you for this comment. Yes, it is possible. We will cite the "Katabatic winds warm and mix the air as it flows downward and cause widespread snow erosion, explaining >3 K higher near-surface temperatures in summer and surface melt doubling

175 in the grounding zone compared with its surroundings." from Lenaerts et al. (2017), and change in Lines 372–374: "A primary limitation of the PDD model is systematically introduced by the temperature-dependency, making it difficult to accurately estimate surface melt strengthened/ weakened or triggered by other components of the surface energy budget that may accompany katabatic winds (Lenaerts et al., 2017) and climatic phenomena such as the...".

**9    P14 L283: Instead of surface temperature, use air temperature as it is the input variable of the PDD.**

180 Thank you for this suggestion. We agree using the term "air temperature" is better than "surface temperature". We will change at P14 L283: "...decreasing  air  temperature...".

**10    P16 L320-321: How do you obtain the quantity of melt from the satellite? Should it be melt area * melt days (instead of melt quantity?). Could you also compute the CMS from RACMO to add a comparison?**

Thank you for pointing this out. We meant to calculate the CMS by the product of pixel area ($km^2$) and the total annual melt
185 duration in that pixel as Trusel et al. (2012) described. In the context of our paper, the CMS is computed by the product of cell area ($km^2$) and the total annual melt days (day) in that cell. We apologize that the text is not clear. For clarity, we will change in Line 320: "...calculated by  the product of cell area ($km^2$) and the total annual melt days (day) in that cell,... ".

The current RACMO2.3p2 data are Antarctic surface melt amount on a monthly temporal resolution. In order to calculate
190 the CMS from RACMO2.3p2, we would need the daily temporal resolution RACMO2.3p2 data. We will consider adding the comparison with the RACMO2.3p2 CMS in future work.

**References**

[revised manuscript text omitted]

---

## Author Comment (AC2)

Author comment to the Referee comment by Devon Dunmire on the manuscript

**Estimating surface melt in Antarctica from 1979 to 2022, using a statistically parameterized positive degree-day model**

Yaowen Zheng, Nicholas R. Golledge, Alexandra Gossart, Ghislain Picard, and Marion Leduc-Leballeur
submitted to The Cryosphere (https://doi.org/10.5194/tc-2022-192)

We gratefully thank the reviewer for the time that she spent reading and reviewing the manuscript. We respond to each of the major and minor comments below. The reviewer's comments are shown in **bold text**, replies are shown in normal text, text from the original manuscript is shown in blue, and proposed changes to the manuscript are shown in red.
* * *
**This work presents a new method to obtain Antarctic surface melt using only near-surface air temperature and is parameterized for different regions of Antarctica. Overall, the work presented is of high quality and will improve our understanding of AIS surface melt. However, I think the presentation of the methods, results and contribution of this work needs to be modified throughout the paper to clarify how this work fits into previous studies and contributes to ongoing efforts to better understand AIS surface melt. Many of the paragraphs are a bit confusing and difficult to read because unnecessary information is present. I feel confident that once the writing is clarified and condensed a bit that this work will be a great contribution! Nice job on the figures as they are all very clear.**

**Major revisions:**

**I think the main thing that is missing for me is a quantified justification for calculating parameters for your PDD in each region or basin. What do you gain from the by-region parameterization of your PDD model? How much better does your model do in each basin/ice-shelf region compared to observations than if you just chose one value for the whole AIS?**

**Many of the sentences throughout the paper use "this" or "it" without specifying what "this" or "it" is referring to. For example, in Line 164: "It shows that most of the cells in Antarctic..."**

**You mention that topography can introduce spatial variability in PDD parameterization. In this work, you parameterize a PDD model for each basin, but what about topographic variability within each basin? I am left wondering how variable the optimal PDD model parameters are withing each basin/region?**

**I think more emphasis needs to be put on why your PDD model is useful throughout the paper(esp. in the intro and discussions sections). This work provides a new method but not really any new information about AIS melt that cannot be obtained from RACMO or satellite observations.To that end, I think the final sentence in the introduction is a bit misleading because really you just compare output from your PDD model with observations and regional climate model output of surface melt. I would suggest focusing on the novelty of the method and potential applications for it, instead of the fact that you use your model to estimate AIS surface melt from 1979-2022 (because this can be and has been done already with satellite observations and RCMs instead).**

Thank you for these very inspiring and constructive comments. We apologize that the text is confusing and the decision of using regional parameters is not very convincing. We thank you for making this great point and have decided to explore the PDD parameters on the cell-level in the revised version of the manuscript. We agree that the cell-level PDD parameterization enhances the novelty of our study. We will make sure that the new version of the manuscript, is proofread to make sentences clearer.

Below are some figures that we will add into the new version of the manuscript regarding to the cell-level PDD parameterization. The according results, discussions and conclusions will also be added into the new version of the manuscript.

[revised manuscript text omitted]

**Minor revisions:**

**1** **I find the abstract a bit vague. There is too much emphasis on introductory material (lines 1-12) while only one**

40     **sentence touches on the results (lines 16,17) and this sentence does not make sense within the rest of the abstract (e.g. what epoch?). I think this sentence does not do a good job of summarizing your results in the abstract and I am left wondering what to take away from the paper.**

Thank you for pointing this out. We agree and will replace the Abstract from: "

Surface melt is one of the primary drivers of ice shelf collapse in Antarctica. Surface melting is expected to increase in

45   the future as the global climate continues to warm, because there is a statistically significant positive relationship between air temperature and melt. Enhanced surface melt will negatively impact the mass balance of the Antarctic Ice Sheet (AIS) and, through dynamic feedbacks, induce changes in global mean sea level (GMSL). However, current understanding of surface melt in Antarctica remains limited in past, present or future contexts. Continental-scale spaceborne observations of surface melt are limited to the satellite era (1979–present), meaning that current estimates of Antarctic surface melt are typically derived from

50   surface energy balance (SEB) or positive degree-day (PDD) models. SEB models require diverse and detailed input data that are not always available and require considerable computational resources. The PDD model, by comparison, has fewer input and computational requirements and is therefor suited for exploring surface melt scenarios in the past and future. The use of PDD schemes for Antarctic melt has been less extensively explored than their application to surface melting of the Greenland Ice Sheet, particularly in terms of a spatially-varying parameterization. Here, we construct a PDD model, force it only with

55   2-m air temperature reanalysis data, and parameterize it by minimizing the error with respect to satellite observations and SEB model outputs over the period 1979 to 2022. We compare the spatial and temporal variability of surface melt from our PDD model over the last 43 years with that of satellite observations and SEB simulations. We find that the PDD model can generally capture the same spatial and temporal surface melt patterns. Although there were at most four years over/under- estimation on ice shelf regions in the epoch, these discrepancies reduce when considering the whole AIS. With the limitations discussed, we

60   suggest that an appropriately parameterized PDD model can be a valuable tool for exploring Antarctic surface melt beyond the satellite era.

    " to"

Surface melt is one of the primary drivers of ice shelf collapse in Antarctica. Surface melting is expected to increase in the future as the global climate continues to warm, because there is a statistically significant positive relationship between

65   air temperature and melt. Enhanced surface melt will impact the mass balance of the Antarctic Ice Sheet (AIS) and, through dynamic feedbacks, induce changes in global mean sea level (GMSL). However, current understanding of surface melt in Antarctica remains limited in past, present and future contexts. Here, we construct a novel cell-level positive degree-day (PDD) model, force it only with 2-m air temperature reanalysis data, and parameterize it spatially by minimizing the error with respect to satellite estimates and SEB model outputs on each computing cell over the period 1979 to 2022. We evaluate the PDD model

70    by performing a goodness-of-fit test and cross-validation. We assess the fidelity of our parameterization method, based on the performance of the PDD model when considering all computing cells as a whole, independently of to the time window chosen for parameterization. We conduct sensitivity experiments by adding $\pm 10\%$ to the training data (satellite estimates and SEB model outputs) used for PDD parameterization. We find that the PDD estimates change analogously to the variations in the training data with steady statistically significant correlations, suggesting the applicability of the PDD model to warmer and

75    colder climate scenarios. Within the limitations discussed, we suggest that an appropriately parameterized PDD model can be a valuable tool for exploring Antarctic surface melt beyond the satellite era.

"

**2    Lines 24-28: You talk about future projects of Antarctic surface melt but what about current Antarctic melt? I think it will help to add a sentence or two that covers the current state of knowledge of AIS melt (i.e. Stokes 2021,**

80    Arthur 2022, Corr 2022).

Thank you for your suggestion. We agree. We will change Lines 21-28 from: "Surface melting is common and well-studied over the Greenland Ice Sheet (GrIS) (e.g. Mernild et al., 2011; Colosio et al., 2021; Sellevold and Vizcaino, 2021), and is known to play an important role in the net mass balance of the ice sheet and changes in global mean sea level (GMSL), both now and in the past (e.g. Ryan et al., 2019). It is likely to become even more important in the future. Even though Antarctica is

85    currently much colder than Greenland, projected Antarctic near-surface warming (e.g. Kittel et al., 2021) means that increased surface melting is to be expected over coming decades – both in terms of area and frequency of melting. However, these are currently less understood over Antarctica than Greenland, either in the past or at present. This is concerning as surface melting will likely become an increasingly important component of Antarctic Ice Sheet (AIS) mass balance through this century and the next. " to "Surface melting is common and well-studied over the Greenland Ice Sheet (GrIS) (e.g. Mernild et al., 2011;

90    Colosio et al., 2021; Sellevold and Vizcaino, 2021), and is known to play an important role in the net mass balance of the ice sheet and changes in global mean sea level (GMSL), both now and in the past (e.g. Ryan et al., 2019). It is likely to become even more important in the future. Antarctica is currently much colder than Greenland. Antarctic ice shelves show no statistically significant trend for the annual melt days (Johnson et al., 2022) and also no significant increase in melt amount in East Antarctica in the past 40 years (Stokes et al., 2022). However, climate projections have suggested that surface melt

95    will increase in the current century (e.g. Trusel et al., 2015; Kittel et al., 2021; Stokes et al., 2022) – both in terms of area and volume of melting (Trusel et al., 2015; Lee et al., 2017). Studies have suggested that Antarctic surface melt can impact ice sheet mass balance through surface thinning and runoff, surface meltwater draining to the bed, and increasing ice shelf vulnerability (Bell et al., 2018; Stokes et al., 2022). However, these are currently less understood over Antarctica than Greenland, either in the past or at present. This is concerning as surface melting will likely become an increasingly important player to Antarctic

100    environment through this century and the next.".

**3 Line 28 – Explain how AIS melt impacts ice sheet mass balance.**

Thank you for pointing this out. We agree. Please see our response to the comment above.

**4 Lines 29 – 47: Much of these paragraphs is unnecessary. I think that these paragraphs draw focus away from the main topic of this paper (which is not hydrofracture and ice-shelf collapse as it may seem by reading these two paragraphs). I think this can be summarized in a few sentences related to why surface melt is important in Antarctica.**

Thank you for this suggestion. We agree. This suggestion is overlapped with the sixth comment by the Anonymous Referee #1. We copy our response to Reviewer #1 comment below:

Thank you for pointing this out. We think it is better to include such information to emphasise that Antarctic surface melting is important. We will keep these paragraphs but will change the wording of the Introduction section to make the structure of the Introduction more logical and easier to read.

**5 Line 48: "surface melt has most likely been accelerated by the rapid increase of atmospheric temperatures...". I do not believe that this statement is correct as I don't think that AIS surface melt has accelerated. Alison Banwell recently reported results at the Cryosphere 2022 conference in Reykjavik showing a statistically significant decrease in surface melt across the AIS (will be published soon). Also, the observational trends reported in Figures 5 and 6 disagree with this statement.**

Thank you for pointing this out. However, this statement is not about the AIS, but it refers to the Antarctic Peninsula. We reference the "Analyses of 50-year meteorological records have since revealed atmospheric warming on the Antarctic Peninsula, and a number of ice shelves have retreated." and "We conclude that ice-shelf extent may well be a sensitive indicator of regional climate change. The pattern of retreat provides evidence of warming in both climate regimes on the Antarctic Peninsula, but due to the high spatial gradients of mean annual air temperature, the warming was achieved by a modest migration of the climate pattern. We have still, however, to determine the precise mechanisms whereby the atmospheric warming had such a catastrophic effect on the ice shelves of the Antarctic Peninsula but it is clear that ice shevles cannot survive periods of warming that last more than a few decades." from Vaughan and Doake (1996), "The Antarctic Peninsula has experienced a major warming over the last 50 years, with temperatures at Faraday/Vernadsky station having increased at a rate of $0.56°C$ $\mathrm{decade}^{-1}$ over the year and $1.09°C$ $\mathrm{decade}^{-1}$ during the winter; both figures are statistically significant at less than the 5% level. Overlapping 30 year trends of annual mean temperatures indicate that, at all but two of the 10 coastal stations for which trends could be computed back to 1961, the warming trend was greater (or the cooling trend less) during the 1961–90 period compared with 1971–2000." from Turner et al. (2005), "Therefore all these studies suggest that the rapid warming on the AP since the 1950s and subsequent cooling since the late-1990s are both within the bounds of the large natural decadal-scale climate variability of the region." from Turner et al. (2016) and "The Antarctic Peninsula experienced rapid warming through

the second half of the twentieth century, but so far this trend has not been sustained during the twenty-first century" from Hogg and Gudmundsson (2017).

For clarity, we have changed at Lines 47-49: "Although the warming taking place over the Antarctic Peninsula has not been consistent over the past two decades (Turner et al., 2016), surface melt there has  likely been accelerated during that period by the rapid increase of local atmospheric temperatures through the late 20th century (Vaughan and Doake, 1996; Turner et al., 2005, 2016; Hogg and Gudmundsson, 2017)."

**6   One thing that I think is missing from the introduction – Why are PDD models helpful? What do they add? I think perhaps some of the info from the abstract on PDD models would be better in the introduction.**

**Line 59 – "The PDD model calculates…" All PDD models or just one in particular?**

**Line 60 – "… based on the temperature-melt relationship". Earlier (line 57) you mention that PDD models also use the precipitation field. Is this just some PDD models? Or are you talking about the ice sheet models that use precipitation fields to determine SMB?**

**Lines 65-67 – "Wake and Marshall (2015)…" I find this sentence to be a bit distracting and confusing. I think you could just simplify to: "Wake and Marshall (2015) suggest that Antarctic surface melt can be estimated solely from monthly temperature".**

**Line 69 – What do you mean by "universal usage"?**

**Line 71 – "Topographic influences" such as what?**

**Line 79 – "… ice shelf region…". Refer to Figure 1.**

**Lines 81 – Specify that you take melt volume from RAMCO.**

Thank you for these eight suggestions. As they are all related to the Introduction section, we will address all of them at once by changing Lines 56–84 from :"

[revised manuscript text omitted]

"

**7 Line 92 – Specify that you use hourly 2-m air temperature ERA5 data in the text (also on line 174).**

Thank you for this suggestion. We will change at Line 92: "...is the hourly 2-m air temperature data which...", and at Line 174: "...ERA5 hourly 2-m air temperature data...".

**8 Line 95 – ERA5 performs better at what than those other models?**

Thank you for pointing this out. ERA5 performs better at the "near-surface temperature" referred to Gossart et al. (2019). We will add it at Line 95: "...ERA5 performs better at the near-surface temperature than its predecessor ERA-Interim...".

**9   Line 109 – "around 6 am and 6 pm" – How close to 6 am and 6 pm? Does the acquisition time vary each day?**

Thank you for pointing this out. According to the Picard and Fily (2006), the SSMR and SSM/I are carried by sun-synchronous satellites. These satellites are designed to be observing convergence of their orbits twice per day with consistent local acquisition times. However, because of the convergence of satellite orbits near the poles, there might be additional observations introduced by the passing of the satellite (Picard and Fily, 2006). Therefore, there might be other observations rather than the 6 am and 6 pm satellite ascending and descending passes. Nevertheless, Picard and Fily (2006) only proceed two passes for simplicity, and the SSMR and SSM/I satellite products we use (https://snow.univ-grenoble-alpes.fr/melting/), are derived from the average of 6am and 6pm. For consistency, to compare the PDD model output with the satellite estimates, we use the 2-m temperature data that have the same time as the satellite products.

**10   Line 110 – "This dataset is being continually updated..." Consider moving this sentence to your Data Availability statement.**

Thank you for this suggestion. We agree and will move this sentence to the Data Availability statement.

**11   Line 115 – "We therefore omit those periods from our analysis" – Which analysis in particular? These periods are not omitted from the trend analysis in section 4.2. I would probably suggest to omit these no-data periods from the trend analysis as well (for both satellite and PDD model trends).**

We thank the reviewer for this comment. For clarity, we will change at Line 115: "...We therefore omit those periods from our analysis comparison to the satellite estimates..". Regarding to the Section 4.2. In the new version of the manuscript, the entire Section 4.2 will be rewritten.

**12   Line 116 – For readability, change "More recently, there is a newly developed..." to "We also use a more recently developed...".**

Thank you for this suggestion. We agree. We will change in Line 116: "More recently, there is a newly We also use a more recently developed satellite melt day dataset which uses a similar algorithm...".

**13   Lines 119-120 – It is not necessary to specify that this product has a "twice finer spatial resolution than satellite SMMR and SSM/I product" since you mention the resolution of both products.**

Thank you for pointing this out. We agree. We will change Lines 119-120: "...This dataset is on a $12.5{\times}12.5$ km$^2$ southern polar stereographic grid which has a twice finer spatial resolution than satellite SMMR and SSM/I product....".

**14** **Line 123 – Consider changing the section heading to "Regional climate model melt output". Additionally, I believe that much of this section is unnecessary. You are just using the melt output from RACMO not doing any of the SEB calculations, correct? Therefore I think this section should focus on describing the RACMO product used instead of explaining SEB modeling as this description is a bit confusing because you do not actually do this in the paper.**

Thank you for pointing this out, but we do not fully agree. We agree that we are only using the melt output from RACMO2.3p2 and are not doing any of the SEB calculations. However, the point here is to tell the reader that the data we use to parameterize the PDD model is obtained using a SEB model. The usage of the section heading "Surface energy balance model data" emphasises that the data we use is from the SEB module/routine of the model. As we mention in the earlier text that the two numerical approaches on estimating the Antarctic surface melt are SEB and PDD models. One of the novelties of this study is that we use a more sophisticated model data (the SEB model data) to parameterize another simple and computationally efficient model (the PDD model). We will replace the section heading from "Surface energy balance model data" to "Regional climate model SEB output".

We think it is better to include the information about the terms of the SEB as the DDF of the PDD model is related to all terms of the SEB (Hock, 2005). Showing the equation of the SEB and description of its terms could somehow indicate that the SEB is more sophisticated than the PDD (if the reader compare the Equation 1 and Equation 2).

**15** **Line 147 – Specify that the 27 drainage basins you use are grounded ice. I would also specify that you consider "all ice shelves", "all grounded ice" and "all AIS ice (both floating and grounded)" as regions in this study as well.**

Thank you for pointing this out. We agree. In the new version of the manuscript, we focus on the cell-level PDD parameters which makes it not not applicable to the new version. These sentences will be deleted.

**16** **Line 157 – "..., which are multiplied...". What is this referring to?**

Thank you for pointing this out, it indeed lacks a reference. This is referring to the PDD model described in the Hock (2005):

$$\sum_{i=1}^{n} M = DDF \sum_{i=1}^{n} T^{+} * \Delta t \tag{1}$$

where DDF is the degree-day factor.

We will add a citation in Line 157: "...which are multiplied by the empirical DDF (mm w.e. $°C^{-1}$ $day^{-1}$) (e.g. Hock, 2005)....".

**17   Lines 161-170 – It is unclear to me why this paragraph (and Fig. 2) is necessary. Has this relationship between melt and temperature already been shown in other work (ie Trusel 2015)?**

Thank you for this comment. We agree. Although this temperature-melt relationship has been discussed in other studies, we include such information derived from the data we use for the rigour of this study and the discussions of our results in Section 4. Nevertheless, we agree that this paragraph and the Figure 2 are not necessary in the main text. We will move this paragraph and the Figure 2 into the Appendices as the "Appendix B: Temperature-melt relationship".

**18   Line 173 – add "binary" before "melt/no-melt signal"**

Thank you for this suggestion. We agree. We will add it in Line 173: "...we firstly focus on the binary melt/no-melt signal...".

**19   Line 180 – Do you use all ERA5 data or just the hours of 6am and 6pm? I am confused because Eq. 3 sets MD* = 1 if at least one hour has T-T0 > 0, but in the line below it sounds like you are only using those two hours (6am and 6pm)?**

Thank you for pointing this out. Theoretically we would use the hourly ERA5 data, but in reality, the hourly satellite estimates for Antarctic surface melt are not available. For consistency, we use the input ERA5 data that have the same time as the satellite estimates as we described in the lines below. For clarity, we will change Lines 173-178 from: "

To parameterize the threshold temperature ($T_0$) for our PDD model, we firstly focus on the melt/no-melt signal. We use the ERA5 2-m air temperature data to force the model and run 101 numerical experiments with a set of $T_0$ ranging from -5.0 $^\circ$C to +5.0 $^\circ$C with 0.1 $^\circ$C intervals. We define a melt day ($MD^\star$) as a day during which there is at least one hour of ERA5 2-m air temperature exceeding the $T_0$. In each $T_0$ experiment, we calculate the total number of melt days from the 1st April of that year to the 31st March of the following year as the "annual number of melt days". The modified Equation 1 can be written as:

$$\text{Annual number of melt days} = \sum_{i=t_1}^{t_2} MD^\star$$

$$t_1 = 01 - \text{April} - \text{Year}$$

$$t_2 = 31 - \text{March} - \text{(Year+1)} \tag{2}$$

$$MD^\star = \begin{cases} 1 & \text{if at least one hour } T - T_0 > 0 \\ 0 & \text{otherwise} \end{cases}$$

" to

"To parameterize the threshold temperature ($T_0$) for our PDD model, we firstly focus on the binary melt/no-melt signal. We use the ERA5 2-m air temperature data to force the model and run 101 numerical experiments with a heuristic set of $T_0$ ranging from -5.0 $^\circ$C to +5.0 $^\circ$C with 0.1 $^\circ$C intervals. Practically, we find a number of cells that exceed the low boundary at -5.0 $^\circ$C,

we therefore expand the lower boundary to -10.0 °C and add another 50 numerical experiments to traverse from -10.0 to -5.1 °C. We define a melt day ($MD^\star$) as a day in which the daily input of the ERA5 2-m air temperature (T) exceeds the $T_0$. Note that the T is either the daily mean of 6 am and 6 pm or the daily mean of 12 am and 12 pm depending on the satellite estimates we compare to (detailed in the paragraph below). In each $T_0$ experiment, we calculate the total number of melt days from 1st April of that year to 31st March of the following year as the "annual number of melt days". The modified Equation 1 can be written as:

$$\text{Annual number of melt days} = \sum_{i=t_1}^{t_2} MD^\star$$

$$t_1 = 01 - \text{April} - \text{Year}$$

$$t_2 = 31 - \text{March} - (\text{Year+1})$$    (3)

$$MD^\star = \begin{cases} 1 & \text{if } T - T_0 > 0 \\ 0 & \text{otherwise} \end{cases}$$

".

**20 Line 184 – I believe this is the first time you use the "RMSE" acronym so you should define it here.**

Thank you for pointing this out. We agree. We have changed Line 184: "...we calculate the root-mean-square error (RMSE) between...".

**21 Line 187 – It might be helpful to introduce the concept of "mask matrices" for each region in Section 2.4.**

**Line 190 – I think it is a bit confusing how you use the work "region". Here you say there are 38 regions which I understand to be the 27 drainage basins + ice shelf regions + all ice shelves + all grounded ice + all of Antarctica? This is not entirely clear in all places in the text because the word "region" is also used to describe the ice shelf regions. Maybe consider changing to "area of interest" when talking about the 38 "regions of interest".**

**Lines 192 – 194 – The text in parenthesis can be deleted because it clutters up the sentence and is mentioned in the data section.**

Thank you for pointing this out. We agree. These three comments are overlapping with the 7th minor comment by the Anonymous Referee #1. We copy our response to that comment below:

We thank the reviewer for these suggestions. We will change Lines 184-197 from:

"In order to obtain the optimal $T_0$, we calculate the RMSE between the time series of the annual number of melt days for the satellite observations and the model experiments. As we treat each computing cell individually, all calculations are carried out on each cell independently in each iteration ($T_0$ experiment).

Next, we explore the optimal $T_0$ for the whole continent and by region. To do this, we multiply the mask matrices (cells inside the region have a value of one, and cells outside the region have a value of zero) by the RMSE of each $T_0$ experiment to generate the RMSE for each $T_0$ experiment on each region. The mask matrices for those regions are defined by multiplying each mask matrix of the 38 regions of interest (Figure 1) by the mask matrix of the satellite observational area (Figure A2 in the Appendix A). Then we calculate the average of RMSE across all computing cells (RMSE per computing cell) in each targeted region in each $T_0$ experiment. Although these three satellite products have different time periods (SSMI and SSM/I covers the period from 1979/1980 to 2020/2021 (1986/1987–1988/1989 and 1991/1992 omitted), AMSR-E covers the period from 2002/2003 to 2010/2011 and AMSR-2 covers the period from 2012/2013 to 2020/2021), we assume their comparability as these satellite products are derived from the same algorithm and threshold (Picard and Fily, 2006). We therefore calculate the average of the regional-average RMSE across three satellites (hereafter, the regional RMSE). Finally, we define the optimal $T_0$ of each targeted region where the $T_0$ experiment has the minimal regional RMSE.".

to

"In order to obtain the optimal $T_0$, we calculate the root-mean-square error (RMSE) between the time series of the annual number of melt days for the satellite estimates and the model experiments in their overlapped years. As we treat each computing cell individually, all calculations are carried out on each cell independently in each iteration ($T_0$ experiment). Although these three satellite products have different time periods, we assume their comparability as these satellite products are derived from the same algorithm and threshold (Picard and Fily, 2006). Therefore, we calculate the mean of RMSE between three satellite estimates for each cell. Finally, we define the optimal $T_0$ of each computing cell where the $T_0$ experiment has the minimal RMSE. If there are multi $T_0$ experiments that have same minimal RMSE for their computing cell, we calculate the mean of those $T_0$ as the optimal $T_0$ (this only happened on the cells that have very low melt days). "

**22    Section 3.2.2 – Perhaps provide an introductory sentence to define the DDF and explain why it is necessary. (I know this is in the intro but might be useful to mention briefly again here)**

Thank you for your suggestion. We agree. We have changed Line 199: "The DDF is a scaling number that controls the amount of melt. It is a lumped parameter that relates to all terms of the SEB (Hock, 2005; Ismail et al., 2023) and is suggested not to be considered as a constant number in PDD models (Ismail et al., 2023). To parameterize the DDF for our PDD model, we...".

**23    Section 3.3 – What exactly are you testing here? I find this section to be very confusing!**

**Line 213 - "45" This number has no context in this sentence. 45 what?**

Thank you for pointing this out. These two comments are related to the Section 3.3. which we will replace as follows: "

**Significance testing**

The two-sample Kolmogorov–Smirnov test (hereafter two-sample KS test) has been used in testing the significant difference between two non-Gaussian climatic distributions when parametric tests are inappropriate (e.g. Deo et al., 2009; Zheng et al., 2021). It has also been used as an alternative way to test the dissimilarity of climatic data as a validation of tests on statistical parameters such as the mean (Zheng et al., 2021). The two-sample KS test non-parametrically tests the distributional dissimilarity between two samples by quantifying the distance of two sample-derived empirical distribution functions (Lanzante, 2021). The null hypothesis is that the two samples are from the same continuous distribution. The test result returns a logical index that either accepts or rejects the null hypothesis at the 5% significance level ($p < 0.05$).

Limited by the duration of satellite era and reanalysis data, the annual data for each computing cell is no larger than 45 with non-normality. To test the significance of the optimal $T_0$ and DDF, we therefore perform the two-sample KS tests between the annual number of melt days/ melt amount from the satellite observations/ RACMO2.3p2 and from the PDD model $T_0$/ DDF experiments. We define a 'same distribution cell' as a cell with no statistically significant evidence from the two-sample KS test for the rejection of the null hypothesis (that the two samples are from the same continuous distribution). To quantify the test result in each targeted region, we calculate the percentage of the same distribution cells for each $T_0$/ DDF experiment on each targeted region. We specifically discuss and interpret the results of this test approach in Appendix B.

" with "

**Goodness-of-fit testing**

The two-sample Kolmogorov–Smirnov test (hereafter two-sample KS test) has been used in testing for significant difference between two non-Gaussian climatic distributions when parametric tests are inappropriate (e.g. Deo et al., 2009; Zheng et al., 2021). It has also been used as an alternative way to test the dissimilarity of climatic data as a validation of tests on statistical parameters such as the mean (Zheng et al., 2021). The two-sample KS test non-parametrically tests the distributional dissimilarity between two samples by quantifying the distance of two sample-derived empirical distribution functions (Lanzante, 2021). The null hypothesis is that the two samples are from the same continuous distribution. The test result returns a logical index that either accepts or rejects the null hypothesis at the 5% significance level ($p < 0.05$).

Limited by the duration of satellite era and reanalysis data, the time series of annual data for each computing cell is no larger than 45 years with non-normality. To test the goodness-of-fit of the parameterized PDD model, we therefore perform the two-sample KS tests between the time series of annual number of melt days/ melt amount from the satellite estimates/ RACMO2.3p2 and from the parameterized PDD model outputs. We define a 'same distribution cell' as a cell with no statistically significant evidence from the two-sample KS test for the rejection of the null hypothesis (that the two samples are from the same continuous distribution).

"

**24** **Line 224** – change "AIS and ice shelves" to "whole AIS and all ice shelves".

**Figure 3 (and 4)** – Your y-label is "RMSE per computing cell". Is this correct or should it be the regional RMSE?"

385 **Line 225** – "Lower ability" – what exactly do you mean by this?

**Lines 226 – 230** – I believe that these sentences can be more succinct and that the part describing the Jakobs et al 2020 study is largely unnecessary and confusing (ie what is "unrecognizability". I think you could shorten this to something like: "In Fig. 3a-a the RMSE at T0=0 oC is larger than at T0=-1.8 oC (our optimal threshold temperature). This finding indicates that using T0=0 oC as a melt threshold may miss events, a finding consistent with other work (Jakobs 390 et al 2020).

**Lines 230 – 234** – What exactly are you trying to say with this sentence? Overall, I think this paragraph can be made much shorter. You are just explaining why you round the DDFs right? I would consider mentioning that you choose to round the DDFs at the beginning of the paragraph (right now it is a bit lost at the end of a long paragraph), and then using the rest of the paragraph to explain why you do this.

395 **Line 253** – "the optimal DDF better estimates…" Which DDF? DDF = 2.8 mm… or the one that is calculated for each basin?

**Line 259-260** – "This may lead to a single…" I am confused why you mention this because in your work you do not use a singly parameterization for all drainage basins (Table 2).

**Section 4.2** – In this section the analysis of melt days and melt volume are intermixed throughout. I would consider 400 separating these two analyses because they use completely different products (satellite obs. Vs RACMO). This section is also really just further evaluation of how your PDD model captures trends/variability in surface melt. You aren't really providing any new information here about AIS melt variability and trends (that cannot be obtained from already existing products). Hence, I think it might be useful to re-frame the results from this section as model evaluation.

405 **Figure 5 & 6 – I think it would be helpful to also provide difference maps between the observations/RACMO output and the PDD results.**

**Line 299 – Why do you think that the PDD model does not capture some of the trends seen in the observations?**

**Line 304 – 306 – "It is worth noting that on the marine edge…". Make the distinction between melt days trend and melt volume trend in this sentence as these are really two different things.**

410 **Figure 7 – Is this figure necessary in the main text? It is simply ERA5 output so why is it important?**

**Line 308 – "West Ice Shelf (part of the ice shelves in Wilkes Land"). By this do you mean to say that West ice shelf is located in Wilkes Land?**

**Lines 315 – 318 – What do you mean by "temporal stability" and what sort of "time series analysis" do you perform? What do you mean by "We gather all 27 drainage basins for the next stage of analysis"?**

415 **Figure 8 (and in-text analysis) – I mentioned this before but I would consider only performing the trend analysis from 1992-2022 so you do not include any of the years with missing satellite data.**

**In this section (4.2) you mention "residuals" many times. What are the residuals? What do you mean by this? I was a bit confused what you meant every time I read the word "residuals"which I think limited my understanding of the last part of section 4.2.**

420 We thank the reviewer for those 16 comments above. These comments are all related to the Section 4.1 and Section 4.2. In the new version of the manuscript, these sections will not exist. We will re-write the whole Section 4.1 and 4.2 based on the new results on the cell-level PDD model. Please refer to the new version of the manuscript for the Section 4.1 and Section 4.2.

25 **Section 4.3 – While RACMO and satellite observations are perhaps closer to the truth than a PDD model there are biases that exist in these products too. Are there studies that mention these biases that you could also discuss**
425 **in this section?**

Thank you for this comment. We agree. This comment is partly overlapping with the major comments by the Anonymous Referee #1. Here i copy our response to that comment below:

Thank you for this very inspiring and constructive comment. Practically, the Antarctic surface melt data are limited for only around 40 years. This limited time period prevents us from exploring the PDD model via the selection of low/ high melt
430 years. This, because the training and testing samples in that case would be really small (which reduces the reliability of the parameterization as the number of data points used for training is small). We agree that we did not explore the biases on the RACMO and the satellite, and there is also the question of the independence of the calibration and evaluation in our study. To

address these questions: Whether the PDD model has applicability to the warmer climates? What do the training biases impact on the PDD model? How to calibrate and evaluate the PDD model from the limited datasets? Here, we conduct a number of new testings and experiments. We will add two new subsections in the Methods section and will change the entire Results and discussion section, and parts of the Abstract and Conclusions accordingly.

Here, we show our proposed changes regarding to the new methods. Please refer to our proposed new manuscript for the according results, discussions and conclusions.

We will add two new subsections (Section 3.3.2 and Section 3.3.3) in the Methods section to describe our new tests and experiments.

For the new subsection, please see our response to the comments by Anonymous Referee #1.

**26  Line 391 – "same surface melt distributions" – what do you mean by this?**

Thank you for pointing this out. In the new version of the manuscript, the first paragraph of the Conclusions: "We have constructed a PDD model based on the temperature-melt relationship (e.g. Hock, 2005; Trusel et al., 2015), and used it to estimate surface melt in Antarctica in the past four decades. We parameterized the PDD model by running numerical experiments on each individual computing cell to iterate over various combinations of the threshold temperature and the DDF (Section 3.2). We selected an optimal parameter combination by locating the minimal RMSE between the PDD and satellite observations, and SEB simulations. We independently performed two-sample KS tests in each experiment in order to quantify the percentage of cells that have statistically significant ($p < 0.05$) same surface melt distributions for each targeted region. We have found that rounding the PDD optimal parameters not only simplifies the calculations, without introducing considerable differences either on the RMSE or two-sample KS percentage, but also avoids suggesting a level of precision defined by the parameterisation experiments that may not be physically realistic." will be replaced with: "We have constructed a PDD model based on the temperature-melt relationship (e.g. Hock, 2005; Trusel et al., 2015), and used it to estimate surface melt in Antarctica through the past four decades. We parameterized the PDD model by running numerical experiments on each individual computing cell to iterate over various combinations of the threshold temperature and the DDF (Section 3.2). We individually selected an optimal parameter combination by locating the minimal RMSE between the PDD and satellite estimates, and SEB simulations, for each computing cell. We independently performed two-sample KS tests on each computing cell in order to assess the goodness-of-fit for the parameterized PDD model. We also temporally and spatially compared the PDD estimations, satellite estimates and RACMO2.3p2 simulations to evaluate the parameterized PDD model. We found that the PDD model has the ability to capture the main spatial and temporal features for a majority of cells in Antarctica under a range of melt regimes (Section 4.2.1). ".

**Technical corrections:**

**27 Line 6 – "Past, present or future contexts": replace "or" with "and"**

465 Thank you for pointing this out. We have changed in Line 6: "...remains limited in past, present  and  future...".

**28 Line 60 – Change "it is" in "Although it is empirical…" to "PDD models are". Doing this will clarify what "it" is referring to throughout that sentence.**

Thank you for pointing this out. We have changed in Line 60: "... Although  PDD models are empirical, it is often sufficient for estimating...".

470 ## 29 Line 103 – change "(once in two days before 1988)" to "(once every two days before 1988)"

Thank you for pointing this out. We have changed in Line 103: "... We use the satellite 42-year daily (once  every two days before 1988) Antarctic surface...".

**30 Line 104 – change "melt and no-melt" to "melt or no-melt"**

Thank you for pointing this out. We have changed in Line 104: "...as a binary of melt  or no-melt on a...".

475 ## 31 Line 144 – change "we use" to "products used"

Thank you for pointing this out. We have changed in Line 143: "... products used in this study...".

**32 Line 151 – change "requires" to "require"**

Thank you for pointing this out. We have changed in Line 151: "...approach enable fast run times and  require low computational...".

480 ## 33 Line 184 – Change "in" to "for"

Thank you for pointing this out. We have changed in Line 183: "... for each $T_0$ experiment.".

**34 Line 225 – Change "at the point which T0 equals 0C" to "where T0 = 0C"**

**Line 288 – "That the PDD model…". Beginning this sentence differently will help the sentence read better.**

Thank you for pointing these out. These two suggestions are related to the Section 4.1 and Section 4.2. We will re-write the
485 entire Section 4.1 and Section 4.2.

[revised manuscript text omitted]

---

## Author Response (AR1)

Author comment to the Editor decision by Brice Noël on the manuscript

**Estimating surface melt in Antarctica from 1979 to 2022, using a statistically parameterized positive degree-day model**

Yaowen Zheng, Nicholas R. Golledge, Alexandra Gossart, Ghislain Picard, and Marion Leduc-Leballeur
submitted to The Cryosphere (https://doi.org/10.5194/tc-2022-192)

We gratefully thank the Editor for the time that he spent reading and reviewing the manuscript and the responses to the referees. We respond to each of the Editor's comments below. The Editor's comments are shown in **bold text**, replies are shown in normal text, text from the original manuscript and the responses is shown in blue, and proposed changes to the

5  manuscript are shown in red.
* * *
**Dear Yaowen Zheng and co-authors,**

**Thank you very much for uploading your responses to our two referees. In their reviews, both referees recommend making the paper more concise, simplified, and clarified in places. I agree with these general comments, which should**

10  **be reflected in your revised manuscript.**

**I appreciate your efforts to address both reviewers' concern about the impact of biases in satellite/climate model products on estimating "optimal" PDD parameters (T0 and DDF). Please, make sure that your proposed sensitivity experiments remain consistent with the new local estimates of PDD parameters (grid-cell scale) that you suggested to reviewer #2. It remains however unclear how this procedure change, i.e., estimating PDD parameters locally (grid-cell**

15  **scale) instead of regionally (sector scale), could impact your results.**

**It is also unclear how additional Figures displayed in your response letter will fit in the revised manuscript, as they are not always discussed or referred to in your answers. Are these illustrations for the reviewers'/editor understanding, if so this should be clarified in your response letter. When using new Figures in your revised manuscript, please comply with the above comment on conciseness, i.e., focusing on relevant results that best convey your message. For instance,**

20  **additional Figure 2 in the review of referee #1 shows details that may not be essential.**

**As a general note, please answer each comment separately (even with a brief sentence), and avoid combining them as in Comment 6, 21, 23 and 24 of reviewer #2. Doing so makes it hard to assess how each point comments are addressed. The same holds for removed or restructured sections as in Comment 24 of reviewer #2. You will find some minor comments from the editor below.**

25 **Note that both referees and the editor will re-assess your revised manuscript before potential acceptance in TC.**

**Best wishes,**

**Brice Noël**

Thank you very much for these very constructive comments. In the new version of the manuscript, the regional scale parameters and discussions will not exist. We will entirely focus on our proposed novel cell-level PDD model parameterization. This does 30 not cahnge the overall finding of our research, but will bring more details and help us to refine the parameterization of the PDD model. We will majorly change the manuscript. Please then refer to our proposed new manuscript and the track-changes file.

Thank you very much for the information about writing response letters, it will help us to write better response letters in the future.

35 ___

**Minor editor comments:**

**When preparing your revised manuscript, please pay particular attention to the wording to avoid confusions.**

**In your response to reviewer #1, please consider the followings:**

**L56: "overlapping period" instead of "overlapped period"**

40 Thank you for pointing this out. We will replace the "overlapped period" with the "overlapping period".

**L65: "reconstructing" instead of "replicating"**

Thank you for pointing this out. We will replace the "replicating" with the "reconstructing".

**L68-68: Do you mean: "However, biases in satellite products are likely due to frequent equipment replacements, i.e., at least 4 times in the period YYYY-YYYY" or something equivalent?**

45 Thank you for pointing this out. We agree. We will replace the sentence: "However, bias is suggested to arise due to the frequent replacement of satellites that happened at least four times during the satellite-era (Picard et al., 2007)." with "However, biases in satellite products are likely due to frequent equipment replacements, i.e., 4 times in the period 1979–2005 (Picard and Fily, 2006; Picard et al., 2007).".

**L70-79: Do you mean? "To explore the sensitivity of PDD parameters and model outputs to biases in both the satellite and RACMO2.3p2 products, we perform two sensitivity experiments. In the first sensitivity experiment, we explore the response of T0 and the PDD melt-day (and CMS) outputs to perturbations in satellite estimates. We increase/decrease (HIGH/LOW run) satellite CMS estimates by 10% (Figure 1a) for each grid-cell then repeat the T0 parameterization as described in Section 3.2.1, respectively. In the second sensitivity experiment, we explore the sensitivity of the DDF and the PDD melt amount outputs to perturbations in RACMO2.3p2 melt estimates. We increase/decrease (HIGH/LOW run) the RACMO2.3p2 melt estimates by 10% (Figure 1b) for each grid-cell then repeat the DDF parameterization as described in Section 3.2.2, respectively. Based on these experiments, we obtain an optimal parameterization for T0 and DDF which are thereafter used in the CONTROL run." Please, clarify what you mean by "satellite estimates" (i.e., CMS) and "RACMO2.3p2 simulations" (melt amount estimates) to avoid confusions.**

Thank you for this suggestion. We agree, but apart from the last sentence. We will change at Lines 70–79 from: "In order to explore how much the biases from satellite estimates and RACMO2.3p2 simulations will influence the parameterization of the PDD model and the outputs from the parameterized PDD model, or in other words, how sensitive the parameterization and PDD model are to the satellite estimates and RACMO2.3p2 simulations, we perform two sensitivity experiments. In the first sensitivity experiment, we explore how sensitive the $T_0$ and the PDD melt-day (and CMS) outputs are to the satellite estimates. We increase (HIGH run) and decrease (LOW run) the satellite estimates by 10% (Figure 1a) for each computing cell then repeat the $T_0$ parameterization as described in Section 3.2.1, respectively. In the second sensitivity experiment, we explore how sensitive the DDF and the PDD melt totals are to the RACMO2.3p2 simulations. Again, we employ an increase and decrease of 10% of the RACMO2.3p2 simulations (Figure 1b) for each computing cell and repeat the DDF parameterization as described in Section 3.2.2. Note that in the context of the sensitivity experiments, our optimal parameterization of $T_0$ and DDF in Section 3.2.1 and Section 3.2.2 constitutes our CONTROL run." to "To explore the sensitivity of PDD parameters and model outputs to biases in both the satellite and RACMO2.3p2 products, we perform two sensitivity experiments. In the first sensitivity experiment, we explore the response of T0 and the PDD melt-day (and CMS) outputs to perturbations in satellite estimates. We increase/decrease (HIGH/LOW run) satellite CMS estimates by 10% (Figure 1a) for each grid-cell then repeat the T0 parameterization as described in Section 3.2.1, respectively. In the second sensitivity experiment, we explore the sensitivity of the DDF and the PDD melt amount outputs to perturbations in RACMO2.3p2 melt estimates. We increase/decrease (HIGH/LOW run) the RACMO2.3p2 melt estimates by 10% (Figure 1b) for each grid-cell then repeat the DDF parameterization as described in Section 3.2.2, respectively. Note that in the context of the sensitivity experiments, our optimal parameterization of $T_0$ and DDF in Section 3.2.1 and Section 3.2.2 constitutes our CONTROL run.".

**L80-84 These lines are unclear. Do you mean? "In addition, these sensitivity experiments enable us to explore potential applications of our PDD model to predict Antarctic surface melt in the future. Although our PDD parameters remain**

80 **stable for the contemporary climate, it is uncertain how they could change in a warmer climate. Exploring the variations in PDD parameters by performing the above sensitivity experiments provides some insights on the model ability to simulate melt under future warming scenarios. "**

Thank you for this suggestion. We agree. We will change at Lines 80–84 from: "In addition, these sensitivity experiments allow us to explore the validity of our PDD model for calculation of future Antarctic surface melt. Even if our PDD parameters are

85 temporally stable for the period that we investigate in this study, the validity of our PDD model for the calculation of future Antarctic surface melt is still ncertain given that the future predictions of Antarctica indicate a warmer climate than at present. Therefore, quantifying the behaviour of our PDD model between these HIGH/ LOW runs will shed a light on the applicability of the PDD model to the warmer climate scenarios." to "In addition, these sensitivity experiments enable us to explore potential applications of our PDD model to predict Antarctic surface melt in the future. Although our PDD parameters remain stable

90 for the contemporary climate, it is uncertain how they could change in a warmer climate. Exploring the variations in PDD parameters by performing the above sensitivity experiments provides some insights on the model ability to simulate melt under future warming scenarios.".

**L186-187: "… calculated by multiplying the cell area (km2) by the total annual melt days (day) in that same cell… "**

Thank you for this suggestion. We agree. We will change at Lines 186–187 from: "...calculated by the product of cell area

95 $(km^2)$ and the total annual melt days (day) in that cell..." to "... calculated by multiplying the cell area $(km^2)$ by the total annual melt days (day) in that same cell ...".

**In your response to reviewer #2, please consider the followings:**

**L70: What do you mean by "fidelity"?**

100 Thank you for pointing this out. We agree that the text is unclear. We will replace the "fidelity" with "accuracy".

**Comment 4: I agree with the two reviewers that L29-47 could be shortened to focus on the impact of melt on the mass balance (e.g., runoff and ice shelf hydrofractures).**

Thank you for your suggestion. We agree. We will remove the Lines 29–47. We will add the impact of melt on the mass balance at Lines 21–28. We will repalce at Lines 21–28 from: "Surface melting is common and well-studied over the Greenland Ice

105 Sheet (GrIS) (e.g. Mernild et al., 2011; Colosio et al.,2021; Sellevold and Vizcaino, 2021), and is known to play an important role in the net mass balance of the ice sheet and changes in global mean sea level (GMSL), both now and in the past (e.g. Ryan

et al., 2019). It is likely to become even more important in the future. Even though Antarctica is currently much colder than Greenland, projected Antarctic near-surface warming (e.g. Kittel et al., 2021) means that increased surface melting is to be expected over coming decades – both in terms of area and frequency of melting. However, these are currently less understood over Antarctica than Greenland, either in the past or at present. This is concerning as surface melting will likely become an increasingly important component of Antarctic Ice Sheet (AIS) mass balance through this century and the next." to "Surface melting is common and well-studied over the Greenland Ice Sheet (GrIS) (e.g. Mernild et al., 2011; Colosio et al., 2021; Sellevold and Vizcaino, 2021), and is known to play an important role in the net mass balance of the ice sheet and changes in global mean sea level (GMSL), both now and in the past (e.g. Ryan et al., 2019). It is likely to become even more important in the future. Antarctica is currently much colder than Greenland. Antarctic ice shelves show no statistically significant trend for the annual melt days (Johnson et al., 2022) and also no significant increase in melt amount in East Antarctica in the past 40 years (Stokes et al., 2022). However, climate projections have suggested that surface melt will increase in the current century (e.g. Trusel et al., 2015; Kittel et al., 2021; Stokes et al., 2022) – both in terms of area and volume of melting (Trusel et al., 2015; Lee et al., 2017). Studies have suggested that Antarctic surface melt can impact ice sheet mass balance through surface thinning and runoff, surface meltwater draining to the bed, and increasing ice shelf vulnerability (Bell et al., 2018; Stokes et al., 2022). However, these are currently less understood over Antarctica than Greenland, either in the past or at present. This is concerning as surface melting will likely become an increasingly important player to Antarctic environment through this century and the next.".

**Comment 5: Here, referee #2 means that previous studies show that surface melt has not shown a significant increase/acceleration since 1980, but rather even a decrease as mentioned by referee #2. In view of this, the sentence "surface melt has most likely been accelerated by the rapid increase of atmospheric temperatures" could be inappropriate, please elaborate.**

Thank you for pointing this out. We agree. We will change at Lines 47–51 from: "Although the warming taking place over the Antarctic Peninsula has not been consistent over the past two decades (Turner et al., 2016), surface melt there has likely been accelerated during the period by the rapid increase of local atmospheric temperatures through the late 20th century (Vaughan and Doake, 1996; Turner et al., 2005, 2016; Hogg and Gudmundsson, 2017). Atmospheric warming in the Antarctic Peninsula during the late 20th century may also have contributed to acceleration of outlet glaciers in the region (Tuckett et al., 2019)." to "Although the warming taking place over the Antarctic Peninsula has not been consistent over the past two decades (Turner et al., 2016), the global mean surface temperature is predicted to increase (Meinshausen et al., 2011).".

**Comment 8: "ERA5 performs better at representing near-surface temperature than its predecessors . . ."**

Thank you for this comment. We agree. We will replace: "ERA5 performs better at the near-surface temperature than its predecessor ERA-Interim..." with "ERA5 performs better at representing near-surface temperature than its predecessors...".

**Comment 9: in L298-300, why not simply stating that 151 numerical experiments were carried out for T0 ranging from -10ºC to +5ºC with a 0.1ºC interval?**

140 Thank you for pointing this out. We agree. We will replace: "We use the ERA5 2-m air temperature data to force the model and run 101 numerical experiments with a heuristic set of $T_0$ ranging from -5.0 °C to +5.0 °C with 0.1 °C intervals. Practically, we find a number of cells that exceed the low boundary at -5.0 °C, we therefore expand the lower boundary to -10.0 °C and add another 50 numerical experiments to traverse from -10.0 to -5.1 °C." with "We use the ERA5 2-m air temperature data to force the model and run 151 numerical experiments for $T_0$ ranging from -10.0 °C to +5.0 °C with a 0.1 °C interval.".

**References**

Bell, R. E., Banwell, A. F., Trusel, L. D., and Kingslake, J.: Antarctic surface hydrology and impacts on ice-sheet mass balance, Nature Climate Change, 8, 1044–1052, 2018.

Colosio, P., Tedesco, M., Ranzi, R., and Fettweis, X.: Surface melting over the Greenland ice sheet derived from enhanced resolution passive microwave brightness temperatures (1979–2019), The Cryosphere, 15, 2623–2646, 2021.

Hogg, A. E. and Gudmundsson, G. H.: Impacts of the Larsen-C Ice Shelf calving event, Nature Climate Change, 7, 540–542, 2017.

Johnson, A., Hock, R., and Fahnestock, M.: Spatial variability and regional trends of Antarctic ice shelf surface melt duration over 1979–2020 derived from passive microwave data, Journal of Glaciology, 68, 533–546, 2022.

Kittel, C., Amory, C., Agosta, C., Jourdain, N. C., Hofer, S., Delhasse, A., Doutreloup, S., Huot, P.-V., Lang, C., Fichefet, T., et al.: Diverging future surface mass balance between the Antarctic ice shelves and grounded ice sheet, The Cryosphere, 15, 1215–1236, 2021.

Lee, J. R., Raymond, B., Bracegirdle, T. J., Chadès, I., Fuller, R. A., Shaw, J. D., and Terauds, A.: Climate change drives expansion of Antarctic ice-free habitat, Nature, 547, 49–54, 2017.

Meinshausen, M., Smith, S. J., Calvin, K., Daniel, J. S., Kainuma, M. L., Lamarque, J.-F., Matsumoto, K., Montzka, S. A., Raper, S. C., Riahi, K., et al.: The RCP greenhouse gas concentrations and their extensions from 1765 to 2300, Climatic change, 109, 213–241, 2011.

Mernild, S. H., Mote, T. L., and Liston, G. E.: Greenland ice sheet surface melt extent and trends: 1960–2010, Journal of Glaciology, 57, 621–628, 2011.

Picard, G. and Fily, M.: Surface melting observations in Antarctica by microwave radiometers: Correcting 26-year time series from changes in acquisition hours, Remote sensing of environment, 104, 325–336, 2006.

Picard, G., Fily, M., and Gallée, H.: Surface melting derived from microwave radiometers: a climatic indicator in Antarctica, Annals of Glaciology, 46, 29–34, 2007.

Ryan, J., Smith, L., Van As, D., Cooley, S., Cooper, M., Pitcher, L., and Hubbard, A.: Greenland Ice Sheet surface melt amplified by snowline migration and bare ice exposure, Science Advances, 5, eaav3738, 2019.

Sellevold, R. and Vizcaino, M.: First application of artificial neural networks to estimate 21st century Greenland ice sheet surface melt, Geophysical Research Letters, 48, e2021GL092 449, 2021.

Stokes, C. R., Abram, N. J., Bentley, M. J., Edwards, T. L., England, M. H., Foppert, A., Jamieson, S. S., Jones, R. S., King, M. A., Lenaerts, J. T., et al.: Response of the East Antarctic Ice Sheet to past and future climate change, Nature, 608, 275–286, 2022.

Trusel, L. D., Frey, K. E., Das, S. B., Karnauskas, K. B., Munneke, P. K., Van Meijgaard, E., and Van Den Broeke, M. R.: Divergent trajectories of Antarctic surface melt under two twenty-first-century climate scenarios, Nature Geoscience, 8, 927–932, 2015.

Tuckett, P. A., Ely, J. C., Sole, A. J., Livingstone, S. J., Davison, B. J., Melchior van Wessem, J., and Howard, J.: Rapid accelerations of Antarctic Peninsula outlet glaciers driven by surface melt, Nature Communications, 10, 1–8, 2019.

Turner, J., Colwell, S. R., Marshall, G. J., Lachlan-Cope, T. A., Carleton, A. M., Jones, P. D., Lagun, V., Reid, P. A., and Iagovkina, S.: Antarctic climate change during the last 50 years, International journal of Climatology, 25, 279–294, 2005.

Turner, J., Lu, H., White, I., King, J. C., Phillips, T., Hosking, J. S., Bracegirdle, T. J., Marshall, G. J., Mulvaney, R., and Deb, P.: Absence of 21st century warming on Antarctic Peninsula consistent with natural variability, Nature, 535, 411–415, 2016.

Vaughan, D. G. and Doake, C.: Recent atmospheric warming and retreat of ice shelves on the Antarctic Peninsula, Nature, 379, 328–331, 1996.

---

## Referee Report (RR1)

Review of *Estimating surface melt in Antarctica from 1979 to 2022, using a statistically parameterized positive degree-day model* by Zheng et al., 2022

Zengh et al., 2022 estimate the melt over the Antarctic Ice Sheet using a PDD model. They also carefully parametrize their model to produce similar results than satellite and RACMO estimations. In general, the authors responded carefully to the reviewers' questions. I found that they improved the manuscript by specifying the missing/unclear elements. I would like to thanks the authors for all the work they did.

My only main remaining comment is that the manuscript remains long with unnecessary too detailed information. For instance, section 2.3 L107-L11 is relatively useless. L176-182: the important is to know the test you use, the hypothesis and the p-level. Only maximum 2 sentences are needed, for instance I could summarize by: We use two-sample Kolmogorov–Smirnov test (hereafter two-sample KS test) to evaluate the dissimilarity between the PDD results and RACMO2.3p2 melt outputs at a confidence level of 5%. This would even summarize a significant part of L183-L188. L190-192 : Could be removed. L231: Does the number of cells matter (ie, is it a relevant information especially if you mention it again a few sentences later)? … I would suggest the authors to make the same exercise into the whole manuscript, it won't change/reduce the quality of the work but should increase the readability of the manuscript.

Caption figure 1 : Consider to remove 'Map of', we know it's a map. (Also for Figure 3, Figure 4)

L224: Not sure why? Because in a warmer climate, the forcing would also be warmer while here you kept ERA5 constant.

L242: Do you know why there is this feature around the Amery Ice Shelves (presence of local rocks?

L270 vs L291 and L298: Could you comment here the apparent opposition between these two sentences?

L312-L314: Could you prove that ERA5 is not suited for this summer? Since RACMO is forced by a reanalysis (ERA5 or ERA-Interim), it is likely that the reanalysis actually represents the events leading to higher melt.

L315-316: If this does not represent too much work, you could test this second justification by training the PDD over only high melt years, or maybe just refer to section 4.2.2?

L459: Following Wille et al., 2019, the authors detected the atmospheric river using ERA-Interim. I guess that we can assume that the new version (ERA5) certainly reflects atmospheric river if its predecessor did.

---

## Referee Report (RR2)

**Statistically parameterizing and evaluating a positive degree-day model to estimate surface melt in Antarctica from 1979 to 2022**

Yaowen Zheng et al.

The authors made a great deal of changes to the manuscript from it's original version and I commend them for all the work they did. A few more concerns remain for me, particularly with regard to the utility of the PDD model. Can you show that it's better than already existing models? Can it really be used into the future? Additionally, I still think that the results section should be condensed and clarified. It is fairly wordy which makes it sometimes unclear. Hopefully my comments can be helpful in guiding the future direction of this manuscript. Once my more major concerns are addressed, I am happy for the paper to be accepted in the Cryosphere.

**Major comments**

My biggest concern is with the applicability of the PDD model to periods outside 1979-2022 (for which melt estimates are already available from satellite observations or RCMs. The authors mention in the conclusion (lines 467-469) the PDD model may be used "to explore Antarctic surface melt in a longer-term context into the future and over periods of the geological past when neither satellite observations nor SEB components are available." However, I worry that the PDD model is too parameterized to the specific observations/RACMO melt estimates from 1979-2022 to have much applicability outside this time, and especially into the future with expected warming. I understand that the authors attempt to address this in section 4.2.2 but I am not entirely convinced by this work, especially as the PDD cumulative melt seems to get worse compared to observations/RACMO from Member 1 to 3. I think it would be more convincing if the authors parameterize 2 different models: 1) using the full 1979-2022 period and 2) using only data from 1979-1989 and compared PDD-derived melt (spatially) from 2012-2022 for these two different models. I'm afraid that if spatial maps of PDD-derived melt look substantially different between these two models then the applicability of this PDD is extremely limited. I think some more work needs to be done to prove that this model, which is highly parameterized, can be applied to future periods.

I am also still missing a quantified justification for calculating spatially varying parameters for your PDD. For example, it would be really interesting to see how figures 4, 5, and 6 differ if you do not determine spatially varying parameters and use the same parameters for each grid cell (or use an older PDD). If you can show that melt estimates improve when you utilize spatially varying PDD parameters, this would help justify your method.

**Minor comments**

In general, I would not recommend starting sentences with "That". For example (L232): "That the dominant number of cells show a negative sign indicates that using $T_0 = 0\ ^{\circ}C$ as a melt threshold may significantly underestimate melt events" can be changed to "The majority of cells have a negative $T_0$, indicating that using $T_0 = 0\ ^{\circ}C$ as a melt threshold may substantially underestimate melt events…" I would recommend re-wording other places with this issue as well (L236, L257, L273, L295, L323, L407)

Also, I am slightly confused by the use of the term "mismatches" used throughout. Is this essentially the bias in each grid cell? For example, is Figure 5d showing a probability distribution of the bias between

the PDD and satellite CMS for all the grid cells? If this is the case, I recommend using the word "bias" as it likely much more familiar and intuitive for the reader. If not, please explain what exactly is meant by "mismatches"

L5/6 – "current understanding of surface melt in Antarctica remains limited". This statement is a bit too vague… Please specify what aspect of surface melt has limited understanding.

Introduction – I think it is important to still mention the potential impact meltwater has on ice-shelf stability (e.g. hydrofracture, ice-shelf disintegration), just perhaps not as extensively as you did in lines 29-46 of the previous version.

L26 – "However, *these* currently less understood…" Please specify what *these* refers to.

L27 – Please be more specific with regards to how meltwater will become an "increasingly important player to the Antarctic environment".

L29 – 35 – I'm sorry but I really have no clue what the point of this paragraph is! It jumps around from the Antarctic Peninsula to the melt-albedo feedback to the potential impact of clouds and atmospheric rivers. Please provide a clarified version of this paragraph with the point you are trying to make.

L51-53: "Topographic influences, such as… (Hock 2005)". This sentence should be reworded for clarity. Perhaps something like: "Spatial and temporal variability in DDF can result from topographic variation, such as the gradient of elevation which affects albedo and direct input solar radiation (Hock, 2003), and seasonal variations in radiation."

L158 – Perhaps I am missing something but what are the "three satellite products"? I thought there was one product from SMMR and SSM/I and another one from AMSR-E and AMSR-2?

Figure 2 – please describe in the caption how panel a) and b) are different.

Section 3.3.2 – do you average the results from the 3 different folds or look at each fold individually?

L213 – it is unclear to me what you mean by "biases in satellite products are likely due to frequent equipment replacements". Has this been reported in other studies? Or do the biases mostly come from some process in the development of the satellite products?

L164/165 – Perhaps combine these two sentences to something like: "The DDF is a scaling parameter that controls the meltwater production and is related to all terms of the SEB…". I am a bit thrown off by the usage of "lumped parameter" in L164.

L234-237: I think these lines can be summed up by something like "The probability distribution of $T_0$ across all grid cells is approximately normal.".

L238: Instead of saying "less than 5% probability", I think it is more intuitive to say something like "more than X standard deviations lower than the mean". I'm left wondering "less than 5% probability than what?"

L243: Replace "optimal DDFs identified by the minimal RMSE from 291 DDF experiments on each computing cell" with "optimal DDFs identified for each computing cell".

L256: Do you mean right-skewed?

L272: This phrase is a bit confusing (and misleading): "are either in a good agreement on estimating surface melt days or amount"

Figure 5c (and 6c): It would be helpful to provide some statistics for this scatter plot. For example, RMSE, R2, slope of the line of best fit.

Figure 5 e-f (and 6e-f): Please put a border around the ice shelves as well as it is difficult to see the ice-shelf edge in some places (e.g. Ross, Ronne-Filchner). Additionally, please more explicitly define what the "difference" represents: is it observations – PDD or PDD – observations? Finally, I wonder if it would also be helpful to show relative difference maps. The largest bias appears to be on the Antarctic peninsula (which is in contradiction to Figure 4 and lines 273-275) and I imagine this is just because more melt occurs on the peninsula than elsewhere and thus the bias naturally appears greater.

Figure 5 h/i (and 6 h/i): These figure panels are quite small and have a lot going on in a relatively small area. I would recommend making them bigger or perhaps making them their own figure and moving to the Supplemental material.

L329-330: What changed to cause a greater difference between the PDD and RACMO melt amounts before 1990 than after?

L336: "There are less than 5% computing cells with mismatches in the mean of lower than -15 mm w.e. or larger than +15 mm w.e. (Figure 6h)." – What do you mean by this? Is it "Less than 5% of all computing cells are 15 mm w.e. below or above the mean"?

L349-351 "The disagreement in trends, therefore, is actually between the satellite/RACMO2.3p2 and ERA5 2-m temperature, rather than between the satellite/RACMO2.3p2 and the PDD model itself." Can you create a figure to demonstrate this?

L355-367: I think this paragraph needs to be simplified and condensed. Essentially, you train on two folds to determine the optimal PDD parameters and test on the third by comparing PDD melt with observations/RACMO, correct? I think shortening this section to would help clarify and make it easier for the reader to follow.

Figure 7g-l: The histograms are not super helpful for me here. Is there a better way to visualize this? The colors for each subpanel are a bit hard to distinguish.

Figure 7s-x: It would be helpful to include other statistics for these scatter plots (RMSE, average bias, slope of line of best fit).

Figure 7o and r: I am confused why the CONTROL line is so different from RACMO2.3p2. Should this not look the same as the PDD line in Figure 5 and 6b? Maybe I am misunderstanding what the CONTROL is.

**Technical corrections**

L6 – Add "grid" before "cell-level"

L10 – delete "to" in "independently of to the time window…"

L39 – add "better" before "suited". Also make sure your spelling of "therefor" is consistent throughout the paper. In L54 you use "therefore".

L58: change "the spatial variability of PDD parameters are rarely considered" to "the spatial variability of PDD parameters *is* rarely considered".

L68: Change "varying" to "our spatially varying"

L129: change "most used" to "most commonly used"

L161 – Change "multi" to "multiple"

L164 – Change "the amount of melt" to "meltwater production". (perhaps do this elsewhere too).

L169 – Change "address" to "determine"

L186 – Add either "melt volume" or "meltwater production" after RACMO2.3p2

L190 – I think you can delete "has been developed since the 20th century (Stone, 1974) and"

L205 – change "foldis" to "fold is"

L232 – replace "dominant number" with "majority"

L234: Change "statistics of $T_0 s$" to "$T_0$ statistics across all grid cells"

L280: Add "the" before "two CMS time series" and change "are in a generally" to "are generally in"

L287: Change "symmetrically to the mean" to "symmetrically *around* the mean"

L288: There are two "the"s before "PDD model"

---

## Referee Report (RR3)

I applaud the authors for the large amount of work they did for these revisions and I think the manuscript is much improved. I especially appreciate the comparisons of the dist-PDD and the uni-PDD models and feel more convinced of the utility of a spatially varying PDD model. I have a few minor comments listed below and am happy to accept the manuscript once these minor concrns are addressed.

L23: Recently published, Banwell et al (2023) (https://doi.org/10.1029/2023GL102744) actually found a significant (but small) decrease in AIS ice shelf melt days using a snow model and microwave observations

L215 – fix T0 (change to subscript)

L343 – I am confused by the statement "because of the linear relationship between air temperature and surface melt". Is there a typo here? Doesn't Figure 9 (and other works) demonstrate a non-linear relationship between air temperature and surface melt?

Figure 7 – Specify in the caption that this is for dist-PDD (same for Figure 8)

L415 – Please add the Banwell et all 2023 citation with Bell et al and Trusel et al.

L435 – I would recommend elaborating that the "dist-PDD" and "uni-PDD" are "spatially varying" and "spatially uniform" PDDs here because it is the start of the conclusion.

L447 – "Underestimation" of what?

Appendix - Figure D4 is not referenced anywhere and should b referenced somewhere in the text

---

## Author Response (AR2)

**Statistically parameterizing and evaluating a positive degree-day model to estimate surface melt in Antarctica from 1979 to 2022**

Yaowen Zheng, Nicholas R. Golledge, Alexandra Gossart, Ghislain Picard, and Marion Leduc-Leballeur
submitted to The Cryosphere (https://doi.org/10.5194/tc-2022-192)

We are grateful to the Editor and Reviewers for the time they spent reading and reviewing our manuscript. We have carefully considered each of their comments, which are presented below. The comments from the Editor and Reviewers are shown in **bold text**, our replies are shown in normal text, text from the original manuscript are shown in blue, and proposed changes to
5 the manuscript are shown in red.
* * *
**Editor decision by Brice Noël:**

**Dear Yaowen Zheng and co-authors,**

**Thank you very much for submitting your revised manuscript, which has been reviewed by our referees. The editor**
10 **joins the referees to acknowledge the great effort you put into revising the manuscript. However, the paper remains too detailed/wordy in places, and sometimes overly technical, which can distract the reader. The manuscript also requires additional clarifications, notably on the selection of a grid-cell parameterisation approach, which should be better motivated. As noted by referee #2, it is currently unclear how the latter parameterisation remains valid for independent applications, i.e., future projections.**

15 **As a general comment, the introduction should be more focused. For instance, comparing Greenland to Antarctica is not strictly necessary (L17-19, L26). As suggested by referee #2, I recommend elaborating on the impact of surface melt on ice shelf viability, e.g., via hydrofracturing, and AIS mass balance (L24-25). The research question/objectives should be more clearly formulated: this study consists in a statistical exercise assessing the robustness of the PDD method to estimate AIS present-day melt.**

20 **Sections 3.3.1 and 3.3.2 are currently too technical. I invite the authors to reformulate these sections with a more appropriate level of detail. Referee #1 provides constructive ideas about this. Both referees spotted inconsistencies in the result section that should be checked. L270 states that the PDD model is "particularly well suited" to estimate melt over the Peninsula sector. L289-292 and L298-299 draw the opposite conclusion.**

**Based on the above, I deem that major revisions are required. Note that referee #2 and the editor will re-assess the** revised manuscript before acceptance in TC.

**Below, the authors can find additional minor suggestions.**

Thank you for your constructive comments. We will take into account both your suggestions and those of the Reviewers to improve our manuscript. Specifically, we will work on refining our wording and phrasing, and streamlining the sections that were recommended. In addition, we plan to conduct additional experiments to investigate the use of single combination parameters and the applicability of PDD to warmer temperatures. We will also modify the color scheme for Figure 3 and Figure 8 to ensure they meet the accessibility requirements of the Coblis – Color Blindness Simulator.
* * *
**Minor suggestions by Brice Noël:**

**Section 2.2: Here the authors should mention that satellites detect the presence of surface liquid water, not necessarily surface melt.**

Thank you for pointing this out. We agree. We will change Lines 89–92 from: "It contains daily estimates as a binary of melt or no-melt on a $25\times25$ km southern polar stereographic grid. It is obtained by applying the melt detecting algorithm (Torinesi et al., 2003; Picard and Fily, 2006) on the scanning Multichannel Microwave Radiometer (SMMR) and three Special Sensor Microwave Imager (SSM/I) observed passive-microwave data from the National Snow and Ice Data Center (NSIDC) (Picard and Fily, 2006)." with "The dataset contains daily estimates as a binary of melt or no-melt on a $25\times25$ km$^2$ southern polar stereographic grid. The dataset is obtained by applying the melt detecting algorithm (Torinesi et al., 2003; Picard and Fily, 2006) to detect the presence of surface liquid water on the scanning Multichannel Microwave Radiometer (SMMR) and three Special Sensor Microwave Imager (SSM/I) observed passive-microwave data from the National Snow and Ice Data Center (NSIDC) (Picard and Fily, 2006).".

**L110: Remove "the" in front of "RACMO2.3p2". This holds for the whole manuscript.**

Thank you for pointing this out. We will remove "the" in front of "RACMO2.3p2" for the whole manuscript.

**Section 3.1: Here the authors should (at least briefly) mention limitations of the PDD method. Otherwise, the reader only learns about shortcomings at the very end of the manuscript (Section 4.3).**

Thank you for pointing this out. We will change at Lines 133–135: "If appropriately parameterized, the temperature-index approach offers accurate performance (Ohmura, 2001) and provides a robust surface melt representation. However, because of

the temperature dependency, the robustness of the temperature-index approach is therefore attributed to the temperature-melt correlation.".

**L190: "The cross-validation (CV) technique ..."**

Thank you for pointing this out. We agree. We change at Lines 190–191: "The cross-validation (CV) technique has been developed since the 20th century (Stone, 1974) and has became a standard technique in the field of climate and weather predictions (e.g. Mason, 2008; Maraun and Widmann, 2018).".

**L217: Here you use the acronym CMS, which is defined for the first time in L277. Please define CMS here first.**

Thank you for pointing this out. We agree. We will change at Lines 215–217: "To explore the sensitivity of PDD parameters and model outputs to biases in both the satellite and RACMO2.3p2 products, we perform two sensitivity experiments. In the first sensitivity experiment, we explore the response of $T_0$, and the PDD melt-day  and cumulative melting surface (CMS) outputs, to perturbations in satellite estimates. The CMS which is also known as a melt index (e.g. Trusel et al., 2012), is calculated by multiplying the cell area $(\mathrm{km}^2)$ by the total annual melt days (day) in that same cell (Trusel et al., 2012). "

We will replace Lines 277–280 from: "To do this, we calculate the cumulative melting surface (CMS) (day $\mathrm{km}^2$) for satellite estimates and PDD outputs, respectively. The CMS which is also known as a melt index (e.g. Trusel et al., 2012), is calculated by multiplying the cell area $(\mathrm{km}^2)$ by the total annual melt days (day) in that same cell (Trusel et al., 2012)." with "To do this, we calculate the CMS (day $\mathrm{km}^2$) for satellite estimates and PDD outputs, respectively.".

**L224-227: These lines could be removed as they anticipate on results discussed later in the text. For instance, you state that PDD parameters remain stable in the contemporary climate, which is discussed much later in the text.**

Thank you for this suggestion. We agree. We will remove the Lines 224–227.

**L257-262: These lines are unclear. Could you remove or reformulate?**

Thank you for this suggestion. We agree. We will remove Lines 257–262: "Figure 3d summarizes the statistics of DDFs. The probability distribution of the DDFs is asymmetrical and strongly left skewed (Figure 3d). ~~We see that nearly 50% of the DDFs are in the range 1 to 2.5 mm w.e. $\mathrm{°C}^{-1}$ $\mathrm{day}^{-1}$. That the majority of the DDFs are low may be associated with the negative $T_0$s defined in the $T_0$ experiments. This is because, (1) the parametrization of the $T_0$ and DDF is sequential. The optimal $T_0$s are substituted into the Equation 2 (Section 3.2.2) as a predefined variable for the DDF experiments, which means our decision on the optimal $T_0$ will influence the decision making for the optimal DDF; (2) a low negative optimal $T_0$ may cause more degrees above the $T_0$ leading to a low optimal DDF that works in conjunction with the sum of the degrees above a vey low $T_0$.~~".

**L311: Here you use the acronym SAM, which is defined for the first time in L345. Please define SAM here first.**

Thank you for pointing this out. We agree. We will change at Lines 311-313: "It is suggested to be driven by the  Southern Annular Mode (SAM), because of an inverse relationship between the number of melt days in Dronning Maud Land and the southward migration of the southern Westerly Winds (Johnson et al., 2022).".

We will change at Lines 344–346: "This could be explained by other players driving surface melting, such as the  SAM (Torinesi et al., 2003; Tedesco and Monaghan, 2009; Johnson et al., 2022) which explains ∼ 11%–36% of the melt day variability (Johnson et al., 2022).".

**From Section 4.2 onward: consider using "bias" instead of "mismatch" in the main text and figures/captions.**

Thank you for this suggestion. We agree. We will replace "mismatch" with "bias" in the main text and figures/ captions.
* * *
**Major comments by Anonymous Referee #1:**

**Zengh et al., 2022 estimate the melt over the Antarctic Ice Sheet using a PDD model. They also carefully parametrize their model to produce similar results than satellite and RACMO estimations. In general, the authors responded carefully to the reviewers' questions. I found that they improved the manuscript by specifying the missing/unclear elements. I would like to thanks the authors for all the work they did.**

**My only main remaining comment is that the manuscript remains long with unnecessary too detailed information. For instance, section 2.3 L107-L11 is relatively useless. L176-182: the important is to know the test you use, the hypothesis and the p-level. Only maximum 2 sentences are needed, for instance I could summarize by: We use two-sample Kolmogorov–Smirnov test (hereafter two-sample KS test) to evaluate the dissimilarity between the PDD results and RACMO2.3p2 melt outputs at a confidence level of 5%. This would even summarize a significant part of L183-L188. L190-192 : Could be removed. L231: Does the number of cells matter (ie, is it a relevant information especially if you mention it again a few sentences later)? ... I would suggest the authors to make the same exercise into the whole manuscript, it won't change/reduce the quality of the work but should increase the readability of the manuscript.**

Thank you very much for your time and for your very constructive comments. We agree that some sentences are unnecessary.

According to your comments, we will change at Lines 107–109: " ".

We will replace at Lines 176–188 from: "The two-sample Kolmogorov–Smirnov test (hereafter two-sample KS test) has been used in testing for significant difference between two non-Gaussian climatic distributions when parametric tests are

inappropriate (e.g. Deo et al., 2009; Zheng et al., 2021). It has also been used as an alternative way to test the dissimilarity of climatic data as a validation of tests on statistical parameters such as the mean (Zheng et al., 2021). The two-sample KS test non-parametrically tests the distributional dissimilarity between two samples by quantifying the distance between two sample-derived empirical distribution functions (Lanzante, 2021). The null hypothesis is that the two samples are from the same continuous distribution. The test result returns a logical index that either accepts or rejects the null hypothesis at the 5% significance level ($p < 0.05$).

Limited by the duration of satellite era and reanalysis data, the time series of annual data for each computing cell is no larger than 45 years with non-normality. To test the goodness-of-fit of the parameterized PDD model, we therefore perform the two-sample KS tests between the time series of annual number of melt days/ melt amount from the satellite estimates/ RACMO2.3p2 and from the parameterized PDD model outputs. We define a 'same distribution cell' as a cell with no statistically significant evidence from the two-sample KS test for the rejection of the null hypothesis (that the two samples are from the same continuous distribution)." with "Limited by the duration of satellite era and reanalysis data, the time series of annual data for each computing cell is no larger than 45 years with non-normality. We use two-sample Kolmogorov–Smirnov test (hereafter two-sample KS test) to evaluate the dissimilarity between the PDD results and RACMO2.3p2 melt outputs at a confidence level of 5%. We define a 'same distribution cell' as a cell with no statistically significant evidence from the two-sample KS test for the rejection of the null hypothesis (that the two samples are from the same continuous distribution).".

We will remove Lines 190–192: "".

We will change at Line 231: "...computing cell . The optimal $T_0$ for almost...".

We will change at Line 237: "...given the large sample size of the $T_0$s . There is...".

**Minor comments by Anonymous Referee #1:**

**Caption figure 1 : Consider to remove 'Map of', we know it's a map. (Also for Figure 3, Figure 4)**

Thank you for this suggestion. We agree. We will change caption figure 1: "The research domain and 27 Antarctic drainage basins (Zwally et al., 2012) used in this study.". We will change caption figure 3: "(a) The optimal $T_0$ (°C) of each computing cell. (b) The optimal DDF ($\mathrm{mm\ w.e.\ °C^{-1}\ day^{-1}}$) for each computing cell." and caption figure 4: "The two-sample KS test results....". We will also remove the "Spatial map of" for Figure 7 and 8.

**L224: Not sure why? Because in a warmer climate, the forcing would also be warmer while here you kept ERA5 constant.**

140    Thank you for pointing this out. We agree that our sensitivity experiments do not provide enough evidence for the applicability of the PDD model to warmer climate scenarios. We will conduct additional temperature-melt sensitivity experiments by adding constant temperature perturbations to ERA5 2-m temperature field and use the changed 2-m temperature to force the PDD model.

[Figure]

**New Figure 9.** (a) scatter plot between annual mean 2-m temperature ($T_{2m}$) and Antarctic annual melt totals for each temperature-melt sensitivity experiment for the period from 1979/1980 to 2021/2022. (b) boxplot of Antarctic annual melt totals for each temperature-melt sensitivity experiment for the period from 1979/1980 to 2021/2022.

We will replace in Lines 11–14 from: "We conduct sensitivity experiments by adding ±10% to the training data (satellite
145    estimates and SEB model outputs) used for PDD parameterization. We find that the PDD estimates change analogously to the variations in the training data with steady statistically significant correlations, suggesting the applicability of the PDD model to warmer and colder climate scenarios." with "We conduct a sensitivity experiment by adding ±10% to the training data (satellite estimates and SEB model outputs) used for PDD parameterization, and a sensitivity experiment by adding constant temperature perturbations (+1 °C, +2 °C, +3 °C, +4 °C, and +5 °C) to the 2-m air temperature field to force the PDD model.
150    We find that the PDD estimates change analogously to the variations in the training data with steady statistically significant correlations, and the PDD estimates increase nonlinearly with the temperature perturbations, suggesting the consistency of our parameterization and the applicability of the PDD model to warmer climate scenarios. "

We will change at Lines 224–227: "
155

by performing the above sensitivity experiments provides some insights on the model ability to simulate melt under future warming scenarios. To assess the applicability of our PDD model in simulating melt under warmer climate scenarios, we conduct temperature-melt sensitivity experiments. To do this, we add constant temperature perturbations of +1 °C, +2 °C, +3 °C, +4 °C, and +5 °C to the whole 43-year (1979/1980 to 2021/2022) ERA5 2-m temperature field to force our PDD model.".

We will change at Lines 421–425: "...with the same order of magnitude to both the satellite estimates and RACMO2.3p2 simulations, suggesting that  our parameterization method is consistent to both the high and low melt scenarios.

Figure 9 shows the results from our temperature-melt sensitivity experiments. We see a nonlinear increase in our dist-PDD estimates of Antarctic surface melt totals as the temperature perturbation gradually rises from +0 °C to +5 °C. It is not surprising that both the mean and standard deviation increase, given the anticipated nonlinear growth in melt volume resulting from the expansion of both the melt area and amount. The nonlinearity of temperature-melt sensitivity of our dist-PDD model is consistent with the nonlinearity temperature-melt relationship that reported by other studies (Trusel et al., 2015; Bell et al., 2018), further implying the applicability of our dist-PDD model to warmer climate scenarios. "

**L242: Do you know why there is this feature around the Amery Ice Shelves (presence of local rocks?**

Thank you for pointing this out. It is possibly related to the presence of local rocks as visible on satellite images. For example, Figure 1 from Fricker et al. (2021) and Figure 1 from Spergel et al. (2021). Future work and potential improvement for our PDD model is therefore to consider the elevation correction of the 2-m temperature and the topographic features such as the presence of local rocks, ice free areas, etc.

Another reason might be that melt occurs frequently over the Amery Ice Shelf, but the total amount of meltwater remains low. Our parameterization result suggests low T0s (Figure 3a in the manuscript) agree well (KS test results from Figure 4a and low bias on mean from Figure 5e) with the satellite estimates. However, the PDD melt amount over Amery Ice Shelf does not agree well with RACMO2.3p2 simulations (KS test results from Figure 4b and positive bias on mean from Figure 6e). Figure 6e suggests a significant positive bias on PDD melt amount to RACMO2.3p2 melt amount over Amery Ice Shelf. This is probably because the T0 is too low to lead more sum of the degrees above the T0. Even with a DDF=1 mm w.e. $°C^{-1}$ $day^{-1}$ will cause much more melt compared to RACMO2.3p2. A future goal and potential improvement for our PDD model is to set a smaller range for the parameterization experiments to limit the parameters to be more physically realistic.

We will change at Line 242: "...The other area is the central Amery Ice Shelf (Figure 3a). We speculate that this feature may be related to the presence of local rocks (e.g., Fricker et al., 2021; Spergel et al., 2021), or it could be a result of frequent surface melt events over the central Amery Ice Shelf (as suggested by the low $T_0$ value), which are likely to have a low intensity (as indicated by the low DDF value).".

**L270 vs L291 and L298: Could you comment here the apparent opposition between these two sentences?**

190   Thank you for pointing this out. This comment overlaps with the 20th Minor comment by the Reviewer Devon Dunmire:
**"Figure 5 e-f (and 6e-f): Please put a border around the ice shelves as well as it is difficult to see the ice-shelf edge in some places (e.g. Ross, Ronne-Filchner). Additionally, please more explicitly define what the "difference" represents: is it observations – PDD or PDD – observations? Finally, I wonder if it would also be helpful to show relative difference maps. The largest bias appears to be on the Antarctic peninsula (which is in contradiction to Figure 4 and lines273-275)**
195   **and I imagine this is just because more melt occurs on the peninsula than elsewhere and thus the bias naturally appears greater."**

   We copy part of our replies to this comment: "The largest absolute bias appears to be on the Antarctic Peninsula and you are correct that this is because more melt occurs in that region. By looking at the relative bias we could indeed see that it is low on the Antarctic Peninsula. This is in agreement with our two-sample KS test results in Figure 4. We will add the following two
200   new figures in Appendix C."

   We will change at Lines 270–273: "Our parameterized PDD model is particularly well-suited for estimating surface melt over the ice shelves in the Antarctic Peninsula, while cells located in other ice shelves, such as the Filchner-Ronne Ice Shelf, ice shelves in Dronning Maud Land, Amery Ice Shelf and Ross Ice Shelf, are   not in a good agreement when estimating both the surface melt days and amount (Figure 4c and
205   d).".

   We will change at Lines 289–295: "The computing cells that have relatively large  absolute differences between the mean annual melt days of PDD outputs and of satellite estimates are mainly located over the ice shelves in the Antarctic Peninsula ($\sim$ -2.5 to -22.5 days), over the Abbot Ice Shelf ($\sim$ -5.5 to -12.5 days over the marine edge and $\sim$ +2.5 to +7.5 days over the interior) and over the Shackleton Ice Shelf ($\sim$ +7.5 to +12.5 days).
210      However, these cells with large absolute differences experience frequent surface melt (Figure 2a and d in the Appendix-D), meaning that the relative differences in melt are low (Figure 2g). In addition, these cells only amount to around 5% of the total computing cells (Figure 1b), and overall for all computing cells, the mean of average differences between the dist-PDD and
215   satellite annual melt days is approximately zero (-0.12 days, Figure 1b)."

   We will change at Lines 298–301: "The computing cells that have relatively large  absolute differences on STD are mainly located over the Wilkins Ice Shelf ($\sim$ +4.5 to +13.5 days) and over the south of Larsen C Ice Shelf ($\sim$ -7.5 to -10.5 days). Similar to the cells that have relatively large  absolute differences in their means, the relative differences are low (Figure D2h) and these cells amount to only a negligible proportion (less than 5%) of the total number of the computing
220   cells.".

**L312-L314: Could you prove that ERA5 is not suited for this summer? Since RACMO is forced by a reanalysis (ERA5 or ERA-Interim), it is likely that the reanalysis actually represents the events leading to higher melt.**

Thank you for pointing this out. We will explore this significant PDD melt bias in both the satellite and RACMO2.3p2 data in the year 1982/1983.

[Figure]

**New Figure E 1.** (a) and (d) 1982/1983 dist-PDD/ satellite meltday anomaly to the dist-PDD/ satellite mean meltday over the period 1979/1980–2020/2021 (with 1982/1983, 1986/1987, 1987/1988, 1988/1989 and 1991/1992 omitted). (g) absolute differences between 1982/1983 dist-PDD and satellite meltday. (b) and (e) 1982/1983 dist-PDD/ RACMO2.3p2 melt amount anomaly to the dist-PDD/ RACMO2.3p2 mean melt amount over the period 1979/1980–2019/2020 (with 1982/1983 omitted). (h) absolute differences between 1982/1983 dist-PDD and RACMO2.3p2 melt amount. (c) and (f) 1982/1983 DJF ERA5/ RACMO2.3p2 2-m air temperature anomaly to the DJF ERA5/ RACMO2.3p2 mean 2-m air temperature over the period 1979/1980–2019/2020 (with 1982/1983 omitted). Note that for all panels the satellite estimates from 2002/2003 to 2010/2011 are the average of SMMR and SSM/I, and AMSR-E. The satellite estimates from 2012/2013 to 2020/2021 are the average of SMMR and SSM/I, and AMSR-2.

225 New Figure E1 shows that both the satellite and RACMO2.3p2 have a significant positive surface melt anomaly during the 1982/1983 period over the ice shelves in the Amundsen Sea, Ross Ice Shelf, Amery Ice Shelf, and Dronning Maud Land, while PDD does not detect this event. Although both RACMO2.3p2 and ERA5 2-m temperature exhibit similar spatial patterns for the anomaly, ERA5 2-m temperature demonstrates a negative bias in the regions where PDD has lower melt days and amounts than both the satellite and RACMO2.3p2 (New Figure E1).

230 **L315-316: If this does not represent too much work, you could test this second justification by training the PDD over only high melt years, or maybe just refer to section 4.2.2?**

Thank you for pointing this out, but we do not fully agree. Using only the high melt years will considerably reduce the sample size, and thus will likely cause the PDD model to overfit to those data (i.e. the high melt year data). Intentionally selecting the high melt years will also remove the signals of multi-decadal climate variability in the timeseries. Therefore, the PDD model
235 will very likely be overfitted to those specific high melt years and will lack the ability to estimate melt when the temperatures are lower. Instead, we propose a new approach to explore our PDD applicability to warmer climate by temperature-melt sensitivity experiments. Please see our reply to your second minor comment.

**L459: Following Wille et al., 2019, the authors detected the atmospheric river using ERA-Interim. I guess that we can assume that the new version (ERA5) certainly reflects atmospheric river if its predecessor did.**

240 Thank you for pointing this out. Collow et al. (2022), the authors used ERA5 to detect the atmospheric river. We agree that it is more appropriate to remove "atmospheric river". We will change at Lines 457–459: "We suggest this underestimation corresponds to SAM-influenced climatic conditions, and that the PDD lacks the ability to accurately capture melt if it arises from effects such as föhn winds or atmospheric rivers that are not reflected in the input ERA5 2-m temperature fields used to force the calculations (e.g. Turton et al., 2020; Wille et al., 2019).".

**Major comments by Devon Dunmire:**

**The authors made a great deal of changes to the manuscript from it's original version and I commend them for all the work they did. A few more concerns remain for me, particularly with regard to the utility of the PDD model. Can you show that it's better than already existing models? Can it really be used into the future? Additionally, I still think that**
250 **the results section should be condensed and clarified. It is fairly wordy which makes it sometimes unclear. Hopefully my comments can be helpful in guiding the future direction of this manuscript. Once my more major concerns are addressed, I am happy for the paper to be accepted in the Cryosphere.**

**My biggest concern is with the applicability of the PDD model to periods outside 1979-2022 (for which melt estimates are already available from satellite observations or RCMs. The authors mention in the conclusion (lines 467-469) the**
255 **PDD model may be used "to explore Antarctic surface melt in a longer-term context into the future and over periods of the geological past when neither satellite observations nor SEB components are available." However, I worry that the PDD model is too parameterized to the specific observations/RACMO melt estimates from 1979-2022 to have much applicability outside this time, and especially into the future with expected warming. I understand that the authors attempt to address this in section 4.2.2 but I am not entirely convinced by this work, especially as the PDD cumulative**
260 **melt seems to get worse compared to observations/RACMO from Member 1 to 3. I think it would be more convincing if the authors parameterize 2 different models: 1) using the full 1979-2022 period and 2) using only data from 1979-1989 and compared PDD-derived melt (spatially) from 2012-2022 for these two different models. I'm afraid that if spatial maps of PDD derived melt look substantially different between these two models then the applicability of this PDD is extremely limited. I think some more work needs to be done to prove that this model, which is highly**
265 **parameterized, can be applied to future periods.**

**I am also still missing a quantified justification for calculating spatially varying parameters for your PDD. For example, it would be really interesting to see how figures 4, 5, and 6 differ if you do not determine spatially varying parameters and use the same parameters for each grid cell (or use an older PDD). If you can show that melt estimates improve when you utilize spatially varying PDD parameters, this would help justify your method.**

270 Thank you very much for your time and constructive comments. While we understand the rationale for using the first decade for model parameterization, we believe that it is more meaningful to consider the multi-decadal temperature variability over Antarctica. Temperature can decrease in one decade and increase in another due to the influence of internal climate variability. For instance, Turner et al. (2020) reported a positive temperature trend in the Antarctic Peninsula from 1979 to 1997, followed by a negative trend from 1999 to 2018 (referred to Figure 10, Turner et al. 2020). Using only one decade of temperature data
275 for the PDD model may not allow us to include such influences of multi-decadal climate variability. Studies have suggested that the multi-decadal climate variability such as the Amundsen Sea Low, Southern Annular Mode and Interdecadal Pacific

oscillation plays an important role on Antarctic temperature variability (e.g. Abram et al., 2014; Stenni et al., 2017; Turner et al., 2020). We therefore argue that using a longer period of temperature data provides a more accurate representation of the overall temperature trends in Antarctica and that the resulting PDD model parameterization therefore provides a more accurate representation of the overall surface melt estimations in Antarctica.

We note that the PDD cumulative CMS (Figure 5b) starts to diverge from the satellite record around the beginning of the first decade, but that they then converge again by the end of the same decade. The significant underestimation of PDD CMS is mainly attributed to the year 1982/1983, as depicted in Figure 5a. Additionally, when comparing the PDD melt amount to RACMO2.3p2, a significant PDD melt bias towards RACMO2.3p2 also occurred during the same year 1982/1983, as shown in Figure 6a.

To investigate the applicability of the PDD model beyond the 1979-2022 period, we propose two new methods: (1) exploring the temperature-melt sensitivity for our PDD model by introducing constant temperature perturbations to the ERA 2-m temperature field, which would force the PDD model, and (2) comparing the PDD model outputs with and without including this period.

The nonlinearity of the temperature-melt sensitivity in our PDD model, resulting from our temperature-melt sensitivity experiments, is consistent with the nonlinear temperature-melt relationship reported by other studies (Trusel et al., 2015; Bell et al., 2018). Additionally, a latest paper published in The Cryosphere, Vincent and Thibert (2023) confirms that "temperature-index models are able to capture non-linear responses of glacier mass balance (MB) to high deviations in air temperature and solid precipitation" and suggests "Given that detailed meteorological variables are highly unpredictable in the future, most glacier-mass projections in response to climate change in large-scale studies spanning the 21st century are still today based on temperature-index models with simple temperature and precipitation variables. It follows that the questions raised here relative to the non-linear responses of surface SMB to meteorological variables are crucial."

The temperature-melt sensitivity experiment (1) overlaps with the second minor comment by Anonymous Referee #1. We copy our reply below: "

Thank you for pointing this out. We agree that our sensitivity experiments do not provide enough evidence for the applicability of the PDD model to warmer climate scenarios. We will conduct an additional temperature-melt sensitivity experiments by adding constant temperature perturbations to ERA5 2-m temperature field and use the changed 2-m temperature to force the PDD model.

We will replace in Lines 11–14 from: "We conduct sensitivity experiments by adding $\pm 10\%$ to the training data (satellite estimates and SEB model outputs) used for PDD parameterization. We find that the PDD estimates change analogously to the variations in the training data with steady statistically significant correlations, suggesting the applicability of the PDD model to warmer and colder climate scenarios." with "We conduct a sensitivity experiment by adding $\pm 10\%$ to the training data (satellite estimates and SEB model outputs) used for PDD parameterization, and a sensitivity experiment by adding constant temperature perturbations (+1 °C, +2 °C, +3 °C, +4 °C, and +5 °C) to the 2-m air temperature field to force the PDD model. We find that the PDD estimates change analogously to the variations in the training data with steady statistically significant

[Figure]

**New Figure 9.** (a) scatter plot between annual mean 2-m temperature ($T_{2m}$) and Antarctic annual melt totals for each temperature-melt sensitivity experiment for the period from 1979/1980 to 2021/2022. (b) boxplot of Antarctic annual melt totals for each temperature-melt sensitivity experiment for the period from 1979/1980 to 2021/2022.

correlations, and the PDD estimates increase nonlinearly with the temperature perturbations, demonstrating the consistency of our parameterization and the applicability of the PDD model to warmer climate scenarios. "

We will change at Lines 224–227: "~~In addition, these sensitivity experiments enable us to explore potential applications of our PDD model to predict Antarctic surface melt in the future. Although our PDD parameters remain stable for the contemporary climate, it is uncertain how they could change in a warmer climate. Exploring the variations in PDD parameters by performing the above sensitivity experiments provides some insights on the model ability to simulate melt under future warming scenarios.~~ To assess the applicability of our PDD model in simulating melt under warmer climate scenarios, we conduct temperature-melt sensitivity experiments. To do this, we add constant temperature perturbations of +1 °C, +2 °C, +3 °C, +4 °C, and +5 °C to the whole 43-year (1979/1980 to 2021/2022) ERA5 2-m air temperature field to force our PDD model.".

We will change at Lines 421–425: "...with the same order of magnitude to both the satellite estimates and RACMO2.3p2 simulations, suggesting that  our parameterization method is consistent to both the high and low melt scenarios. Overall, the PDD model is less sensitive than the satellite estimates and RACMO2.3p2 simulations, which indicates that our PDD model can reduce the bias that the satellite and RACMO2.3p2 have on the melt products, even though their biases are unclear (Picard et al., 2007; Mottram et al., 2021).

Figure 9 shows the results from our temperature-melt sensitivity experiments. We see a nonlinear increase in our dist-PDD estimates of Antarctic surface melt totals as the temperature perturbation gradually rises from +0 °C to +5 °C. It is not surprising that both the mean and standard deviation increase, given the anticipated nonlinear growth in melt volume resulting from the expansion of both the melt area and amount. The nonlinearity of temperature-melt sensitivity of our dist-PDD model

330   is consistent with the nonlinearity temperature-melt relationship that reported by other studies (Trusel et al., 2015; Bell et al., 2018), further implying the applicability of our dist-PDD model to warmer climate scenarios. " "

Next, we explore why there is a significant PDD melt bias to both the satellite and RACMO2.3p2 data on the year 1982/1983.

[Figure]

**New Figure E 1.** (a) and (d) 1982/1983 dist-PDD/ satellite meltday anomaly to the dist-PDD/ satellite mean meltday over the period 1979/1980–2020/2021 (with 1982/1983, 1986/1987, 1987/1988, 1988/1989 and 1991/1992 omitted). (g) absolute differences between 1982/1983 dist-PDD and satellite meltday. (b) and (e) 1982/1983 dist-PDD/ RACMO2.3p2 melt amount anomaly to the dist-PDD/ RACMO2.3p2 mean melt amount over the period 1979/1980–2019/2020 (with 1982/1983 omitted). (h) absolute differences between 1982/1983 dist-PDD and RACMO2.3p2 melt amount. (c) and (f) 1982/1983 DJF ERA5/ RACMO2.3p2 2-m air temperature anomaly to the DJF ERA5/ RACMO2.3p2 mean 2-m air temperature over the period 1979/1980–2019/2020 (with 1982/1983 omitted). Note that for all panels the satellite estimates from 2002/2003 to 2010/2011 are the average of SMMR and SSM/I, and AMSR-E. The satellite estimates from 2012/2013 to 2020/2021 are the average of SMMR and SSM/I, and AMSR-2.

We will add a new Appendix E: "New Figure E1d and e suggest that there is a positive surface melt anomaly in the ice shelves around Amundsen Sea, Ross Ice Shelf, Amery Ice Shelf, and ice shelves in Dronning Maud Land during the period 1982/1983.

335 However, our dist-PDD model does not capture this event (New Figure E1a and b). Our dist-PDD model is significantly negatively biased towards both surface melt days and surface melt amounts compared to satellite estimates and RACMO2.3p2 simulations for this 1982/1983 event (New Figures E1g and h).

Both ERA5 and RACMO2.3p2 exhibit similar spatial patterns for the 1982/1983 DJF 2-m air temperature anomaly (New Figure E1c and f). Although RACMO2.3p2 is forced by ERA5 2-m air temperature, its 2-m air temperature is consistently

340 warmer than that of ERA5 during the 1982/1983 DJF period. This is particularly noticeable in the computing cells over the ice shelves around the Amundsen Sea, Ross Ice Shelf, Amery Ice Shelf, and Dronning Maud Land, where we see significant negative biases for dist-PDD surface melt days and amounts compared to satellite and RACMO2.3p2. These cells also align with the cells where negative ERA5 2-m air temperature biases towards RACMO2.3p2 are found.

We then assess the goodness-of-fit of the dist-PDD model after removing the 1982/1983 period for dist-PDD, satellite,

345 and RACMO2.3p2. The exclusion of the 1982/1983 period significantly improves the accuracy of the dist-PDD model in comparison to satellite and RACMO2.3p2 (New Figure E2). Although there is a slight negative bias of dist-PDD (excluding 1982/1983) cumulative CMS compared to satellite data (excluding 1982/1983) in the first decade, the two cumulative CMS curves converge after approximately the first decade and remain almost completely overlapped for the rest of the time period (New Figure E1a). Similarly, the cumulative melt curves for dist-PDD (excluding 1982/1983) and RACMO2.3p2 (excluding

350 1982/1983) show a slight divergence in the first decade but remain parallel for the rest of the time period (New Figure E2b). By the end of the integration period, the relative difference between dist-PDD and satellite CMS decreased from -3.06% to -0.73% (New Figure E2a), while the relative difference between dist-PDD and RACMO2.3p2 melt amounts decreased from -9.81% to -7.52% (New Figure E2b). These improvements are consistent across correlations and OLS linear regression analyses, as shown in New Table E1, indicating the enhanced performance of the dist-PDD model in estimating both surface melt days and

355 amounts compared to satellite and RACMO2.3p2 after excluding the 1982/1983 period.

On the basis of this additional experimentation we are able to confidently conclude that our model is accurate for the vast majority of the time series, and that any previously apparent bias was almost entirely due to the anomalous conditions of a single year."

[Figure]

**New Figure E 2.** (a) cumulative CMS for satellite estimates and PDD/ PDD (1982/1983 omitted) outputs from 1979/1980 to 2020/2021 (with 1986/1987 to 1988/1989 and 1991/1992 omitted). (b) cumulative annual melt amount for RACMO2.3p2 simulations and PDD/ PDD (1982/1983 omitted) outputs from 1979/1980 to 2019/2020.

**New Table E 1.** The Spearman's $\rho$ and P-value for PDD/ PDD (1982/1983 omitted) CMS/ melt amounts with the satellite CMS/ RACMO2.3p2 melt amounts. Slope, $R^2$, RMSE and P-value for the OLS fit between PDD/ PDD (1982/1983 omitted) CMS/ melt amounts and satellite CMS/ RACMO2.3p2 melt amounts. Note that the satellite estimates from 2002/2003 to 2010/2011 are the average of SMMR and SSM/I, and AMSR-E. The satellite estimates from 2012/2013 to 2020/2021 are the average of SMMR and SSM/I, and AMSR-2. All the PDD with satellite statistics are calculated over the period from 1979/1980 to 2020/2021 (with 1986/1987 to 1988/1989 and 1991/1992 omitted). All the PDD with RACMO2.3p2 statistics are calculated over the period from 1979/1980 to 2019/2020.

| Member | Spearman's $\rho$ | P-value | OLS slope | $R^2$ | RMSE (day $km^2$/ mm w.e.) | P-value |
|---|---|---|---|---|---|---|
| PDD v.s. satellite | 0.5203 | P < 0.01 | 0.3004 | 0.229 | $3.38 \times 10^6$ | P < 0.01 |
| PDD[a] v.s. satellite[a] | 0.5778 | P < 0.01 | 0.3894 | 0.325 | $3.19 \times 10^6$ | P < 0.01 |
| PDD v.s. RACMO2.3p2 | 0.8052 | P < 0.01 | 0.5307 | 0.55 | $1.42 \times 10^4$ | P < 0.01 |
| PDD[a] v.s. RACMO2.3p2[a] | 0.8486 | P < 0.01 | 0.6582 | 0.712 | $1.15 \times 10^4$ | P < 0.01 |

[a] 1982/1983 is omitted.

We will use the same parameterization method to generate spatially uniform threshold and melt coefficient for the PDD
360    model, used for all computing cells (hereafter, "uni-PDD"). We will then compare the uni-PDD with the PDD (dist-PDD)
obtained by using spatially varying parameters in order to estimate the added value of spatially varying the PDD parameters.
In New Figure 4, we calculate the two-sample KS test for the uni-PDD outputs and add the results as panels (a) and (b) to the
New Figure 4. By comparing panels (a) and (c) of the New Figure 4, we can see that the proportion of cells with the same
distribution for melt days increased. The same trend is observed for the melt amount (panels (b) and (d) of New Figure 4).

365    We will also add results from uni-PDD to our New Figure 5 and 6. We will move the statistics from New Figure 5c and 6c
to New Table 3 and 4. We will calculate the OLS slope, $R^2$, and RMSE for both the dist-PDD v.s. satellite/ RACMO2.3p2 and
uni-PDD v.s. satellite/ RACMO2.3p2.

[revised manuscript text omitted]

**Minor comments by Devon Dunmire:**

**540** **In general, I would not recommend starting sentences with "That". For example (L232): "That the dominant number of cells show a negative sign indicates that using T 0 = 0 o C as a melt threshold may significantly underestimate melt events" can be changed to "The majority of cells have a negative T0 ,indicating that using T0 = 0 o C as a melt threshold may substantially underestimate melt events..." I would recommend re-wording other places with this issue as well (L236, L257, L273, L295, L323, L407)**

**545** Thank you for this suggestion. We agree. We will replace at Lines 232–233: "That the dominant number of cells show a negative sign indicates that using T0 = 0 ∘C as a melt threshold may significantly underestimate melt events,..." with "The majority of cells have a negative T0, indicating that using T0 = 0 o C as a melt threshold may substantially underestimate melt events,...".

We will replace at Lines 236–237: "That the probability distribution of T0s is close to the normal distribution is not surpris-
**550** ing,..." with "It is not surprising that the probability distribution of T0s is close to the normal distribution,...".

L257 is overlapped with the 7th minor suggestion by the Editor. We will remove the Line 257.

We will replace at Line 273 from: "That the PDD model performs well in the Antarctic Peninsula is exciting,..." with "It is especially encouraging that the PDD model performs well in the Antarctic Peninsula".

We will replace at Line 295 from: "That the PDD model captures the main spatial patterns of melt is not surprising,..." with
**555** "It is not surprising that the PDD model captures the main spatial patterns of melt,...".

We will replace at Lines 323–325 from: "That the PDD is in a good agreement with RACMO2.3p2 on the annual melt amount is also evident by the statistically significant strong positive correlation (Spearman's $\rho$ = 0.81, p < 0.05, Figure 6c)." with "It is also evident by the statistically significant strong positive correlation (Spearman's $\rho$ = 0.8052, p < 0.01, Table 4) that the dist-PDD is in a good agreement with RACMO2.3p2 annual melt amount".

**560** We will replace at Lines 407–408 from: "That the T0 decreases/ increases with the increase/ decrease of the satellite estimates is expected,..." with "It is expected that the T0 decreases/ increases with the increase/ decrease of the satellite estimates,...".

**Also, I am slightly confused by the use of the term "mismatches" used throughout. Is this essentially the bias in each grid cell? For example, is Figure 5d showing a probability distribution of the bias between the PDD and satellite CMS for all the grid cells? If this is the case, I recommend using the word "bias" as it likely much more familiar and**
**565** **intuitive for the reader. If not, please explain what exactly is meant by "mismatches"**

Thank you for pointing this out. We agree that the term "mismatches" is unclear. This comment is overlapped with the last minor comment by the Editor. We will replace "mismatch" with "bias" in the main text and figures/ captions.

**L5/6 – "current understanding of surface melt in Antarctica remains limited". This statement is a bit too vague... Please specify what aspect of surface melt has limited understanding.**

570    Thank you for pointing this out. We will replace at Lines 5–6 from: "However, current understanding of surface melt in Antarctica remains limited in past, present and future contexts." with "However, the current understanding of surface melt in Antarctica remains limited in terms of the uncertainties of quantifying surface melt and understanding the driving processes of surface melt in past, present, and future contexts.".

**Introduction – I think it is important to still mention the potential impact meltwater has on ice-shelf stability (e.g.**
575   **hydrofracture, ice-shelf disintegration), just perhaps not as extensively as you did at Lines 29-46 of the previous version.**

Thank you for this suggestion. We will replace at Lines 25–27: "...Studies have suggested that Antarctic surface melt can impact ice sheet mass balance through surface thinning and runoff, surface meltwater draining to the bed, and increasing ice shelf vulnerability (Bell et al., 2018; Stokes et al., 2022)....". with "Studies have suggested that Antarctic surface melt
580   can impact ice sheet mass balance through surface thinning and runoff, and increasing ice shelf vulnerability that potentially influenced by the production of meltwater which can pond, drain and contribute to the structural weakness of ice shelves (Glasser and Scambos, 2008; Bell et al., 2018; Stokes et al., 2022).".

**L26 – "However, these currently less understood..." Please specify what these refers to.**

Thank you for pointing this out. We apologize that the text is not clear. Here we cite the "To move beyond simple projections
585   of modern Greenland hydrology to a warmer Antarctica requires an improved understanding of surface hydrology on ice shelves and ice sheets. There are profound knowledge gaps in our understanding of the role of firn densification, the roles of hydrofracture and meltwater-loading-induced-flexure on ice-shelf fracture and calving, and how effective surface rivers are in buffering ice shelves from collapse —these must be addressed to inform our grasp of surface hydrology." from Bell et al. (2018). We will replace at Line 26 from: "However, these are currently less understood over Antarctica than Greenland, either
590   in the past or at present." with "However, the roles of surface meltwater production in relation to ice shelf hydrofracture, surface rivers acting as buffers and ice shelf surface hydrology, are currently less understood over Antarctica than Greenland (Bell et al., 2018).".

**L27 – Please be more specific with regards to how meltwater will become an "increasingly important player to the Antarctic environment".**

595   Thank you for pointing this out. We apologize that the text is not clear. Here we will cite "The impact of surface hydrology on ice-sheet mass balance in other parts of Antarctica will grow as the extent and intensity of surface melt increases. The

ponding of meltwater on ice shelves could contribute to their collapse. Whether water is exported by ice-shelf rivers will depend on surface slope, surface conditions and the ice-shelf stress state. If predictions of increased melting are accurate, by 2100 ice shelves in the Antarctic Peninsula will probably have collapsed and all remaining ice shelves including the large Ross, Filchner-Ronne and Amery ice sheets will undergo firn densification due to the increased surface melt." and "On the grounded portions of East and West Antarctica, surface lowering due to runoff and connectivity to the bed (modes 1 and 2, Fig. 3) could become significant by 2100 in certain regions. Regions where 2100 melt rates similar to those observed in Greenland today develop on grounded Antarctic ice include the Pine Island catchment and portions of Wilkes Land, East Antarctica. " from Bell et al. (2018). We will also cite Lee et al. (2017) to include the information of potential impact of future surface melt on Antarctic terrestrial biodiversity.

The surface melting will not only impact the ice dynamics (through the influence to the collapse of ice shelves), but also the ecosystems. Therefore surface melting will likely play an increasingly important role in the Antarctic environment.

We will replace at Lines 26–28 from: "This is concerning as surface melting will likely become an increasingly important player to Antarctic environment through this century and the next." with "This is concerning as surface melting will likely become an increasingly important player in the Antarctic environment through this century and the next. Surface melting will not only impact the dynamics of the ice shelves and ice sheet through meltwater production (e.g. Bell et al., 2018), but will also impact the habitat of the Antarctic biodiversity (Lee et al., 2017).".

**L29 – 35 – I'm sorry but I really have no clue what the point of this paragraph is! It jumps around from the Antarctic Peninsula to the melt-albedo feedback to the potential impact of clouds and atmospheric rivers. Please provide a clarified version of this paragraph with the point you are trying to make.**

Thank you for this suggestion. We apologize that the text is not clear. We discuss some information about the potential impact of clouds and atmospheric rivers in Section 4.3. We will remove this paragraph at Lines 29–35: "~~Although the warming taking place over the Antarctic Peninsula has not been consistent over the past two decades (Turner et al., 2016), the global mean surface temperature is predicted to increase (Meinshausen et al., 2011). Moreover, the positive feedback of albedo, in which the absorption of shortwave radiation increases when snow melts to water, amplifies this melting (Lenaerts et al., 2017). However, recent studies have found large inter-annual variability of surface melt in Antarctica with no statistically significant trend (Kuipers Munneke et al., 2012; Johnson et al., 2022). Projecting Antarctic surface melt is therefore still a challenge, partly because of uncertainties introduced by clouds (Kittel et al., 2022), atmospheric rivers (e.g. Clem et al.,2022), or other localized climate phenomena.~~".

625 **L51-53: "Topographic influences, such as... (Hock 2005)". This sentence should be reworded for clarity. Perhaps something like: "Spatial and temporal variability in DDF can result from topographic variation, such as the gradient of elevation which affects albedo and direct input solar radiation (Hock, 2003), and seasonal variations in radiation."**

Thank you for this suggestion. We apologize that the text is not clear. We will replace at Lines 51–53 from: "Topographic influences, such as the gradient of elevation which affects albedo and direct input solar radiation (Hock, 2003), are generally

630 strongest in mountainous terrain, together with seasonal variations in radiation, and can introduce spatial and temporal variabilities of DDF, respectively (Hock, 2005). " with "Spatial and temporal variability in DDF can result from topographic variation, such as the gradient of elevation which affects albedo and direct input solar radiation (Hock, 2003), and seasonal variations in radiation.".

**L158 – Perhaps I am missing something but what are the "three satellite products"? I thought there was one product**

635 **from SMMR and SSM/I and another one from AMSR-E and AMSR-2?**

Thank you for pointing this out. We apologize for that the text is not clear. The three satellite products are: (1) SMMR and SSM/I for the period 1979–2021; (2) AMSR-E for the period 2002–2011; (3) AMSR-2 for the period 2012–2021.

We will change at Lines 157–159: "...Although these three satellite products have different time periods (Table 1), we assume their comparability as these satellite products are derived from the same algorithm and threshold (Picard and Fily, 2006)....".

640 **Figure 2 – please describe in the caption how panel a) and b) are different.**

Thank you for this suggestion. We will replace the Figure 2 caption from: "Schematic overview of the time periods for each CV folders and the HIGH, LOW sensitivity experiments." with "Schematic overview of the time periods for each CV folders and the HIGH, LOW sensitivity experiments. (a) is for satellite estimates and PDD melt day calculations. (b) is for RACMO2.3p2 simulations and PDD melt amount calculations.".

645 **Section 3.3.2 – do you average the results from the 3 different folds or look at each fold individually?**

Thank you for pointing this out. We don't average the results from the 3 different folds. The three members of our cross-validation are independent. We look at each member of the cross-validation individually.

We will change at Lines 189–192: "The 3-fold CV has three independent members.    In Member 1, we take the first

650 and second fold to parameterize the PDD model and test the model on the third fold. In Member 2, we take the first and third fold to parameterize the PDD model and test the model on the second fold. In Member 3, we take the second and third fold to parameterize the PDD model and test the model on the first fold.".

**L213 – it is unclear to me what you mean by "biases in satellite products are likely due to frequent equipment replacements". Has this been reported in other studies? Or do the biases mostly come from some process in the development of the satellite products?**

Thank you for pointing this out. As end users of the satellite products, we are not experts in satellite retrievals routines and their biases and refer to papers documenting these biases, authored by developers of these satellite products. We reference Picard and Fily (2006): "However, the period 1979–2005 includes observations from 4 different sensors whose characteristics vary. As a consequence, sensor replacement may induce artifacts in the derived melting information which may, in turn, bias the climatic analysis of the series. " and Picard et al. (2007): "As a consequence, a pixel experiencing no climatic change would have a slightly negative trend in melt duration, due to the satellite replacements (Picard and Fily, 2006). As a corollary, observed melt duration trends are negatively biased.", which points out that the satellite replacements can introduce biases to the satellite melt products.

We will change at Lines 213–214: "However, biases in satellite products are likely due to the inconsistency in the character-istics of satellite sensors caused by frequent equipment replacements, i.e., 4 times in the period 1979–2005 (Picard and Fily, 2006; Picard et al., 2007).".

**L164/165 – Perhaps combine these two sentences to something like: "The DDF is a scaling parameter that controls the meltwater production and is related to all terms of the SEB...". I am a bit thrown off by the usage of "lumped parameter" in L164.**

Thank you for this suggestion. We will replace at Lines 164–166 from: "The DDF is a scaling number that controls meltwater production. It is a lumped parameter that relates to all terms of the SEB (Hock, 2005; Ismail et al., 2023) and is suggested not to be considered as a constant number in PDD models (Ismail et al., 2023)." with "The DDF is a scaling parameter that controls the meltwater production and is related to all terms of the SEB (Hock, 2005).".

**L234-237: I think these lines can be summed up by something like "The probability distribution of T0 across all grid cells is approximately normal.".**

Thank you for this suggestion. We agree. We will replace at Lines 234–237 from: "Figure 3c summarizes the $T_0$ statistics across all grid cells. The skewness of $T_0$s is -0.63 indicating a slight left asymmetry of the probability distribution of $T_0$s. The kurtosis is slightly larger than 3 which is the kurtosis of a normal distribution. We fit a normal distribution with the same mean and standard deviation (STD) (red curve in Figure 3c). It is not surprising that the probability distribution of $T_0$s is close to the normal distribution, given the large sample size of the $T_0$s." with "The probability distribution of $T_0$ across all grid cells is approximately normal (Figure 3c).".

**L238: Instead of saying "less than 5% probability", I think it is more intuitive to say something like "more than X standard deviations lower than the mean". I'm left wondering "less than 5% probability than what?"**

Thank you for this suggestion. We agree. We will replace in Line 237–238 from: "There is a small number of cells distributed below -5.5 °C with less than 5% probability (Figure 3c)." with "There is a small number of cells distributed below -5.5 °C which is around 1.96 standard deviations lower than the mean (-5.57 °C, Figure 3c).".

**L243: Replace "optimal DDFs identified by the minimal RMSE from 291 DDF experiments on each computing cell" with "optimal DDFs identified for each computing cell".**

Thank you for this suggestion. We will replace at Lines 243–244 from: "Figure 3b shows the spatial map of the optimal DDFs identified by the minimal RMSE from 291 DDF experiments on each computing cell." with "Figure 3b shows the spatial map of the optimal DDFs identified for each computing cell.".

**L256: Do you mean right-skewed?**

Thank you for pointing this out. We apologize for the mistake. It should be "right-skewed". We will change at Lines 256–257: "Figure 3d summarizes the statistics of DDFs. The probability distribution of the DDFs is asymmetrical and strongly  right-skewed (Figure 3d).".

**L272: This phrase is a bit confusing (and misleading): "are either in a good agreement on estimating surface melt days or amount"**

Thank you for pointing this out. We will change at Lines 270–273: "Our parameterized PDD model is particularly well-suited for estimating surface melt over the ice shelves in the Antarctic Peninsula, while cells located in other ice shelves, such as the Filchner-Ronne Ice Shelf, ice shelves in Dronning Maud Land, Amery Ice Shelf and Ross Ice Shelf,  do not perform as well for both the surface melt days and amount (Figure 4).".

**Figure 5c (and 6c): It would be helpful to provide some statistics for this scatter plot. For example, RMSE, R2, slope of the line of best fit.**

Thank you for this suggestion. We will calculate the RMSE, R2 and slope of the line of best fit. However, because Figure 5c and Figure 6c are already quite busy, we propose to move the statistics to two new tables:

**New Table 3.** Summary of the statistics for Figure 5c. The Spearman's $\rho$ and P-value for dist-PDD/ uni-PDD CMS with the satellite CMS. Slope, $R^2$, RMSE and P-value for the OLS fit between dist-PDD/ uni-PDD CMS and satellite CMS. Note that the satellite estimates from 2002/2003 to 2010/2011 are the average of SMMR and SSM/I, and AMSR-E. The satellite estimates from 2012/2013 to 2020/2021 are the average of SMMR and SSM/I, and AMSR-2. All the statistics are calculated over the period from 1979/1980 to 2020/2021 (with 1986/1987 to 1988/1989 and 1991/1992 omitted).

| Member | Spearman's $\rho$ | P-value | OLS slope | $R^2$ | RMSE (day km$^2$) | P-value |
|---|---|---|---|---|---|---|
| uni-PDD v.s. satellite | 0.4881 | $P < 0.05$ | 0.3421 | 0.208 | $4.09 \times 10^6$ | $P < 0.05$ |
| dist-PDD v.s. satellite | 0.5203 | $P < 0.01$ | 0.3004 | 0.229 | $3.38 \times 10^6$ | $P < 0.05$ |

**New Table 4.** Summary of the statistics for Figure 6c. The Spearman's $\rho$ and P-value for dist-PDD/ uni-PDD melt amount with the RACMO2.3p2 melt amount. Slope, $R^2$, RMSE and P-value for the OLS fit between dist-PDD/ uni-PDD melt amount and RACMO2.3p2 melt amount. All the statistics are calculated over the period from 1979/1980 to 2019/2020.

| Member | Spearman's $\rho$ | P-value | OLS slope | $R^2$ | RMSE (mm w.e.) | P-value |
|---|---|---|---|---|---|---|
| uni-PDD v.s. RACMO2.3p2 | 0.7052 | $P < 0.01$ | 0.9416 | 0.091 | $2.16 \times 10^4$ | $P < 0.01$ |
| dist-PDD v.s. RACMO2.3p2 | 0.8052 | $P < 0.01$ | 0.5307 | 0.55 | $1.42 \times 10^4$ | $P < 0.01$ |

**Figure 5 e-f (and 6e-f): Please put a border around the ice shelves as well as it is difficult to see the ice-shelf edge in some places (e.g. Ross, Ronne-Filchner). Additionally, please more explicitly define what the "difference" represents: is it observations – PDD or PDD – observations? Finally, I wonder if it would also be helpful to show relative difference maps. The largest bias appears to be on the Antarctic peninsula (which is in contradiction to Figure 4 and lines 273-275) and I imagine this is just because more melt occurs on the peninsula than elsewhere and thus the bias naturally appears greater.**

Thank you for this suggestion. We will add a border of mask area to the spatial maps in Figure 5 and 6 (Figure 5 and Figure 6 above). The "difference" represents the absolute difference between the PDD outputs and satellite/ RACMO2.3p2 (PDD - satellite/ RACMO2.3p2). We will change the title of the colorbar for clarification.

You are correct that the relative difference explains the opposition between Line 270, Line 291 and Line 298, as the Anonymous Referee #1 also pointed out. The largest absolute bias appears to be on the Antarctic Peninsula and you are correct that this is because more melt occurs on the peninsula. Plotting the relative bias actually reduces the bias on the Peninsula in relation to the ther areas. This is in agreement with our two-sample KS test results in Figure 4. We will add the following two new figure in Appendix C.

[Figure]

**New Figure D 2.** (a) to (f) mean, STD and trend of dist-PDD/ satellite melt days for the period 1979/1980 to 2020/2021, respectively. (g) to (i) relative difference between dist-PDD and satellite melt day mean, STD and trend for the period 1979/1980 to 2020/2021, respectively. Note that for all panels the satellite estimates from 2002/2003 to 2010/2011 are the average of SMMR and SSM/I, and AMSR-E. The satellite estimates from 2012/2013 to 2020/2021 are the average of SMMR and SSM/I, and AMSR-2. For all panels the period 1986/1987, 1987/1988, 1988/1989 and 1991/1992 are omitted.

[Figure]

**New Figure D 4.** (a) to (f) mean, STD and trend of dist-PDD/ RACMO2.3p2 melt amounts for the period 1979/1980 to 2019/2020, respectively. (g) to (i) relative difference between dist-PDD and RACMO2.3p2 melt amount mean, STD and trend for the period 1979/1980 to 2019/2020, respectively.

**Figure 5 h/i (and 6 h/i): These figure panels are quite small and have a lot going on in a relatively small area. I would recommend making them bigger or perhaps making them their own figure and moving to the Supplemental material.**

Thank you for this suggestion. We agree. We will move these panels to their own figures and move them to Appendix C.

[Figure]

**New Figure D 1.** (a) probability histogram for the biases between the dist-PDD and satellite CMS. Red dashed vertical line indicates the mean of all biases. (b) and (c) probability histograms for the biases between the dist-PDD outputs and satellite estimates on mean, STD and trend. Red dashed vertical line indicates the mean of all biases between means. Blue vertical line indicates the mean of all biases between STDs. Black dashed vertical line indicates the mean of all biases between trends. Note that for all panels the satellite estimates from 2002/2003 to 2010/2011 are the average of SMMR and SSM/I, and AMSR-E. The satellite estimates from 2012/2013 to 2020/2021 are the average of SMMR and SSM/I, and AMSR-2.

[Figure]

**New Figure D 3.** (a) probability histogram for the biases between the dist-PDD and RACMO2.3p2 melt amounts. Red dashed vertical line indicates the mean of all biases. (b) and (c) probability histograms for the biases between the dist-PDD outputs and RACMO2.3p2 simulations on mean, STD and trend. Red dashed vertical line indicates the mean of all biases between means. Blue vertical line indicates the mean of all biases between STDs. Black dashed vertical line indicates the mean of all biases between trends.

**L329-330: What changed to cause a greater difference between the PDD and RACMO melt amounts before 1990 than after?**

Thank you for this question. We believe this is related to the melt event of 1982/1983. Please refer to our response to your major comments and our proposed new Appendix E. In addition, one of the reasons could be the decadal climate variability. Turner et al. (2020) reported that there was a positive temperature trend in Antarctic Peninsula for the period 1979–1997 and a negative temperature for the period 1999–2018 (referred to Figure 10, Turner et al. 2020). Because the Antarctic Peninsula is one of the most intensive melting regions in Antarctica, changes in that area can therefore influence the changes of our PDD annual Antarctic melt totals.

We also list some limitations of the PDD model in Section 4.3 which might be helpful to explain the bias of the PDD model to satellite/ RACMO2.3p2. We assume some climatic phenomena that might be able to influence the surface melt that cannot be captured by the PDD model.

**L336: "There are less than 5% computing cells with mismatches in the mean of lower than -15 mm w.e. or larger than +15 mm w.e. (Figure 6h)." – What do you mean by this? Is it "Less than 5% of all computing cells are 15 mm w.e. below or above the mean"?**

Thank you for pointing this out. We will replace at Lines 336–337 from: "...here are less than 5% computing cells with mismatches in the mean of lower than -15 mm w.e. or larger than +15 mm w.e. (Figure 6h). T..." with "...Less than 5% of the total number of all computing cells are 15 mm w.e. below or above the bias on mean....".

**L349-351 "The disagreement in trends, therefore, is actually between the satellite/RACMO2.3p2 and ERA5 2-m temperature, rather than between the satellite/RACMO2.3p2 and the PDD model itself." Can you create a figure to demonstrate this?**

Thank you for pointing this out. Please refer to our Figure C6 in the Appendix C. New Figure C3c and New Figure C5c show a positive trend of PDD melt day and amount over ice shelves in West Antarctic Peninsula and Dronning Maud Land, and the Amery Ice Shelf. This spatial pattern of the positive trend is in agreement with the spatial pattern of the positive trend of mean DJF ERA5 2-m temperature (Figure C6).

**L355-367: I think this paragraph needs to be simplified and condensed. Essentially, you train on two folds to determine the optimal PDD parameters and test on the third by comparing PDD melt with observations/RACMO, correct? I think shortening this section to would help clarify and make it easier for the reader to follow.**

Thank you for pointing this out. Yes, you are correct. We train on two folds to determine the optimal PDD parameters and test on the third by comparing PDD melt with observations/RACMO. According to your comment above on the Section 3.2.2,

[Figure]

**New Figure D 5.** Trend of the mean DJF ERA5 2-m temperature on each computing cell during the period 1979/1980–2019/2020. Black dots mark the trends that are statistically significant (p < 0.05).

we will change at Lines 189–192: "The 3-fold CV has three independent members.   In Member 1, we take the first and second fold to parameterize the PDD model and test the model on the third fold. In Member 2, we take the first and third fold to parameterize the PDD model and test the model on the second fold. In Member 3, we take the second and third fold to parameterize the PDD model and test the model on the first fold.". This information given in Section 3.2.2 overlaps with the information at Lines 354–363. We therefore remove Lines 354–363 to streamline the manuscript and to improve its readability: "         ". We think the Table 2 should be in the "Methods" Section instead of the "Results and discussion" Section. We will move the Table 2 to Section 3.3.2.

 **Figure 7g-l: The histograms are not super helpful for me here. Is there a better way to visualize this? The colors for each subpanel are a bit hard to distinguish.**

Thank you for pointing this out. We agree that the histograms are a bit hard to visually distinguish because they mostly overlap. The overlapping of these histograms indicates that the distribution of parameters of each CV member are analogous to the distribution of the CONTROL parameters.

775 In order to better visualize this, we convert the histograms to curves (Figure 7).

[Figure]

**New Figure 7.** (a) to (f) differences between the $T_0$/ DDF parameterized in each member of the $T_0$/ DDF 3-fold CV and the optimal $T_0$/ DDF, respectively. (g) to (l) probability distributions for the $T_0$/ DDF of each $T_0$/ DDF 3-fold CV and the optimal $T_0$/ DDF, respectively. Black vertical lines indicate the mean of optimal $T_0$s/ DDFs. Red dotted vertical lines indicate the mean of $T_0$/ DDF for each member, respectively. (m) to (r) cumulative CMS/ annual melt amount for satellite estimates/ RACMO2.3p2 simulations, CONTROL (which is the PDD model run with optimal $T_0$ and DDF) and each member for the period of the testing-fold, respectively. We calculate the difference of cumulative CMS/ annual melt amount between each member and the CONTROL, at the end of the testing fold, respectively. (s) to (x) scatter plots for the CMS/ annual melt amount of each 3-fold CV member against the CONTROL, respectively. The Spearman's $\rho$ and its statistical significance, and the slope, RMSE and average bias for the OLS fit, for the testing fold between each member and the CONTROL are calculated, respectively.

**Figure 7s-x: It would be helpful to include other statistics for these scatter plots (RMSE, average bias, slope of line of best fit).**

Thank you for this suggestion. We will add RMSE, average bias and slope of the line of best fit to the Figure 7s-x ((Figure 7) above). The average bias for each panel from (s) to (x) are: 2.1731e-09, -6.1611e-09, -2.2925e-09, 4.3096e-11, -5.1971e-13, 2.2348e-11. The biases are distributed symmetrically to zero therefore the average bias for each CV member are very close to zero.

**Figure 7o and r: I am confused why the CONTROL line is so different from RACMO2.3p2. Should this not look the same as the PDD line in Figure 5 and 6b? Maybe I am misunderstanding what the CONTROL is.**

[Figure]

**Figure 0.** (a) cumulative CMS for satellite estimates and PDD outputs from 1979/1980 to 2020/2021 (with 1986/1987 to 1988/1989 and 1991/1992 omitted. (a) cumulative CMS for satellite estimates and PDD outputs from 1979/1980 to 1995/1996 (with 1986/1987 to 1988/1989 and 1991/1992 omitted

Thank you for pointing this out. You are correct that the CONTROL line on Figure 7o and r should look same as the PDD line in Figure 5 and 6b, and they are exactly the same. The size of the figure and the range of x and y axis introduce the visual bias: zooming into the red box area of Figure 0a and setting the x and y axis to be exactly the same x and y axis of Figure 7o reveals that the PDD line on Figure 0b is exactly the same as the CONTROL line on Figure 7o. Same as the Figure 6a and Figure 7r.

**Technical corrections by Devon Dunmire:**

**L6 – Add "grid" before "cell-level"**

Thank you for this suggestion. We will change in Line 6: "...Here, we construct a novel grid cell-level positive degree-day...".

**L10 – delete "to" in "independently of to the time window..."**

Thank you for this suggestion. We will change in Line 10: "...computing cells as a whole, independently of  the time ...".

**L39 – add "better" before "suited". Also make sure your spelling of "therefor" is consistent throughout the paper. In L54 you use "therefore".**

Thank you for pointing this out. We will change in Line 39: "...input and computational requirements and is therefore better suited...".

**L58: change "the spatial variability of PDD parameters are rarely considered" to "the spatial variability of PDD parameters is rarely considered".**

Thank you for pointing this out. We will change in Line 58: "...the spatial variability of PDD parameters  is rarely considered...."

**L68: Change "varying" to "our spatially varying"**

Thank you for this suggestion. We will change in Line 68: "...our spatially varying model parameters agains...".

**L129: change "most used" to "most commonly used"**

Thank you for this suggestion. We will change in Line 129: "...temperature-index models are the most commonly used method...".

**L161 – Change "multi" to "multiple"**

Thank you for this suggestion. We will change in Line 161: "...there are multiple T0 experiments that...".

**810   L164 – Change "the amount of melt" to "meltwater production". (perhaps do this elsewhere too).**

Thank you for this suggestion. We will change in Line 164: "...number that controls  meltwater production....". We will also change in Line 42: "...which controls  meltwater production....".

**L169 – Change "address" to "determine"**

Thank you for this suggestion. We will change in Line 169: "In order to  determine the optimal DDF,...".

**815   L186 – Add either "melt volume" or "meltwater production" after RACMO2.3p2**

Thank you for this suggestion. We will change in Line 186: "...RACMO2.3p2 melt volume and from the...".

**L190 – I think you can delete "has been developed since the 20th century (Stone, 1974) and"**

Thank you for this suggestion. This comment overlaps with the Major comments by Anonymous Referee #1. We will delete the whole paragraph.

**820   L205 – change "foldis" to "fold is"**

Thank you for pointing this out. We will change in Line 205: "...the third  fold is used...".

**L232 – replace "dominant number" with "majority"**

Thank you for this suggestion. We will change in Line 232: "...the  majority of cells...".

**L234: Change "statistics of T0 s" to "T0 statistics across all grid cells"**

825   Thank you for this suggestion. We will change in Line 234: "...the  T0 statistics across all grid cells....".

**L280: Add "the" before "two CMS time series" and change "are in a generally" to "are generally in"**

Thank you for this suggestion. We will change in Line 280: "...that the two CMS time series are  generally in good agreement...".

**L287: Change "symmetrically to the mean" to "symmetrically around the mean"**

830  Thank you for this suggestion. We will change in Line 287: "...distributed symmetrically  around the mean which...".

**L288: There are two "the"s before "PDD model"**

Thank you for pointing this out. We will change in Line 288: "...see the  PDD model...".

---

## Author Response (AR3)

Author responses to Editor and Reviewer comments on the manuscript

**Statistically parameterizing and evaluating a positive degree-day model to estimate surface melt in Antarctica from 1979 to 2022**

Yaowen Zheng, Nicholas R. Golledge, Alexandra Gossart, Ghislain Picard, and Marion Leduc-Leballeur
submitted to The Cryosphere (https://doi.org/10.5194/tc-2022-192)

We are grateful to the Editor and Reviewers for the time they spent reading and reviewing our manuscript. We have carefully considered each of their comments, which are presented below. The comments from the Editor and Reviewer are shown in **bold text**, our replies are shown in normal text, text from the original manuscript is shown in blue, and proposed changes to
5 the manuscript are shown in red.
* * *
**Editor decision by Brice Noël:**

**Dear Yaowen Zheng and co-authors,**

**I join the reviewer to congratulate you on a revised manuscript that clarifies remaining concerns and much improves**
10 **the robustness of the results. The reviewer has a few minor suggestions, that I invite the authors to consider in a revised version. You can find below some additional edits. When the authors have addressed these minor comments, I am in principle happy to accept the manuscript for publication in The Cryosphere.**

Thank you very much for your comments. We will take into account both your suggestions and those of the reviewer to improve our manuscript.

15 ---

**Editor minor comments:**

**L8: I suggest "... (PDD) model, forced with 2-m air temperature reanalysis data, and spatially parameterized by minimizing ..."**

Thank you for this suggestion. We will replace in Line 8 from: "...force it only with 2-m air temperature reanalysis data, and
20 parameterize it spatially by minimizing the error with respect to..." with "...forced with 2-m air temperature reanalysis data, and spatially parameterized by minimizing the error with respect to...".

**L15-16: PDD estimates of what? Please clarify: melt amounts / extent?**

Thank you for pointing this out. We will replace at Lines 14–17 from: "We find that the PDD estimates change analogously to the variations in the training data with steady statistically significant correlations, and the PDD estimates increase nonlinearly with the temperature perturbations, demonstrating the consistency of our parameterization and the applicability of the PDD model to warmer climate scenarios." with "We find that the PDD melt extent and amounts change analogously to the variations in the training data with steady statistically significant correlations, and the PDD melt amounts increase nonlinearly with the temperature perturbations, demonstrating the consistency of our parameterization and the applicability of the PDD model to warmer climate scenarios.".

**L28-29: Do you mean "that increase ice shelf vulnerability, as meltwater can pond, drain and further contribute to the structural weakness of ice shelves."?**

Thank you for pointing this out. Yes. We will replace at Lines 27–30 from: "Studies have suggested that Antarctic surface melt can impact ice sheet mass balance through surface thinning and runoff, and increasing ice shelf vulnerability that potentially influenced by the production of meltwater which can pond, drain and contribute to the structural weakness of ice shelves (Glasser and Scambos, 2008; Bell et al., 2018; Stokes et al., 2022)." with "Studies have suggested that Antarctic surface melt can impact ice sheet mass balance through surface thinning and runoff that can increase ice shelf vulnerability, as meltwater can pond, drain and further contribute to the structural weakness of ice shelves (Glasser and Scambos, 2008; Bell et al., 2018; Stokes et al., 2022).".

**L36: The authors could in 'one sentence' introduce Regional Climate Models that use SEB modules to estimate surface melt. See e.g. Wessem et al. (2018), Agosta et al. (2019) and models presented in the model intercomparison of Mottram et al. (2021).**

Thank you for this suggestion. We agree. We will change at Lines 35–38: "Continental-scale spaceborne observations of surface melt are limited to the satellite era (1979–present), meaning that current estimates of Antarctic surface melt are typically derived from surface energy balance (SEB) or positive degree-day (PDD) models. SEB models are employed in Regional Climate Models such as the Regional Atmospheric Climate MOdel (RACMO) (Van Wessem et al., 2018) and Modèle Atmosphérique Régional (MAR) (Agosta et al., 2019). PDD models are employed in ice sheet models such as the SImulation COde for POLythermal Ice Sheets (SICOPOLIS) (Nowicki et al., 2013), Ice Sheet System Model (ISSM) (Larour et al., 2012), and Parallel Ice Sheet Model (PISM) (Winkelmann et al., 2011). SEB models require diverse and detailed input data that are not always available and require considerable computational resources."

**L65: Please add a reference to the RACMO2 data set.**

Thank you for pointing this out. We agree. We will change in Line 65: "days from three satellite products and the Regional Atmospheric Climate Model version 2.3p2 (RACMO2.3p2) (Van Wessem et al., 2018) surface melt".

**L155: "in the overlapping years."**

Thank you for pointing this out. We agree. We will replace in Line 155 from: "...in their overlapped years..." with "...in the overlapping years...".

**L219: 0 should be a subscript in 'To'.**

Thank you for pointing this out. We agree. We will replace in Line 219 from: "...optimal T0 values selected through 151 T0 experiments..." with "...optimal $T_0$ values selected through 151 $T_0$ experiments...".

**Table 3 and 4 captions: Please add "Ordinary Least Square" to define the acronym "OLS".**

Thank you for pointing this out. We agree. We will change in Table 3 and 4 captions: "...for the Ordinary Least Squares (OLS) fit..."

**L343: I suggest "... because of the correlation between ..."**

Thank you for this suggestion. We agree. We will replace in Line 343 from: "...because of the linear relationship between air temperature and surface melt..." with "...because of the correlation between air temperature and surface melt...".

**L370: Here and elsewhere could you consider "we show that" or "we find that" instead of "we see".**

Thank you for this suggestion. We agree. There are 13 "we see" in total. We will replace these "we see" with "we show that".

**L370: Replace 'statistically significantly, strongly' by 'significantly'.**

Thank you for this suggestion. We agree. We will replace at Lines 370–371 from: "...dist-PDD estimates are statistically significantly, strongly ($\rho \geq 0.99$, $p \leq 0.05$) correlated..." with "...dist-PDD estimates are significantly ($\rho \geq 0.99$, $p \leq 0.05$) correlated...".

**L373-377: Cut this long sentence into two sentences.**

Thank you for this suggestion. We agree. We will replace at Lines 373–377 from: "Although the $T_0$ Member 1 dist-PDD estimates and dist-PDD CONTROL estimates are strongly correlated to the training fold (black dots in Figure 7s), which is not surprising as the $T_0$ Member 1 dist-PDD is parameterized by those dist-PDD CONTROL estimates, the $T_0$ Member 1 dist-PDD estimates and dist-PDD CONTROL estimates are not statistically significantly correlated ($\rho = 0.19$, p $\geq 0.05$) to the testing fold (red dots, Figure 7s)." with "The $T_0$ Member 1 dist-PDD estimates and dist-PDD CONTROL estimates are strongly correlated to the training fold (black dots in Figure 7s), which is not surprising as the $T_0$ Member 1 dist-PDD is parameterized by those dist-PDD CONTROL estimates. The $T_0$ Member 1 dist-PDD estimates and dist-PDD CONTROL estimates are not statistically significantly correlated ($\rho = 0.19$, p $\geq 0.05$) to the testing fold (red dots, Figure 7s). ".

**L383: I suggest "might explain" instead of "might be".**

Thank you for this suggestion. We agree. We will replace in Line 382 from: "...over the testing-fold period might be the disagreement between..." with "...over the testing-fold period might explain the disagreement between...".

**L388-389: "To that is parameterized by … data sample used to parameterize … data length used to estimate …".**

Thank you for this suggestion. We agree. We will replace at Lines 388–390 from: "optimal $T_0$ that parameterized by the full 38-year period. However, the data sample that used to parameterize the Member 1 $T_0$ is only 2/3 the full data length which parameterized the optimal $T_0$, giving us less confidence on the reliability of the Member 1 $T_0$s for the full 38-year period." with "optimal $T_0$ that is parameterized by the full 38-year period. However, the data sample used to parameterize the Member 1 $T_0$ is only 2/3 the full data length used to estimate the optimal $T_0$, giving us less confidence on the reliability of the Member 1 $T_0$s for the full 38-year period.".

**Figure 8 caption L5: "shaded areas indicate …".**

Thank you for pointing this out. We agree. We will replace Figure 8 caption Line 5 from: "...shaded areas indicates the..." with "...shaded areas indicate the...".

**L396: Do you mean "is expected to increase the occurrence of temperatures above …".**

Thank you for this suggestion. Yes. We will replace in Line 396 from: "...is expected to allow more temperatures above the threshold to produce more melt days, and vice versa." with "...is expected to increase the occurrence of temperatures above the threshold to produce more melt days, and vice versa.".

**L401-403: "sensitive . . . to" instead of "sensitive . . . on".**

Thank you for pointing this out. We agree. We will replace at Lines 401–403 from: "...is less sensitive than the satellite estimates on the low melt scenario, where the dist-PDD estimates only decrease by 9.78% for the integrated 38-year CMS, when the satellite estimates decrease by 10%. Although the dist-PDD model is more sensitive than the satellite estimates on the high melt scenario..." with "...is less sensitive than the satellite estimates to the low melt scenario, where the dist-PDD estimates only decrease by 9.78% for the integrated 38-year CMS, when the satellite estimates decrease by 10%. Although the dist-PDD model is more sensitive than the satellite estimates to the high melt scenario...".

**L402: "while" instead of "when".**

Thank you for pointing this out. We agree. We will replace in Line 402 from: "...38-year CMS, when the..." with "...38-year CMS, while the...".

**L443: "surface melt estimates from using uni-PDD."**

Thank you for this suggestion. We agree. We will replace at Lines 443–444 from: "...Antarctic surface melt estimations from using spatially uniform PDD parameters (uni-PDD),..." with "...Antarctic surface melt estimates from using uni-PDD,...".

**L472-473: The link to IMBIE-3 data does not work, please update.**

Thank you for pointing this out. We will replace at Lines 472–473 from: "The Zwally Antarctic drainage basin (Zwally et al., 2012) data are available from http://imbie.org/imbie-3/drainage-basins/." with "The Zwally Antarctic drainage basin (Zwally et al., 2012) data are available from http://imbie.org/imbie-3/drainage-basins/ and https://earth.gsfc.nasa.gov/cryo/data/polar-altimetry/antarctic-and-greenland-drainage-systems (last access: 18 July 2023).".

**Figure E1 c, f and i and caption: 'ERA5' instead of 'EAR5'.**

Thank you for pointing this out. We will replace Figure E1 with the New Figure E1.

**L518: "first decade and almost overlap for the rest"**

Thank you for this suggestion. We agree. We will replace in Line 518 from: "...first decade and remain almost completely overlapped for the rest of the time period..." with "...first decade and almost overlap for the rest of the time period...".

[Figure]

**New Figure E 1.** (a) and (d) 1982/1983 dist-PDD/ satellite meltday anomaly to the dist-PDD/ satellite mean meltday over the period 1979/1980–2020/2021 (with 1982/1983, 1986/1987, 1987/1988, 1988/1989 and 1991/1992 omitted). (g) absolute differences between 1982/1983 dist-PDD and satellite meltday. (b) and (e) 1982/1983 dist-PDD/ RACMO2.3p2 melt amount anomaly to the dist-PDD/ RACMO2.3p2 mean melt amount over the period 1979/1980–2019/2020 (with 1982/1983 omitted). (h) absolute differences between 1982/1983 dist-PDD and RACMO2.3p2 melt amount. (c) and (f) 1982/1983 DJF ERA5/ RACMO2.3p2 2-m air temperature anomaly to the DJF ERA5/ RACMO2.3p2 mean 2-m air temperature over the period 1979/1980–2019/2020 (with 1982/1983 omitted). (i) absolute differences between 1982/1983 DJF EAR5 and RACMO2.3p2 2-m air temperature. Note that for all panels the satellite estimates from 2002/2003 to 2010/2011 are the average of SMMR and SSM/I, and AMSR-E. The satellite estimates from 2012/2013 to 2020/2021 are the average of SMMR and SSM/I, and AMSR-2.

**Minor comments by Devon Dunmire:**

**L23: Recently published, Banwell et al (2023) (https://doi.org/10.1029/2023GL102744) actually found a significant (but small) decrease in AIS ice shelf melt days using a snow model and microwave observations**

Thank you for pointing this out. We will replace at Lines 23–24 from: "Antarctic ice shelves show no statistically significant trend for the annual melt days (Johnson et al., 2022) and also no significant increase in melt amount in East Antarctica in the past 40 years (Stokes et al., 2022)." with "Antarctic ice shelves show statistically significant negative trend for the annual melt days (Banwell et al., 2023) and no significant increase in melt amount in East Antarctica in the past 40 years (Stokes et al., 2022).".

**L215 – fix T0 (change to subscript)**

Thank you for pointing this out. This comment is overlapped with the seventh comment by the Editor. We will replace in Line 219 from: "...optimal T0 values selected through 151 T0 experiments..." with "...optimal $T_0$ values selected through 151 $T_0$ experiments...".

**L343 – I am confused by the statement "because of the linear relationship between air temperature and surface melt". Is there a typo here? Doesn't Figure 9 (and other works) demonstrate a non-linear relationship between air temperature and surface melt?**

Thank you for pointing this out. This comment is overlapped with the ninth comment by the Editor. We will replace in Line 343 from: "...because of the linear relationship between air temperature and surface melt..." with "...because of the correlation between air temperature and surface melt...".

**Figure 7 – Specify in the caption that this is for dist-PDD (same for Figure 8)**

Thank you for this suggestion. We agree. We will add 'This analysis is based on dist-PDD.' at the end of the captions for Figure 7 and 8.

**L415 – Please add the Banwell et all 2023 citation with Bell et al and Trusel et al.**

Thank you for this suggestion. We agree. We will replace in Line 415 from: "... that reported by other studies (Trusel et al., 2015; Bell et al., 2018),..." with "... that reported by other studies (Trusel et al., 2015; Bell et al., 2018; Banwell et al., 2023),...".

**145 L435 – I would recommend elaborating that the "dist-PDD" and "uni-PDD" are "spatially varying" and "spatially uniform" PDDs here because it is the start of the conclusion.**

Thank you for this suggestion. We agree. We will replace in Line 435 from: "We have constructed a dist-PDD model and a uni-PDD model based on the..." with "We have constructed a PDD model with spatially varying PDD parameters (dist-PDD) and a PDD model with spatially uniform PDD parameters (uni-PDD) based on the...".

**150 L447 – "Underestimation" of what?**

Thank you for this suggestion. We agree. We will replace in Line 447 from: "...with the exception of an underestimation in the ice shelves of the western..." with "...with the exception of an underestimation of melt days and amounts in the ice shelves of the western...".

**Appendix - Figure D4 is not referenced anywhere and should b referenced somewhere in the text**

155 Thank you for pointing this out. We agree. We will change at Lines 325–327: "Figure 6d to i show the spatial maps for the difference between the mean, STD and trend of the dist-PDD/ uni-PDD annual melt amount and RACMO2.3p2 mean annual melt amount for the period from 1979/1980 to 2019/2020. The spatial maps for the mean, STD and trend of the dist-PDD/ uni-PDD annual melt amount and RACMO2.3p2 mean annual melt amount for the same period are shown in Figure D4 in the Appendix D. Consistent with the PDD melt day estimates, using the dist-PDD model improves the accuracy of estimating 160 surface melt amount compared to".

**References**

Agosta, C., Amory, C., Kittel, C., Orsi, A., Favier, V., Gallée, H., van den Broeke, M. R., Lenaerts, J., van Wessem, J. M., van de Berg, W. J., et al.: Estimation of the Antarctic surface mass balance using the regional climate model MAR (1979–2015) and identification of dominant processes, The Cryosphere, 13, 281–296, 2019.

165 Banwell, A. F., Wever, N., Dunmire, D., and Picard, G.: Quantifying Antarctic-Wide Ice-Shelf Surface Melt Volume Using Microwave and Firn Model Data: 1980 to 2021, Geophysical Research Letters, 50, e2023GL102 744, 2023.

Bell, R. E., Banwell, A. F., Trusel, L. D., and Kingslake, J.: Antarctic surface hydrology and impacts on ice-sheet mass balance, Nature Climate Change, 8, 1044–1052, 2018.

Glasser, N. and Scambos, T. A.: A structural glaciological analysis of the 2002 Larsen B ice-shelf collapse, Journal of Glaciology, 54, 3–16, 170 2008.

Johnson, A., Hock, R., and Fahnestock, M.: Spatial variability and regional trends of Antarctic ice shelf surface melt duration over 1979–2020 derived from passive microwave data, Journal of Glaciology, 68, 533–546, 2022.

Larour, E., Seroussi, H., Morlighem, M., and Rignot, E.: Continental scale, high order, high spatial resolution, ice sheet modeling using the Ice Sheet System Model (ISSM), Journal of Geophysical Research: Earth Surface, 117, 2012.

175 Nowicki, S., Bindschadler, R. A., Abe-Ouchi, A., Aschwanden, A., Bueler, E., Choi, H., Fastook, J., Granzow, G., Greve, R., Gutowski, G., et al.: Insights into spatial sensitivities of ice mass response to environmental change from the SeaRISE ice sheet modeling project I: Antarctica, Journal of Geophysical Research: Earth Surface, 118, 1002–1024, 2013.

Stokes, C. R., Abram, N. J., Bentley, M. J., Edwards, T. L., England, M. H., Foppert, A., Jamieson, S. S., Jones, R. S., King, M. A., Lenaerts, J. T., et al.: Response of the East Antarctic Ice Sheet to past and future climate change, Nature, 608, 275–286, 2022.

180 Trusel, L. D., Frey, K. E., Das, S. B., Karnauskas, K. B., Munneke, P. K., Van Meijgaard, E., and Van Den Broeke, M. R.: Divergent trajectories of Antarctic surface melt under two twenty-first-century climate scenarios, Nature Geoscience, 8, 927–932, 2015.

Van Wessem, J. M., Van De Berg, W. J., Noël, B. P., Van Meijgaard, E., Amory, C., Birnbaum, G., Jakobs, C. L., Krüger, K., Lenaerts, J., Lhermitte, S., et al.: Modelling the climate and surface mass balance of polar ice sheets using RACMO2–Part 2: Antarctica (1979–2016), The Cryosphere, 12, 1479–1498, 2018.

185 Winkelmann, R., Martin, M. A., Haseloff, M., Albrecht, T., Bueler, E., Khroulev, C., and Levermann, A.: The Potsdam Parallel Ice Sheet Model (PISM-PIK) Part 1: Model description, The Cryosphere, 5, 715–726, https://doi.org/10.5194/tc-5-715-2011, 2011.

Zwally, H. J., Giovinetto, M. B., Beckley, M. A., and Saba, J. L.: Antarctic and Greenland drainage systems, GSFC cryospheric sciences laboratory, 2012.

---

## Author Response (AR4)

Author responses to Editor comments on the manuscript

**Statistically parameterizing and evaluating a positive degree-day model to estimate surface melt in Antarctica from 1979 to 2022**

Yaowen Zheng, Nicholas R. Golledge, Alexandra Gossart, Ghislain Picard, and Marion Leduc-Leballeur
submitted to The Cryosphere (https://doi.org/10.5194/tc-2022-192)

We are grateful to the Editor for the time they spent reading and reviewing our manuscript. We have carefully considered each of their comments, which are presented below. The comments from the Editor are shown in **bold text**, our replies are shown in normal text, text from the original manuscript is shown in blue, and proposed changes to the manuscript are shown

5   in red.
* * *
**Editor decision by Brice Noël:**

**Dear Yaowen Zheng and co-authors,**

**Thank you for submitting your revised manuscript and response letter. Your clarifications and corrections have**

10   **addressed all remaining comments. Therefore, I am happy to accept your manuscript for publication in The Cryosphere. Before publication, you should consider the technical corrections listed below.**

**Congratulations on the acceptance of your manuscript, and on your thorough revisions!**

**Thank you again for publishing your research with TC,**

Thank you very much for your comments. We will take into account your technical corrections to improve our manuscript.

15
* * *
**Editor technical corrections:**

**L382: "estimates are not significantly correlated" statistically is not necessary as you provide the statistics in brackets.**

Thank you for this suggestion. We agree. We will replace in Line 382 from: "estimates are not statistically significantly correlated..." with "estimates are not significantly correlated...".

**L406-409: "is less sensitive to the low melt scenario than the satellite estimates, as the dist-PDD ... while the satellite ..."
and "is less sensitive to the high melt scenario than the satellite estimates, ...".**

Thank you for this suggestion. We agree. We will replace at Lines 406–409 from: "Figure 8e shows that the dist-PDD model is less sensitive than the satellite estimates to the low melt scenario, where the dist-PDD estimates only decrease by 9.78% for the integrated 38-year CMS, while the satellite estimates decrease by 10%. Although the dist-PDD model is more sensitive than the satellite estimates to the high melt scenario, where we show that that dist-PDD increases by 10.84% on the 38-year integrated CMS with the 10% increase of the satellite estimates, this increase" with "Figure 8e shows that the dist-PDD model is less sensitive to the low melt scenario than the satellite estimates, as the dist-PDD estimates only decrease by 9.78% for the integrated 38-year CMS while the satellite estimates decrease by 10%. Although the dist-PDD model is more sensitive to the high melt scenario than the satellite estimates, where we show that dist-PDD increases by 10.84% on the 38-year integrated CMS with the 10% increase of the satellite estimates, this increase"

**L420: "that is reported by other studies".**

Thank you for this suggestion. We agree. We will replace in Line 420 from: "...relationship that reported by other studies..." with "...relationship that is reported by other studies...".

**L440-441: "We have constructed a PDD model with spatially varying parameters (dist-PDD) and with spatially uniform parameters (uni-PDD) based on the temperature-melt relationship".**

Thank you for this suggestion. We agree. We will replace at Lines 440–441 from: "We have constructed a PDD model with spatially varying PDD parameters (dist-PDD) and a PDD model with spatially uniform PDD parameters (uni-PDD) based on the temperature-melt relationship..." with "We have constructed a PDD model with spatially varying parameters (dist-PDD) and with spatially uniform parameters (uni-PDD) based on the temperature-melt relationship...".

**L449: "We found that our dist-PDD model improves the accuracy of Antarctic surface melt estimates compared to the uni-PDD setting, and has ...".**

Thank you for this suggestion. We agree. We will replace at Lines 448–449 from: "...We found that our dist-PDD model improves accuracy on Antarctic surface melt estimates from using uni-PDD, and has the ability to capture the..." with "...We found that our dist-PDD model improves the accuracy of Antarctic surface melt estimates compared to the uni-PDD setting, and has the ability to capture the...".

**L511: "Our dist-PDD model shows significant negative bias in both surface melt days and amounts compared to ...".**

Thank you for this suggestion. We agree. We will replace at Lines 511-512 from: "...Our dist-PDD model is significantly negatively biased towards both surface melt days and surface melt amounts compared to satellite estimates and RACMO2.3p2 simulations..." with "...Our dist-PDD model shows significant negative bias in both surface melt days and amounts compared to satellite estimates and RACMO2.3p2 simulations...".
* * *
**Additional changes by authors:**

We acknowledge the acceptance of this manuscript. We are publishing the annual dist-PDD and uni-PDD models data alongside this manuscript. However, before the final publication of this manuscript, we do not know its DOI. We have reserved a Zenodo DOI for the publication of the annual dist-PDD and uni-PDD models data. Once this manuscript is published and we know the DOI, we will publish the data on Zenodo with the reserved DOI: https://doi.org/10.5281/zenodo.7131459.

We will replace at Lines 480-482 from: "...The annually PDD model data (this study) is available in this study. Higher temporal resolution (monthly, daily and hourly) PDD model data (this study) is available by contacting yaowen.zheng@vuw.ac.nz." with "...The annual dist-PDD and uni-PDD models data from this study are available at https://doi.org/10.5281/zenodo.7131459. Data with higher temporal resolution (monthly, daily, and hourly) for dist-PDD and uni-PDD models from this study can be obtained by contacting yaowen.zheng@vuw.ac.nz."